# AugMask: Training Diffusion Models on Incomplete Tabular Data via Stochastic Augmentation and Masking

Jungkyu Kim [1]   Taeyoung Park [1*]   Kibok Lee [1*]

## Abstract

Score-based diffusion models have emerged as prominent deep generative models; however, their application to tabular data remains challenging because their backbones assume fully specified inputs, whereas real-world tabular data often contain missing values. We propose **AugMask**, a plug-and-play training framework that adapts missing-unaware backbones to incomplete data by separating conditioning from supervision. Aug-Mask 1) constructs numeric inputs via conditional stochastic augmentation using lightweight auxiliary models, and 2) applies denoising supervision only to observed coordinates. In effect, augmented missing entries serve as *uncertain conditioning context rather than training targets*. We connect this training rule to a Rao–Blackwellized objective and show that marginalizing missing entries yields a variance-weighted sensitivity penalty, discouraging over-reliance on uncertain completions. Across diverse datasets and missingness regimes, AugMask enables standard diffusion-based tabular generators to outperform specialized missing-aware baselines.

## 1. Introduction

Deep generative models have been successfully applied to synthesize tabular data, addressing challenges including data scarcity and privacy (Jordon et al., 2018). However, real-world tabular data often contain missing values, hindering the direct application of these models to practical settings (Cui et al., 2024). Standard generative architectures are typically designed under the assumption of fully observed inputs. This makes them *missing-unaware*, creat-

ing a critical mismatch with the inherent incompleteness of real-world tabular data.

Bridging this gap is non-trivial because standard neural-network-based generators cannot inherently handle undefined values (NaNs) and therefore require fully specified inputs. A common way is to make a model *missing-aware* by first replacing missing entries with a constant placeholder, such as zeros, column means, or special tokens, and then modifying the objective or inputs to reduce the placeholder's influence (Mattei & Frellsen, 2019; Nazabal et al., 2020; Ouyang et al., 2023). Depending on the method, this may involve weighting or masking the loss to observed coordinates, or providing the observation mask as the condition. However, zero-filling forces the model to treat missing entries identically to valid zeros (Collier et al., 2020) and can induce *sparsity bias*, making predictions depend on the missingness level (Yi et al., 2019). In turn, constant placeholders induce an *artificially consistent* input pattern and can introduce imputation-induced distortions in the learned network (Ipsen et al., 2022). These modifications can change which coordinates contribute to the objective, but not what the model receives as input. Even when missing coordinates are excluded from the loss, the filled values remain in the input and can become deterministic missingness cues.

On the other hand, instead of using a constant placeholder, missing entries can be iteratively updated during training via *self-imputation*, as in diffusion-based *Expectation-Maximization* (EM) approaches such as DiffPuter (Zhang et al., 2025). While this can mitigate imputation-induced distortions from constant placeholders, it introduces a fundamental optimization challenge, where the model is trained on its own evolving predictions. Because diffusion objectives *seek diversity rather than accuracy* (Chen et al., 2024), such a strategy can yield overly dispersed completions in practice. Feeding these shifting completions back into training creates a *moving target*, resulting in noisy gradients and non-stationary dynamics that complicate convergence.

These issues suggest that training from incomplete data should provide stable numeric inputs without treating filled-in missing values as ground-truth targets. We propose **AugMask**, a *training strategy* that separates these roles. It uses *stochastic augmentation* to construct plausible context-

---

*Co-corresponding authors. [1]Department of Statistics and Data Science, Yonsei University, South Korea. Correspondence to: Kibok Lee <kibok@yonsei.ac.kr>, Taeyoung Park <tpark@yonsei.ac.kr>.

*Proceedings of the 43rd International Conference on Machine Learning*, Seoul, South Korea. PMLR 306, 2026. Copyright 2026 by the author(s).

dependent inputs and *masks* the loss so that only originally observed coordinates contribute to training. The stochasticity of the augmentation is crucial as it encodes uncertainty inherent in missingness. High conditional variance discourages reliance on uncertain fills, while low variance permits stronger use as context. A local analysis of the objective formalizes this by showing that marginalizing missing coordinates over plausible completions yields a *variance-weighted sensitivity penalty*. Our contributions are as follows:

- **A training principle for incomplete tabular diffusion.** We introduce AugMask[1], which separates conditioning from supervision: stochastic completions provide numeric, uncertainty-aware context, while the denoising loss is applied only to observed coordinates.

- **A marginalization view of stochastic augmentation.** We analyze AugMask through a Rao–Blackwellized objective and show that marginalizing uncertain missing entries yields a variance-weighted sensitivity penalty, explaining why stochastic completions are useful beyond point-wise imputation accuracy.

- **A plug-and-play recipe for missing-unaware backbones.** We provide a lightweight framework that adapts standard score-based diffusion models for incomplete tabular data without architectural changes, and validate it across datasets, missingness regimes, and missing-aware baselines.

## 2. Preliminaries

We observe partially observed samples $(\mathbf{x}^{\text{obs}}, \mathbf{m})$ and train a denoiser $D_\theta$ by applying the loss only to truly observed coordinates. Restricting the denoising loss to observed coordinates is a standard choice under missingness, but it does *not* solve by itself the practical issue that standard neural networks cannot process inputs containing NaN. Missing parts must therefore be replaced with numeric values, which then become part of the denoiser's *conditioning* signal.

**Score-based diffusion models.** Let $\mathbf{X}^0 \sim p_0$ denote a clean sample at diffusion time $t = 0$ from the data, and sample $t \sim \pi(t)$, where $\pi$ is a distribution over noise levels with the corresponding noise scale $\sigma(t)$. The forward process that corrupts the clean sample is defined as $q(\mathbf{X}^t \mid \mathbf{X}^0, t) = \mathcal{N}(\mathbf{X}^0, \sigma(t)^2 I)$. Under Gaussian corruption, the denoising score matching is equivalent (up to a $t$-dependent scaling) to the denoising autoencoder (DAE) objective (Vincent, 2011). We use the DAE form for clarity:

$$\mathcal{L}_\theta^{\text{DAE}} = \mathbb{E}_{t \sim \pi, \, \mathbf{X}^0, \, \mathbf{X}^t \sim q(\cdot \mid \mathbf{X}^0, t)} \left[ \left\| D_\theta(\mathbf{X}^t, t) - \mathbf{X}^0 \right\|_2^2 \right]. \tag{1}$$

**Incomplete data.** Let $\mathbf{X} = (X_1, \ldots, X_d)^\top$ denote a random tabular sample, and let $\mathbf{M} \in \{0, 1\}^d$ be an observation mask with $M_j = 1$ if $X_j$ is observed. Let $\mathbf{X}^{\text{obs}}$ denote the observed parts of $\mathbf{X}$ induced by $\mathbf{M}$; accordingly, we observe a realization of $(\mathbf{X}^{\text{obs}}, \mathbf{M})$ rather than that of the complete $\mathbf{X}$, which we write as $(\mathbf{x}^{\text{obs}}, \mathbf{m})$. In our implementation, we write missing entries as:

$$\tilde{x}_j = \begin{cases} x_j^{\text{obs}}, & m_j = 1, \\ \text{NaN}, & m_j = 0. \end{cases} \tag{2}$$

Our main method focuses on ignorable missingness, namely Missing Completely At Random (MCAR) and Missing At Random (MAR). Details are provided in Appendix C.3.

**Masked denoising objective and the completion problem.** Following the likelihood-bound perspective for diffusion models (Song et al., 2021), prior work shows that restricting the denoising loss to observed coordinates gives a principled objective under incomplete data (Ouyang et al., 2023). In practice, this objective is evaluated as an empirical average over partially observed samples $(\mathbf{x}^{\text{obs}}, \mathbf{m})$ from the training set, with randomness from $t \sim \pi$ and the forward noising process. However, since $D_\theta$ cannot take NaN as input, missing entries must be replaced with *numeric completions*. A common choice is a constant placeholder $\mathbf{c}^{\text{pl}}$ (e.g., zero-filling) that forms

$$\bar{x}_j^0 := m_j \, x_j^{\text{obs}} + (1 - m_j) \, c_j^{\text{pl}}. \tag{3}$$

With $\bar{\mathbf{x}}^0$ as input to the forward process, the denoising objective over the observed coordinates becomes

$$\mathcal{L}_\theta^{\text{MissDiff}} := \mathbb{E}_{t \sim \pi, \bar{\mathbf{x}}^t \sim q(\cdot \mid \bar{\mathbf{x}}^0, t)} \left[ \left\| \mathbf{m} \odot \left( D_\theta(\bar{\mathbf{x}}^t, t) - \bar{\mathbf{x}}^0 \right) \right\|_2^2 \right]. \tag{4}$$

Notably, masking removes supervision on missing coordinates, but it *does not remove conditioning*. The denoiser still receives the completion values $\mathbf{c}^{\text{pl}}$ as part of its input, so the model may learn to depend on a deterministic signal correlated with missingness.

## 3. Method

### 3.1. Augmented Data

To provide a fully numeric input without forcing the denoiser to condition on a deterministic placeholder, we replace missing entries with samples from a context-dependent augmentation rule. For each feature $j$, let $T(\cdot \mid \mathbf{x}^{\text{obs}}, \mathbf{m})$ denote the distribution used to fill $X_j$ when $m_j = 0$. We draw $Z_j \sim T(\cdot \mid \mathbf{x}^{\text{obs}}, \mathbf{m})$ and define the augmented clean input

$$x_j^{A,0} := m_j \, x_j^{\text{obs}} + (1 - m_j) \, Z_j. \tag{5}$$

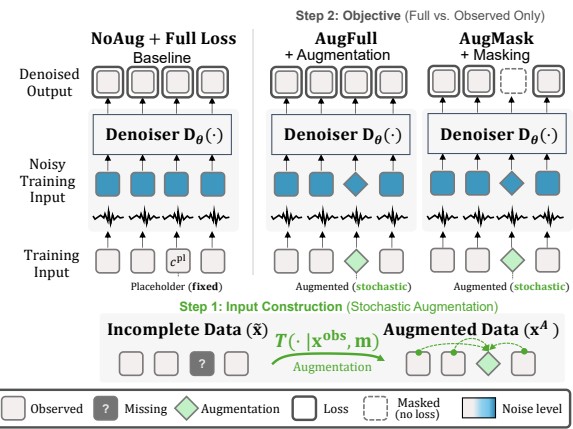

*Figure 1.* **AugMask: Two-step training strategy.** Stochastic conditional augmentation for inputs and loss masking for targets.

The constant-placeholder in Eq. (3) is recovered as the degenerate case $T = \delta_{c_j^{\mathrm{pl}}}$, for which $\mathrm{Var}_T(Z_j \mid \mathbf{x}^{\mathrm{obs}}, \mathbf{m}) = 0$. The choice of $T$ determines the information that enters the denoiser through the completed input. We distinguish three families. A **constant placeholder** is context-free and deterministic. A **deterministic conditional augmentation** is context-dependent but still uses a point estimate, e.g., $T = \delta_{\hat{\mu}_j(\mathbf{x}^{\mathrm{obs}}, \mathbf{m})}$. A **stochastic conditional augmentation** is both context-dependent and non-degenerate, sampling completions with nonzero conditional variance. We use the term *augmentation* deliberately. Sampled completions provide possible conditioning contexts for the denoiser, not ground-truth targets to reconstruct.

## 3.2. AugMask: Augmented Conditioning with Observed-Only Supervision

AugMask trains a feature-space denoiser on incomplete tabular data through two steps, illustrated in Fig. 1. First, **augmentation** constructs a fully numeric input $\mathbf{x}^{A,0}$ by sampling missing entries from $T(\cdot \mid \mathbf{x}^{\mathrm{obs}}, \mathbf{m})$. Second, **observed-only supervision** applies the denoising loss only to the coordinates that were originally observed. Thus, augmented missing entries enter the model through noisy input $\mathbf{x}^{A,t}$, but are not treated as supervised targets.

Given $\mathbf{x}^{A,0}$ and its noised version $\mathbf{x}^{A,t} \sim q(\cdot \mid \mathbf{x}^{A,0}, t)$, AugMask optimizes

$$\mathcal{L}_\theta^{\mathrm{AugMask}} := \mathbb{E}_{\mathbf{Z}, t, \mathbf{x}^{A,t}} \left[ \left\| \mathbf{m} \odot \left( D_\theta(\mathbf{x}^{A,t}, t) - \mathbf{x}^{A,0} \right) \right\|_2^2 \right]. \tag{6}$$

The expectation includes incomplete training sample, augmentation draw, diffusion time, and forward noise. The mask removes supervision on the augmented values, while the augmented values provide the conditioning context.

For comparison, **AugFull** uses the same augmented inputs

but applies the denoising loss to all coordinates:

$$\mathcal{L}_\theta^{\mathrm{AugFull}} := \mathbb{E}\left[ \left\| D_\theta(\mathbf{x}^{A,t}, t) - \mathbf{x}^{A,0} \right\|_2^2 \right]$$
$$= \mathcal{L}_\theta^{\mathrm{AugMask}} + \mathbb{E}\left[ \left\| (\mathbf{1} - \mathbf{m}) \odot \left( D_\theta(\mathbf{x}^{A,t}, t) - \mathbf{x}^{A,0} \right) \right\|_2^2 \right]. \tag{7}$$

Thus, AugFull treats completed missing entries as targets, whereas AugMask uses them only as context. This comparison isolates whether the model should fit the completed values or only use them as context.

*Remark* 3.1 (AugMask vs. AugFull). AugFull can reduce estimation variance when the augmentation rule $T$ is accurate and informative, but it can introduce bias when $T$ is misspecified because the model is supervised on inaccurately completed values. AugMask avoids this pseudo-target bias by not supervising originally missing coordinates. Appendix B gives a Gaussian toy example in which the additional AugFull term transitions from variance reduction to bias domination as augmentation mismatch increases.

*Remark* 3.2 (Scope of applicability). AugFull is broadly applicable to missing-unaware generative models because it only requires complete numeric inputs. AugMask, in contrast, relies on an observed-coordinate reconstruction objective and is therefore most natural for denoising or score-based models operating directly in feature space. Extending observed-only supervision to latent diffusion, VAEs, or adversarial objectives is nontrivial because missing coordinates interact with inference, decoding, or discrimination; we leave these extensions to future work.

## 3.3. Theoretical Analysis: Conditional Variance as Sensitivity Regularization

We next explain why stochasticity in $T$ is useful beyond point-wise imputation accuracy. The analysis studies how the denoiser's reconstruction of observed coordinates changes as an augmented missing coordinate varies.

**One-missing-coordinate setup.** To build intuition, suppose only coordinate $j$ may be missing and the remaining block $-j$ is observed. Fix a clean observed block $\mathbf{x}_{-j}^0$. For a completion value $z$, define $\mathbf{x}^{A,0}(z) := (\mathbf{x}_{-j}^0, z)$ and $\mathbf{x}^{A,t}(z) = \mathbf{x}^{A,0}(z) + \sigma(t)\boldsymbol{\varepsilon}$ with $\boldsymbol{\varepsilon} \sim \mathcal{N}(\mathbf{0}, I)$. Let

$$g_\theta(z) := \mathbb{E}_{t, \boldsymbol{\varepsilon}} \left[ \left\| D_\theta(\mathbf{x}^{A,t}(z), t)_{-j} - \mathbf{x}_{-j}^0 \right\|_2^2 \right] \tag{8}$$

be the observed-coordinate reconstruction loss induced by completion $z$. Any augmentation rule $T$ induces the marginalized per-sample objective

$$l^T(\theta; \mathbf{x}_{-j}^0) := \mathbb{E}_{Z_j \sim T(\cdot \mid \mathbf{x}_{-j}^0)} \left[ g_\theta(Z_j) \right]. \tag{9}$$

As an oracle benchmark within this family, define the **Rao–Blackwellized (RB) ideal** by using the true conditional

distribution:

$$l^{\mathrm{RB}}(\theta; \mathbf{x}_{-j}^0) := \mathbb{E}_{Z_j \sim p_{\mathrm{data}}(\cdot | \mathbf{x}_{-j}^0)}\left[g_\theta(Z_j)\right]. \quad (10)$$

**Proposition 3.3** (Local RB approximation)**.** *Let* $\mu_j :=$ $\mathbb{E}[Z_j \mid \mathbf{x}_{-j}^0]$ *and* $\sigma_j^2 := \mathrm{Var}(Z_j \mid \mathbf{x}_{-j}^0)$ *under the oracle conditional distribution. Under the local smoothness and moment conditions in Appendix A,*

$$l^{\mathrm{RB}}(\theta; \mathbf{x}_{-j}^0) = g_\theta(\mu_j) + \frac{1}{2}\sigma_j^2 g_\theta''(\mu_j) + R_j, \quad (11)$$

*where* $R_j$ *is a higher-order local remainder. Moreover, writing the mean reconstruction function as* $\bar{\phi}_\theta(z) :=$ $\mathbb{E}_{t,\varepsilon}\left[D_\theta(\mathbf{x}^{A,t}(z), t)_{-j}\right]$, *and the error as* $e_\theta(z) := \bar{\phi}_\theta(z) - \mathbf{x}_{-j}^0$, *Eq.* (10) *admits the approximation:*

$$l^{\mathrm{RB}}(\theta; \mathbf{x}_{-j}^0) \approx \|e_\theta(\mu_j)\|_2^2 + \sigma_j^2 \|\bar{\phi}_\theta'(\mu_j)\|_2^2, \quad (12)$$

*where derivatives are with respect to the scalar value* $z$.

**Interpretation.** The second term in Eq. (12) is the leading nonnegative component of the local RB correction. It measures how strongly the denoiser's mean reconstruction of the observed block changes as the completed value $z$ varies. When the conditional variance $\sigma_j^2$ is large, the objective penalizes sensitivity to that uncertain completion, discouraging the denoiser from relying too strongly on it. When $\sigma_j^2$ is small, the completion is more reliable and can be used more directly as context.

This view clarifies the role of different completion rules. A constant placeholder has zero augmentation variance and may also have mismatched mean. It is therefore both biased and unregularized. A deterministic conditional mean can reduce mean mismatch, but it remains degenerate and therefore omits the variance-weighted sensitivity term. Stochastic conditional augmentation activates this term, so the denoiser's reliance on completed context adapts to conditional uncertainty. Thus, AugMask does not require the augmentation rule to be a perfect point imputer; its role is to provide plausible context together with uncertainty. The approximation is local: higher-order terms may matter for heavily skewed or heavy-tailed conditionals, as detailed in Appendix A.

**Extension to multiple missing coordinates.** Let $S = \{j : m_j = 0\}$ and $O = \{j : m_j = 1\}$ denote the missing and observed coordinate blocks for a sample. For a missing block $S$, define the observed-coordinate reconstruction loss induced by a block completion $\mathbf{z}_S$ as

$$g_\theta(\mathbf{z}_S) := \mathbb{E}_{t,\varepsilon}\left[\left\|D_\theta(\mathbf{x}^{A,t}(\mathbf{z}_S), t)_O - \mathbf{x}_O^0\right\|_2^2\right], \quad (13)$$

where $\mathbf{x}^{A,0}(\mathbf{z}_S)_O = \mathbf{x}_O^0$ and $\mathbf{x}^{A,0}(\mathbf{z}_S)_S = \mathbf{z}_S$. A general augmentation rule induces

$$l^T(\theta; \mathbf{x}_O^0, \mathbf{m}) = \mathbb{E}_{\mathbf{Z}_S \sim T(\cdot | \mathbf{x}_O^0, \mathbf{m})}\left[g_\theta(\mathbf{Z}_S)\right]. \quad (14)$$

The joint RB ideal corresponds to $T^\star(\cdot \mid \mathbf{x}_O^0, \mathbf{m}) = p_{\mathrm{data}}(\cdot \mid \mathbf{x}_O^0, \mathbf{m})$. A local multivariate Taylor expansion around the conditional mean $\boldsymbol{\mu} = \mathbb{E}[\mathbf{Z}_S \mid \mathbf{x}_O^0, \mathbf{m}]$ with covariance $\Sigma = \mathrm{Cov}(\mathbf{Z}_S \mid \mathbf{x}_O^0, \mathbf{m})$ gives

$$l^{\mathrm{RB}}(\theta; \mathbf{x}_O^0, \mathbf{m}) \approx g_\theta(\boldsymbol{\mu}) + \frac{1}{2}\mathrm{tr}\left(H_g(\boldsymbol{\mu})\Sigma\right). \quad (15)$$

Under a Gauss–Newton sensitivity approximation, $H_g(\boldsymbol{\mu}) \approx 2\Gamma$ with $\Gamma = J_{\bar{\phi}}(\boldsymbol{\mu})^\top J_{\bar{\phi}}(\boldsymbol{\mu})$, where $J_{\bar{\phi}}$ is the Jacobian of the mean reconstruction function with respect to the missing block. Then the leading term becomes

$$\mathrm{tr}(\Gamma\Sigma) = \underbrace{\sum_{j \in S} \Gamma_{jj}\Sigma_{jj}}_{P_{\mathrm{diag}}} + 2\underbrace{\sum_{\substack{j < k \\ j, k \in S}} \Gamma_{jk}\Sigma_{jk}}_{\Delta}. \quad (16)$$

The diagonal term $P_{\mathrm{diag}}$ is the additive variance-weighted sensitivity penalty captured by coordinate-wise stochastic augmentation, while $\Delta$ is the cross-coordinate interaction correction involving both conditional covariance and denoiser coupling.

An augmenter can model the full covariance $\Sigma$ and capture both $P_{\mathrm{diag}}$ and $\Delta$. Our practical augmenter uses the per-coordinate sampler in Eq. (17), which captures the diagonal uncertainty terms but ignores cross-coordinate interactions. A useful diagnostic for this gap is $\frac{|\Delta|}{P_{\mathrm{diag}}} \leq \max_{j \in S} \sum_{k \neq j} I_k A_{jk} |\rho_{jk|O}|$, where $I_k := \mathbf{1}\{k \in S\}$ is the indicator for the missing parts, $A_{jk} := |\Gamma_{jk}|/\sqrt{\Gamma_{jj}\Gamma_{kk}}$ measures denoiser coupling, and $\rho_{jk|O} := \Sigma_{jk}/\sqrt{\Sigma_{jj}\Sigma_{kk}}$ is the conditional correlation. Averaging $I_k$ introduces the co-missingness rate $q_{jk} := \Pr(I_k = 1 | I_j = 1, x_O)$, yielding a co-missingness-weighted quantity, $\sum_{k \neq j} q_{jk} A_{jk} |\rho_{jk|O}|$, that bounds $\frac{|\Delta|}{P_{\mathrm{diag}}}$ on average. Thus, the factorized approximation is most accurate when co-missing features are weakly correlated or coupled by the denoiser, and may be less accurate under block missingness, strong redundancy, or MNAR mechanisms.

### 3.4. Practical Implementation: Factorized Conditional Stochastic Augmentation

AugMask is defined for any missing-block augmentation rule $T(\cdot \mid \mathbf{x}^{\mathrm{obs}}, \mathbf{m})$. While $T$ may, in principle, be a joint conditional sampler, we use a lightweight factorized variant as default:

$$T_{\mathrm{F}}(\mathbf{z}_S \mid \mathbf{x}^{\mathrm{obs}}, \mathbf{m}) = \prod_{j \in S} T_j\left(z_j \mid \tilde{\mathbf{x}}_{-j}, \mathbf{m}_{-j}\right). \quad (17)$$

Each factor conditions on all available covariates and their missingness pattern, but samples missing coordinates independently given this context. Thus, the factorization is a practical implementation choice of the AugMask objective.

**Algorithm 1** Factorized Conditional Stochastic Augmentation via Per-Feature Models

1: **Input:** Incomplete dataset $\mathcal{D} = \{(\tilde{\mathbf{x}}_i, \mathbf{m}_i)\}_{i=1}^N$; smoothing $\alpha$; stability $\epsilon$; category cardinalities $\{K_j\}$.

2: **Output:** Augmented dataset $\mathcal{D}^A$ (constructed once and kept fixed during training).

3: **Models:** $\hat{f}_j^{\text{NUM}(\mu)}$ is a conditional mean regressor for $X_j$, $\hat{f}_j^{\text{NUM}(\sigma)}$ is a log-variance regressor, and $\hat{f}_j^{\text{CAT}}$ is a conditional classifier (probability predictor).

4: **Initialize** $\mathbf{x}_i^A \leftarrow \tilde{\mathbf{x}}_i$ for all $i$.

5: **for** $j = 1$ **to** $d$ **do**

6:     $\Omega_{\text{obs}} \leftarrow \{ i \mid m_{ij} = 1 \}$.

7:     Let $\mathbf{u}_i = (\tilde{\mathbf{x}}_{i,-j}, \mathbf{m}_{i,-j})$.

8:     $\mathcal{X}_{\text{train}} \leftarrow \{\mathbf{u}_i \mid i \in \Omega_{\text{obs}}\}$.

9:     **if** feature $j$ is continuous **then**

10:         Fit $\hat{f}_j^{\text{NUM}(\mu)}$ using $\mathcal{X}_{\text{train}}$ and targets $\{\tilde{x}_{ij}\}_{i \in \Omega_{\text{obs}}}$.

11:         $r_i \leftarrow \tilde{x}_{ij} - \hat{f}_j^{\text{NUM}(\mu)}(\mathbf{u}_i)$ for $i \in \Omega_{\text{obs}}$.

12:         $v_i \leftarrow \log(r_i^2 + \epsilon)$ for $i \in \Omega_{\text{obs}}$.

13:         Fit $\hat{f}_j^{\text{NUM}(\sigma)}$ using $\mathcal{X}_{\text{train}}$ and targets $\{v_i\}_{i \in \Omega_{\text{obs}}}$.

14:         **for** $i = 1$ **to** $N$ **do**

15:             **if** $m_{ij} = 0$ **then**

16:                 $\hat{\mu}_{ij} \leftarrow \hat{f}_j^{\text{NUM}(\mu)}(\mathbf{u}_i)$.

17:                 $\hat{\sigma}_{ij} \leftarrow \exp\left(\frac{1}{2}\hat{f}_j^{\text{NUM}(\sigma)}(\mathbf{u}_i)\right)$.

18:                 $x_{ij}^A \sim \mathcal{N}(\hat{\mu}_{ij}, \hat{\sigma}_{ij}^2)$.

19:             **end if**

20:         **end for**

21:     **else if** feature $j$ is categorical **then**

22:         Fit $\hat{f}_j^{\text{CAT}}$ using $\mathcal{X}_{\text{train}}$ and labels $\{\tilde{x}_{ij}\}_{i \in \Omega_{\text{obs}}}$.

23:         **for** $i = 1$ **to** $N$ **do**

24:             **if** $m_{ij} = 0$ **then**

25:                 $\hat{\mathbf{p}}_{ij} \leftarrow \hat{f}_j^{\text{CAT}}(\mathbf{u}_i)$.

26:                 $\hat{\mathbf{p}}_{ij} \leftarrow (1-\alpha)\hat{\mathbf{p}}_{ij} + \alpha \cdot \frac{1}{K_j}\mathbf{1}$.

27:                 $x_{ij}^A \sim \text{Categorical}(\hat{\mathbf{p}}_{ij})$.

28:             **end if**

29:         **end for**

30:     **end if**

31: **end for**

32: $\mathcal{D}^A \leftarrow \{(\mathbf{x}_i^A, \mathbf{m}_i)\}_{i=1}^N$.

33: **return** $\mathcal{D}^A$.

---

Motivated by Proposition 3.3, the practical goal is to estimate both conditional location and uncertainty rather than to optimize a standalone imputer. Algorithm 1 constructs a single augmented dataset by fitting per-feature models on observed cells without any hyperparameter tuning and sampling missing cells from the corresponding factors.

**Feature-wise factors.** For continuous features, $T_j$ is modeled as a heteroskedastic Gaussian: we fit a conditional mean regressor $\hat{\mu}_{ij} \approx \mathbb{E}[X_{ij} \mid \tilde{\mathbf{x}}_{i,-j}, \mathbf{m}_{i,-j}]$ and a second regressor on log-squared residuals to estimate $\hat{\sigma}_{ij}^2$ (Har-

---

*Table 1.* **Augmentation rules and computational cost.** The highlighted column is our default factorized stochastic implementation of $T$. The bottom row reports the mean wall-clock time for constructing the augmented dataset on Adult.

| | **Univariate Baselines** | | | | **Multivariate Baselines** | | | |
|---|---|---|---|---|---|---|---|---|
| Property | Zero | Mean | Noise | E-CDF | MICE | Hyper. | LGBM-D | **LGBM-S** |
| Stochastic? | ✗ | ✗ | ✓ | ✓ | ✗ | ✗ | ✗ | ✓ |
| Multivariate? | ✗ | ✗ | ✗ | ✗ | ✓ | ✓ | ✓ | ✓ |
| Model-based? | ✗ | ✗ | ✗ | ✗ | ✓ | ✓ | ✓ | ✓ |
| Efficiency | • | • | • | • | ○ | ○ | • | • |
| Time (s) | 0.041 | 0.084 | 0.107 | 0.176 | 26.22 | 109.4 | 6.04 | 9.87 |

• High efficiency    ○ Iterative/Lower efficiency

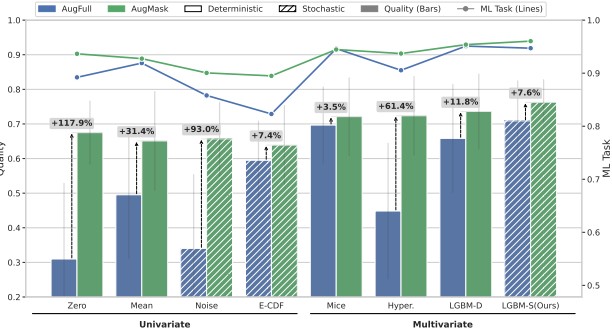

*Figure 2.* **Comparison of augmentation rules and masking.** Sample Quality (bars) and ML Task (lines) metrics across univariate and multivariate methods. **AugMask** (green) shows consistent gains over **AugFull** (blue). Deterministic modules are shown as solid; stochastic modules are shown with hatched patterns.

---

vey, 1976), then sample $Z_{ij} \sim \mathcal{N}(\hat{\mu}_{ij}, \hat{\sigma}_{ij}^2)$. For categorical features, $T_j$ is a conditional classifier estimating $\hat{\mathbf{p}}_{ij}$, from which we sample $Z_{ij} \sim \text{Categorical}(\hat{\mathbf{p}}_{ij})$ after label smoothing $\hat{\mathbf{p}}_{ij} \leftarrow (1-\alpha)\hat{\mathbf{p}}_{ij} + \alpha K_j^{-1}\mathbf{1}$ with $\alpha = 0.05$.

**Augmentation rule selection and scope.** We instantiate Eq. (17) with LightGBM (Ke et al., 2017) using default hyperparameters for all datasets and features. Gradient-boosted trees are efficient for mixed-type tabular data and can directly incorporate the mask $\mathbf{m}_{i,-j}$ as context (Grinsztajn et al., 2022). Unless stated otherwise, we use **LGBM-S**; **LGBM-D** denotes the deterministic variant that uses Light-GBM point predictions without sampling. Table 1 summarizes alternatives. See Appendix C.5 for more details.

Figure 2 supports two design choices. First, AugMask improves over AugFull when both use the same augmented inputs, suggesting that completions should serve as context rather than targets. Second, stochastic conditional augmentation provides better context than its deterministic variant. The factorized LightGBM stage is run once offline and can be parallelized across features, making it practical for the moderate-dimensional benchmarks studied here. For ultra-high-dimensional data, strong block missingness, or dependence among co-missing features, AugMask can instead be paired with a joint conditional sampler for $T$.

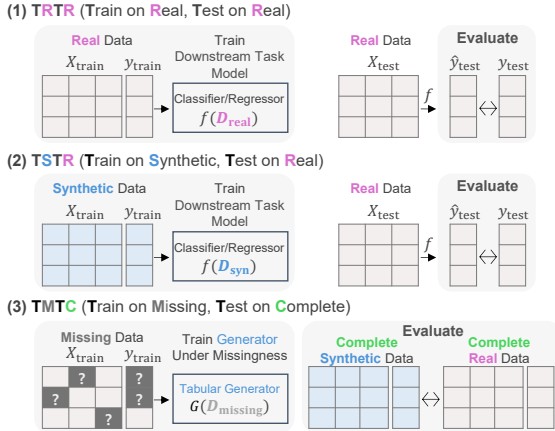

**(1) TRTR (Train on Real, Test on Real)**

**(2) TSTR (Train on Synthetic, Test on Real)**

**(3) TMTC (Train on Missing, Test on Complete)**

*Figure 3.* **Evaluation protocols for missing-aware generators**.
(1) **TRTR** (**T**rain on **R**eal, **T**est on **R**eal) trains and evaluates
downstream models on complete real data, providing an upper
bound. (2) **TSTR** (**T**rain on **S**ynthetic, **T**est on **R**eal) trains down-
stream task models on *synthetic data* and evaluates on real test
data (Esteban et al., 2017). (3) **TMTC** (**T**rain on **M**issing, **T**est on
**C**omplete) trains generator on incomplete data and its synthetic
samples are evaluated against the complete real data.

*Table 2.* Baseline models categorized by their missing-awareness.

| Category | Model | Reference |
|---|---|---|
| *Missing-aware* | MIWAE | (Mattei & Frellsen, 2019) |
| | MissDiff | (Ouyang et al., 2023) |
| | ForestDiff | (Jolicoeur-Martineau et al., 2024) |
| | DiffPuter | (Zhang et al., 2025) |
| *Missing-unaware* | TVAE / CTGAN | (Xu et al., 2019) |
| | ARF | (Watson et al., 2023) |
| | TabSyn | (Zhang et al., 2023) |
| | TabDDPM | (Kotelnikov et al., 2023) |
| | TabDiff | (Shi et al., 2025) |
| | CDTD | (Mueller et al., 2025) |

**Constructing the augmented dataset.** For each training
sample, we draw the missing block once, $\mathbf{Z}_{i,S_i} \sim T(\cdot \mid \mathbf{x}_i^{\mathrm{obs}}, \mathbf{m}_i)$, construct $\mathbf{x}_i^A$, and keep $\mathcal{D}^A = \{(\mathbf{x}_i^A, \mathbf{m}_i)\}_{i=1}^N$
fixed during diffusion training. This gives stochastic but sta-
tionary conditioning: uncertainty enters through the initial
draw, while inputs do not drift across epochs. Appendix C.5
discusses cycling cached augmentations.

## 4. Experiments

We assess whether our training strategy enables *missing-
unaware* models to generate faithful complete synthetic
tabular data when they are trained from incomplete data.

### 4.1. Experimental Setup

**Baselines.** We assess our training strategy on *missing-
unaware* models against *missing-aware* baselines (Table 2).

**Training strategies.** We consider three strategies (Fig. 1):
(1) Baseline (**NoAug**) uses feature-wise mean/mode filling
with no loss masking, serving as control. (2) **AugFull** ap-
plies stochastic augmentation (Alg. 1) without loss-masking
(applied to all missing-unaware baselines), measuring the
gain from augmentation. (3) **AugMask** further applies loss-
masking to observed parts (applied to diffusion baselines
only), measuring the gain from loss-masking.

**Datasets and Missingness.** We follow the prior
dataset curation of Zhang et al. (2023); see Ap-
pendix C.1. We simulate MCAR/MAR missingness at ra-
tios $\{0.1, 0.3, 0.5, 0.7, 0.9\}$ (C.3). For preprocessing, we
fit the transformer/scaler on observed entries (C.2) before
constructing the augmented dataset (sampled once for sta-
bility; C.5). We split train/val/test as 60/20/20.

**Implementation details.** We provide details in the Ap-
pendix: baseline implementations (C.6), loss masking for
diffusion models (C.7), and hardware specifications (C.8).

### 4.2. Evaluation Protocols Under Missingness

We evaluate **generative quality**, which measures how faith-
ful the synthetic data are, and **machine learning utility**,
which measures how useful they are for downstream tasks.

**Evaluation metrics.** We evaluate using the protocols de-
scribed in Figure 3, focusing on two main measures.

1. **Generative Quality:** It measures how accurately the
   generator recovers the data distribution from incom-
   plete observations. Following Zhang et al. (2023),
   we report *sample-level* fidelity criterion with (i) $\alpha$-
   **Precision** and (ii) $\beta$-**Recall** (Alaa et al., 2022), and
   (iii) *feature-level* pairwise correlations metric (**Trend**)
   and (iv) *column-level* density proximity (**Shape**).

2. **Machine Learning Utility:** We evaluate downstream
   task performance using AUROC (classification) or
   RMSE (regression) under **TSTR** protocol and normal-
   ize the values using the **TRTR** performance.

## 5. Analysis

### 5.1. Quantitative Comparison to State of the Art

Table 3 shows two effects of AugMask. First, condi-
tional stochastic augmentation alone already provides use-
ful missing-aware context: **+AugFull** improves missing-
unaware baselines across architectures, including latent dif-
fusion (TabSyn: Trend** +8.0; Shape* +9.8). Second, for
feature-space diffusion backbones, masking the loss further
improves performance by preventing completed entries from
becoming supervised pseudo-targets. With **+AugMask**,
score-based diffusion reaches the best overall ranks among

*Table 3.* **MCAR Results (90 Scenarios): Comparative Rank Analysis.** Cells display Average Rank (↓) ± Standard Deviation across 90 scenarios: 6 datasets × 5 missing ratios [0.1–0.9] × 3 seeds. Ranks are computed per experimental block; **Summary Rank** is the mean rank across all metrics within each block. Δ denotes absolute rank gain over the +NoAug baseline. ** and * indicate statistical **Gain** and **Match** via Wilcoxon test ($p < 0.05$) against the *best-performing missing-aware baseline* per block among: MIWAE, MissDiff, DiffPuter, or ForestDiff. Boldface denotes results statistically superior to this entire baseline group. See Appendix D for MAR and detailed results.

| Method | Generative Quality | | | | | | | | Utility | | Summary | |
| | α-Precision | | β-Recall | | Trend | | Shape | | ML Task | | | |
| | Rank | Δ | Rank | Δ | Rank | Δ | Rank | Δ | Rank | Δ | **Rank** | Δ |
| --- | --- | --- | --- | --- | --- | --- | --- | --- | --- | --- | --- | --- |
| **(A) Missing-aware baselines** | | | | | | | | | | | | |
| MIWAE | $6.9 \pm 3.8$ | – | $10.0 \pm 4.6$ | – | $8.9 \pm 4.6$ | – | $8.4 \pm 4.6$ | – | $10.4 \pm 4.9$ | – | 8.9 | – |
| MissDiff | $16.1 \pm 4.6$ | – | $17.3 \pm 4.4$ | – | $13.3 \pm 4.0$ | – | $15.7 \pm 2.9$ | – | $16.8 \pm 2.7$ | – | 15.8 | – |
| DiffPuter | $12.7 \pm 4.5$ | – | $13.2 \pm 4.9$ | – | $12.8 \pm 3.9$ | – | $12.8 \pm 3.7$ | – | $17.2 \pm 4.2$ | – | 13.7 | – |
| ForestDiff | $10.7 \pm 4.1$ | – | $5.4 \pm 4.9$ | – | $8.4 \pm 4.3$ | – | $9.5 \pm 4.0$ | – | $6.4 \pm 3.7$ | – | 8.1 | – |
| **(B) Missing-unaware baselines** | | | | | | | | | | | | |
| TVAE | $18.1 \pm 3.3$ | – | $18.1 \pm 2.6$ | – | $17.7 \pm 1.9$ | – | $18.7 \pm 2.2$ | – | $15.7 \pm 3.0$ | – | 17.7 | – |
| └ +AugFull | $11.7 \pm 5.2$ | (+6.4) | $11.8 \pm 4.8$ | (+6.3) | $10.6 \pm 2.8$ | (+7.2) | $11.0 \pm 1.8$ | (+7.7) | $10.9 \pm 3.4$ | (+4.9) | 11.2 | (+6.5) |
| CTGAN | $12.4 \pm 3.9$ | – | $16.7 \pm 3.0$ | – | $15.9 \pm 3.1$ | – | $16.5 \pm 2.6$ | – | $17.6 \pm 2.6$ | – | 15.8 | – |
| └ +AugFull | $12.1 \pm 4.7$ | (+0.4) | $13.6 \pm 5.0$ | (+3.1) | $12.4 \pm 3.2$ | (+3.5) | $11.1 \pm 3.6$ | (+5.4) | $15.0 \pm 3.8$ | (+2.7) | 12.8 | (+3.0) |
| ARF | $17.1 \pm 2.7$ | – | $18.4 \pm 1.9$ | – | $20.4 \pm 0.8$ | – | $20.0 \pm 0.9$ | – | $16.7 \pm 3.3$ | – | 18.5 | – |
| └ +AugFull | $8.4 \pm 5.3$ | (+8.7) | $12.1 \pm 4.8$ | (+6.3) | $18.4 \pm 2.9$ | (+2.0) | $12.1 \pm 5.7$ | (+7.9) | $11.0 \pm 5.2$ | (+5.7) | 12.4 | (+6.1) |
| TabSyn | $13.6 \pm 3.6$ | – | $13.7 \pm 2.0$ | – | $13.3 \pm 2.8$ | – | $14.9 \pm 3.4$ | – | $13.2 \pm 3.5$ | – | 13.7 | – |
| └ +AugFull | $6.8 \pm 2.4$ | (+6.8) | $7.6 \pm 1.9$ | (+6.1) | $\mathbf{5.3 \pm 1.9}$** | (+8.0) | $5.0 \pm 2.6$* | (+9.8) | $6.5 \pm 3.3$ | (+6.7) | 6.3* | (+7.5) |
| TabDDPM | $15.5 \pm 3.2$ | – | $12.0 \pm 2.3$ | – | $15.0 \pm 2.2$ | – | $13.5 \pm 1.9$ | – | $11.4 \pm 2.9$ | – | 13.5 | – |
| └ +AugFull | $7.1 \pm 5.6$* | (+8.3) | $7.5 \pm 3.1$ | (+4.5) | $8.8 \pm 5.1$ | (+6.2) | $7.5 \pm 5.0$* | (+6.0) | $6.5 \pm 4.0$ | (+4.9) | 7.5 | (+6.0) |
| └ **+AugMask** | $8.0 \pm 5.4$ | **(+7.4)** | $\mathbf{5.3 \pm 4.2}$* | **(+6.7)** | $8.6 \pm 5.1$ | **(+6.4)** | $7.6 \pm 4.9$* | **(+5.9)** | $5.7 \pm 5.3$* | **(+5.7)** | 7.0* | **(+6.4)** |
| CDTD | $13.1 \pm 3.3$ | – | $14.4 \pm 3.3$ | – | $13.2 \pm 2.8$ | – | $15.5 \pm 2.4$ | – | $12.8 \pm 3.4$ | – | 13.8 | – |
| └ +AugFull | $5.6 \pm 2.6$* | (+7.5) | $6.4 \pm 2.6$ | (+8.1) | $\mathbf{4.9 \pm 2.3}$** | (+8.3) | $5.3 \pm 1.7$* | (+10.3) | $6.8 \pm 2.9$ | (+6.0) | 5.8* | (+8.0) |
| └ **+AugMask** | $\mathbf{4.0 \pm 3.3}$** | **(+9.1)** | $4.2 \pm 3.2$* | **(+10.3)** | $\mathbf{3.0 \pm 2.2}$** | **(+10.2)** | $\mathbf{3.5 \pm 2.4}$** | **(+12.1)** | $4.4 \pm 2.7$* | **(+8.4)** | 3.8** | **(+10.0)** |
| TabDiff | $16.4 \pm 3.1$ | – | $11.0 \pm 3.1$ | – | $12.1 \pm 2.8$ | – | $13.1 \pm 3.2$ | – | $11.1 \pm 3.8$ | – | 12.7 | – |
| └ +AugFull | $6.1 \pm 4.0$* | (+10.3) | $6.1 \pm 3.7$ | (+4.8) | $\mathbf{3.8 \pm 2.3}$** | (+8.3) | $\mathbf{4.0 \pm 2.3}$** | (+9.1) | $5.9 \pm 2.9$ | (+5.3) | 5.2** | (+7.6) |
| └ **+AugMask** | $\mathbf{4.7 \pm 4.9}$** | **(+11.7)** | $\mathbf{3.3 \pm 4.4}$** | **(+7.7)** | $\mathbf{2.3 \pm 2.2}$** | **(+9.8)** | $\mathbf{3.0 \pm 3.0}$** | **(+10.1)** | $\mathbf{3.0 \pm 2.4}$** | **(+8.1)** | 3.2** | **(+9.5)** |

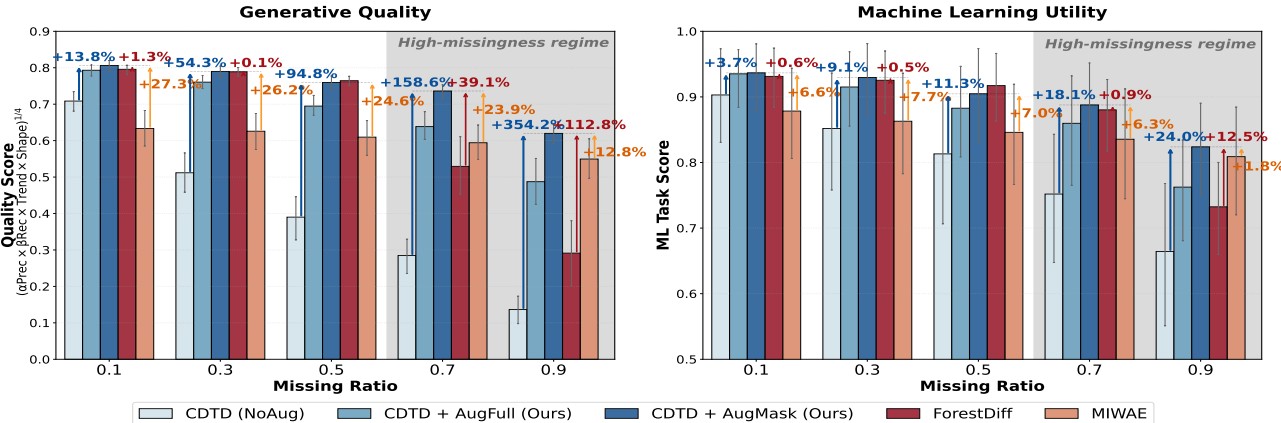

*Figure 4.* **Robustness across missing ratios (MCAR).** Our *stochastic augmentation* and *masking* strategy for the missing-unaware **CDTD** baseline shows a hierarchical gain (*NoAug → AugFull → AugMask*) that matches or surpasses top missing-aware baselines (**ForestDiff**, **MIWAE**) in **Sample Quality** (Left) and **Utility** (Right). See Appendix D for MAR and further comparison.

compared methods (TabDiff: 3.2**; CDTD: 3.8**), while also improving downstream utility (ML Task +8.1/+8.4). Thus, the gain comes not only from constructing complete inputs, but also from how those completions are used during training: as conditioning context rather than targets.

## 5.2. Robustness Across Missing Ratios

Figure 4 shows that the benefit of AugMask becomes more pronounced as missingness increases. The trajectory follows a clear hierarchy: **NoAug** degrades sharply under high

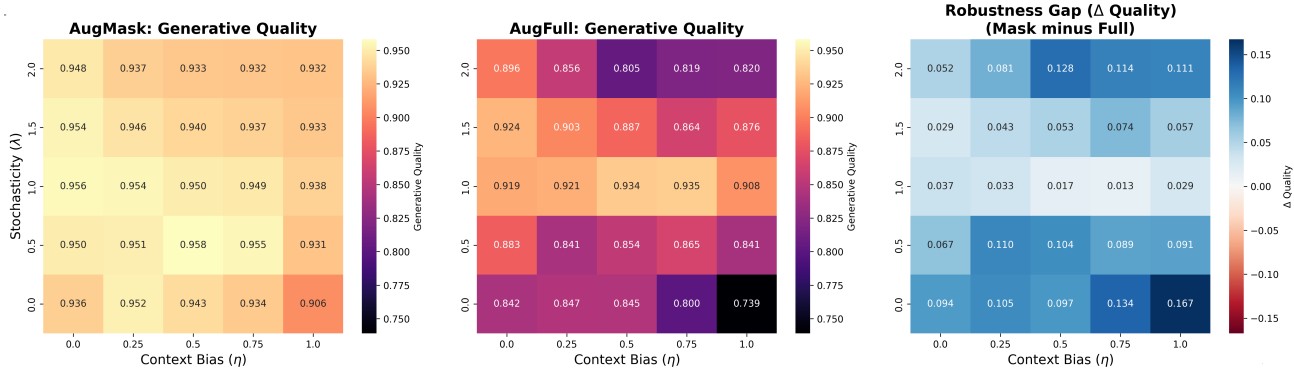

Figure 5. **Sensitivity to augmentation misspecification.** We perturb one fitted LGBM-S augmenter by varying context bias $\eta$ and stochasticity $\lambda$. Here, $\eta = 0$ preserves the conditional LGBM estimate and $\eta = 1$ reverts to a marginal baseline; $\lambda = 1$ keeps the estimated uncertainty, while $\lambda = 0$ gives deterministic filling. The right panel reports $\Delta$ Quality = AugMask $-$ AugFull under matched augmented inputs. The gap remains positive across all settings, showing that AugMask is less vulnerable to misspecified completions because it uses them as context rather than supervised targets.

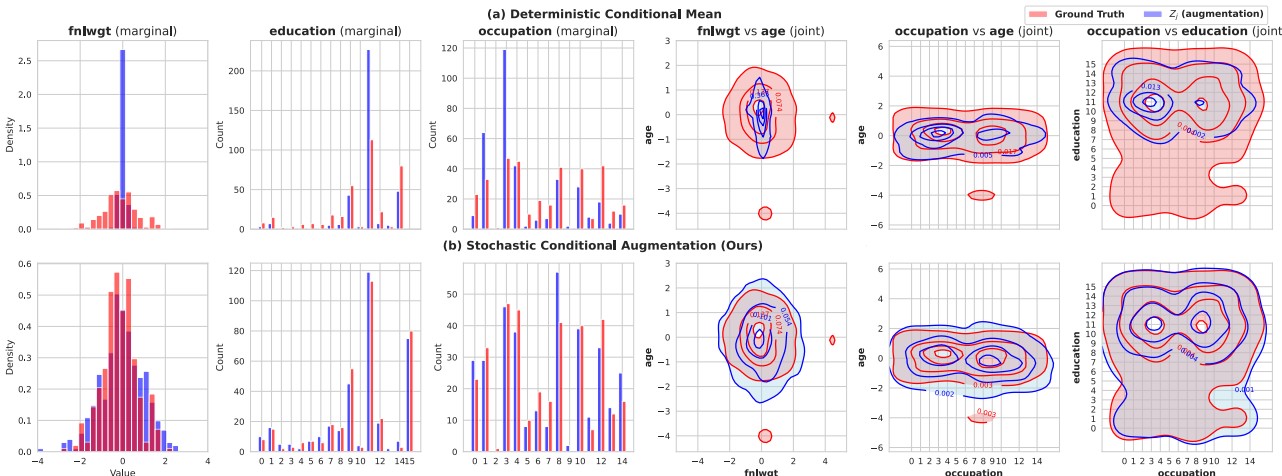

Figure 6. **Visualizing augmentations (LGBM-D vs. LGBM-S).** We overlay *marginal* densities (columns 1-3) and *joint* kernel densities (columns 4-6) of the ground truth (unknown) and augmented data. **(a) Deterministic conditional mean (LGBM-D):** Significant *shrinkage* occurs from ignoring conditional variance (e.g., fnlwgt), resulting in artificial spikes and restricted support in the joint plots. **(b) Stochastic conditional augmentation (Ours, LGBM-S):** It captures marginal and conditional variability. While (a) matches first moments, preserving second moment provides better context about the data distribution. Categorical values are integer-encoded for clarity.

missingness, **AugFull** recovers part of the lost distributional structure through stochastic context, and **AugMask** further stabilizes both sample quality and downstream utility. At the extreme $p_{\text{miss}} = 0.9$ regime, AugMask avoids the severe degradation observed in other missing-aware baselines. Under high uncertainty, completions are useful as context, but treating them as ground-truth targets can amplify errors.

### 5.3. Sensitivity to Augmentation Misspecification

A natural question is whether AugMask depends mainly on having a strong augmenter. We test this by starting from one fitted LGBM-S augmenter and perturbing only its context bias $\eta$ and stochasticity $\lambda$, while using the same augmented inputs for AugMask and AugFull.

For each missing continuous feature, we sample $Z_{ij} \sim \mathcal{N}\left(\tilde{\mu}_{ij}(\eta), \tilde{\sigma}_{ij}^2(\lambda)\right), \tilde{\mu}_{ij}(\eta) = (1-\eta)\hat{\mu}_{ij} + \eta\bar{\mu}_j, \tilde{\sigma}_{ij}(\lambda) = \lambda\hat{\sigma}_{ij}$, where $\left(\hat{\mu}_{ij}, \hat{\sigma}_{ij}\right)$ are the LGBM-S conditional estimates and $\bar{\mu}_j$ is the marginal mean. For categorical features, define $\mathbf{q}_{ij}(\eta) = (1-\eta)\hat{\mathbf{p}}_{ij} + \eta\bar{\boldsymbol{\pi}}_j, \tilde{\mathbf{p}}_{ij}(\eta, \lambda) = \frac{\mathbf{q}_{ij}(\eta)^{1/\lambda}}{\sum_k q_{ijk}(\eta)^{1/\lambda}}(\lambda > 0)$, where $\hat{\mathbf{p}}_{ij}$ is the conditional class-probability estimate and $\bar{\boldsymbol{\pi}}_j$ is the marginal class frequency. For $\lambda = 0$, we use the deterministic value, i.e., a point mass at $\arg\max_k q_{ijk}(\eta)$. Thus, $\eta = 0$ preserves conditional context, while $\eta = 1$ reverts to marginal context; $\lambda = 1$ keeps the estimated stochasticity, $\lambda < 1$ makes samples under-dispersed, $\lambda > 1$ makes them over-dispersed and $\lambda = 0$ gives a deterministic variant.

Figure 5 shows that the robustness gap is positive across the

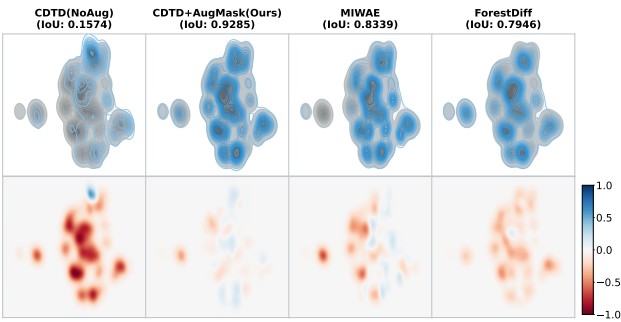

*Figure 7.* **UMAP comparison of real and synthetic data.** Generators are trained on incomplete Adult dataset ($p_{\text{miss}} = 0.7$). **Top row**: Grey kernel density in the background indicates the complete real data. Blue contours represent synthetic density, with overlap measured by IoU. **Bottom row**: Kernel density estimate residuals ($\text{KDE}_{\text{Syn}} - \text{KDE}_{\text{Real}}$). Blue indicates synthetic over-representation; red indicates mode collapse relative to ground truth.

entire $5 \times 5$ sweep. The endpoints recover original rules: $(\eta, \lambda) = (0, 1)$ is **LGBM-S**, $(0, 0)$ is closest to **LGBM-D**, and $(1, 0)$ collapses to marginal mean/mode filling. As $\eta$ or $\lambda$ deviates from the default setting, AugFull becomes more vulnerable because it is trained to reconstruct misspecified inputs.

### 5.4. Qualitative Analysis

Figure 6 visualizes why deterministic conditional completion can be insufficient. Although **LGBM-D** matches conditional means, it collapses conditional variability, producing spiky marginals and restricted joint support. In contrast, **LGBM-S** restores realistic marginal and joint variation by sampling from an estimated conditional distribution. This is consistent with the variance-weighted sensitivity view: stochastic completions expose the denoiser to plausible alternatives rather than a single overconfident point estimate.

Figure 7 shows the effect on generated samples under heavy missingness. Without augmentation, the baseline diffusion model (CDTD) suffers severe support mismatch and mode collapse (IoU = 0.157). With AugMask, the generated density overlaps the real-data density much more closely (IoU = 0.929), capturing the modes better. Together with the misspecification study, these results indicate that AugMask improves the distributional coverage.

### 6. Related Work

**Generative models for complete/incomplete tabular data.** Deep generative models for complete tabular data include GAN-based (Xu et al., 2019; Zhao et al., 2021), VAE-based (Ma et al., 2020), and diffusion/score-based approaches (Kotelnikov et al., 2023; Kim et al., 2022; Lee et al., 2023; Mueller et al., 2025; Shi et al., 2025). In contrast, methods for incomplete data have primarily focused

on imputation (Yoon et al., 2018; Mattei & Frellsen, 2019; Ipsen et al., 2020; Zheng & Charoenphakdee, 2022; Zhang et al., 2025). Fewer methods directly address generation from incomplete data: MissDiff (Ouyang et al., 2023) utilizes an observed-only loss but neglects input conditioning, while a recent tree-based model (Jolicoeur-Martineau et al., 2024) addresses both imputation and generation.

**Marginal score matching.** Givens et al. (2025) formulate score matching under missingness by integrating out missing entries so that the objective depends only on observed coordinates, and estimate this objective via importance sampling. AugMask pursues a similar marginalization principle in a practical training pipeline, where stochastic conditional augmentation samples plausible missing values as input context, while the loss is applied only to originally observed coordinates. This yields an objective regularized by variance-induced sensitivity that reduces reliance on uncertain missing entries without requiring importance sampling.

**Augmentation as implicit regularization.** Stochastic perturbations and data augmentation are often viewed as implicit regularization, where injected randomness induces Jacobian-type penalties and improves robustness (Bishop, 1995; Wager et al., 2013; Rifai et al., 2011; Dao et al., 2019). Here, stochasticity instead addresses *missingness*. Values are unobserved, yet diffusion backbones require fully specified inputs. We sample context-conditioned augmentations, using their uncertainty to limit dependence on missing coordinates. This links marginalizing out missing values to sensitivity regularization; both make missing entries effectively ignorable, with uncertainty setting the strength.

### 7. Conclusion

We propose **AugMask**, a plug-and-play training strategy that enables missing-unaware diffusion models to train on incomplete tabular data. By using conditional stochastic augmentation while excluding completed entries from supervision, AugMask achieves the best performance among the compared methods.

**Scope and limitations.** Our strategy targets score-based models that operate directly in feature space, where observed-only supervision is natural. Extending AugMask to latent diffusion pipelines (Zhang et al., 2023) is non-trivial, as missingness-induced uncertainty must be transmitted through the encoder/decoder and the training objective redefined in latent space. Handling missingness in such high-dimensional settings is an important direction. Finally, we focus on generation; extensions to imputation-specific objectives and privacy-related settings are left for future work.

## Acknowledgements

This work was supported by the National Research Foundation of Korea (NRF) grant funded by the Korea government (MSIT) (RS-2023-00217705, RS-2024-00341749, RS-2026-25487501), the MSIT (Ministry of Science and ICT), Korea, under the ICAN (ICT Challenge and Advanced Network of HRD) support program (RS-2023-00259934), Developing the Next-Generation General AI with Reliability, Ethics, and Adaptability (RS-2025-02283048), and the Leading Generative AI Human Resources Development (RS-2026-25544647) supervised by the IITP (Institute for Information & Communications Technology Planning & Evaluation), and the Ministry of Trade, Industry and Resources (MOTIR), Korea, under the project "Industrial Technology Infrastructure Program" (RS-2024-00466693).

## Impact Statement

This paper studies training score-based diffusion models for incomplete tabular data. Improved generative modeling under missingness can help researchers and practitioners construct synthetic benchmarks, study robustness to missing data, and reduce the barrier to using tabular datasets in settings where missing values are common. At the same time, AugMask does not provide formal privacy guarantees; generated samples should be audited for memorization and sensitive attribute leakage before release. Missingness patterns can also encode social, clinical, institutional, or measurement biases. We suggest domain-specific validation, fairness and privacy audits, and transparent reporting of the data source, missingness, and evaluation protocol before using generated data in high-stakes applications.

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

## Appendix Table of Contents

## A. Proofs

**Section structure.**  The structure of this section is as follows: (i) We establish the Rao-Blackwellization for augmentation-induced objectives as the variance reducing objective, and (ii) prove the second-order local approximation of the RB ideal around the conditional mean (Proposition 3.3).

**Lemma A.1** (Rao-Blackwellization for augmentation-induced objectives).  *Let $(\mathbf{X}^0, \mathbf{M}) \sim p_{\text{data}}$ be a random sample and mask, and let $\mathbf{X}_{\text{obs}}^0$ denote the observed part. Let $T(\cdot \mid \mathbf{x}_{\text{obs}}, \mathbf{m})$ be any augmentation rule for the missing part. Draw a context $(\mathbf{X}_{\text{obs}}^0, \mathbf{M})$, sample an augmentation $Z \sim T(\cdot \mid \mathbf{X}_{\text{obs}}^0, \mathbf{M})$, and evaluate the (random) loss*

$$U_T := l(\theta; \mathbf{X}_{\text{obs}}^0, \mathbf{M}, Z).$$

*Define the conditional expectation of this loss given the observed context as*

$$\bar{U}_T := \mathbb{E}\big[U_T \mid \mathbf{X}_{\text{obs}}^0, \mathbf{M}\big]$$

*We view $U_T$ as single-sample (random) estimate of the population objective $L^T(\theta) = \mathbb{E}[U_T]$. Then the following properties hold:*

*(a)  **Population identity.***
$$L^T(\theta) := \mathbb{E}[U_T] = \mathbb{E}[\bar{U}_T] = \mathbb{E}_{\mathbf{X}_{\text{obs}}^0, \mathbf{M}}\big[l^T(\theta; \mathbf{X}_{\text{obs}}^0, \mathbf{M})\big].$$

*(b)  **Variance reduction.***
$$\text{Var}(U_T) = \mathbb{E}\big[\text{Var}(U_T \mid \mathbf{X}_{\text{obs}}^0, \mathbf{M})\big] + \text{Var}(\bar{U}_T) \quad \implies \quad \text{Var}(\bar{U}_T) \leq \text{Var}(U_T).$$

*The term $\mathbb{E}[\text{Var}(U_T \mid \mathbf{X}_{\text{obs}}^0, \mathbf{M})]$ is the avoidable variance induced by augmentation randomness beyond what is determined by the observed context.*

*(c)  **Minimal MSE (best predictor using the incomplete data).**  For any estimator $g(\mathbf{X}_{\text{obs}}^0, \mathbf{M})$ that depends solely on the observed context,*
$$\mathbb{E}\big[(U_T - \bar{U}_T)^2\big] \leq \mathbb{E}\big[(U_T - g(\mathbf{X}_{\text{obs}}^0, \mathbf{M}))^2\big],$$
*with equality if and only if $g(\mathbf{X}_{\text{obs}}^0, \mathbf{M}) = \bar{U}_T$.*

*Proof of Lemma A.1.* **(a)** By the law of iterated expectations,

$$\mathbb{E}[\bar{U}_T] = \mathbb{E}\big[\mathbb{E}[U_T \mid \mathbf{X}^0_{\text{obs}}, \mathbf{M}]\big] = \mathbb{E}[U_T].$$

By definition, $\mathbb{E}[U_T]$ matches the population loss $L^T(\theta)$.

**(b)** We apply the law of total variance conditioning on the pair $(\mathbf{X}^0_{\text{obs}}, \mathbf{M})$:

$$\text{Var}(U_T) = \mathbb{E}\big[\text{Var}(U_T \mid \mathbf{X}^0_{\text{obs}}, \mathbf{M})\big] + \text{Var}\big(\mathbb{E}[U_T \mid \mathbf{X}^0_{\text{obs}}, \mathbf{M}]\big) = \mathbb{E}\big[\text{Var}(U_T \mid \mathbf{X}^0_{\text{obs}}, \mathbf{M})\big] + \text{Var}(\bar{U}_T).$$

Since variances are non-negative, $\text{Var}(\bar{U}_T) \leq \text{Var}(U_T)$.

**(c)** Let $g(\mathbf{X}^0_{\text{obs}}, \mathbf{M})$ be any estimator depending only on the observed context. Decompose the error $U_T - g$ as:

$$U_T - g = (U_T - \bar{U}_T) + (\bar{U}_T - g).$$

Squaring and taking expectations yields:

$$\mathbb{E}[(U_T - g)^2] = \mathbb{E}[(U_T - \bar{U}_T)^2] + \mathbb{E}[(\bar{U}_T - g)^2] + 2\mathbb{E}\big[(U_T - \bar{U}_T)(\bar{U}_T - g)\big].$$

We analyze the cross-term by conditioning on the context $(\mathbf{X}^0_{\text{obs}}, \mathbf{M})$. Since the term $(\bar{U}_T - g)$ depends only on $(\mathbf{X}^0_{\text{obs}}, \mathbf{M})$, it is fixed given the context and can be moved out of the conditional expectation:

$$\mathbb{E}\big[(U_T - \bar{U}_T)(\bar{U}_T - g) \mid \mathbf{X}^0_{\text{obs}}, \mathbf{M}\big] = (\bar{U}_T - g) \cdot \mathbb{E}\big[U_T - \bar{U}_T \mid \mathbf{X}^0_{\text{obs}}, \mathbf{M}\big].$$

By definition, $\mathbb{E}[U_T \mid \mathbf{X}^0_{\text{obs}}, \mathbf{M}] = \bar{U}_T$, so the inner expectation is zero. The relation simplifies to:

$$\mathbb{E}[(U_T - g)^2] = \mathbb{E}[(U_T - \bar{U}_T)^2] + \mathbb{E}[(\bar{U}_T - g)^2] \geq \mathbb{E}[(U_T - \bar{U}_T)^2].$$

Equality holds if and only if $\mathbb{E}[(\bar{U}_T - g)^2] = 0$, implying $g = \bar{U}_T$. $\qquad\square$

*Remark* A.2 (Fixed-context view vs. population view). In the main text, we fix an observed block $\mathbf{x}^0_{\text{obs}}$ (defined by a mask $\mathbf{m}$) and study randomness arising strictly from the augmentation $Z$ (and the diffusion process). In this fixed-context view, the quantity $l^T(\theta; \mathbf{x}^0_{\text{obs}}, \mathbf{m}) = \mathbb{E}_{Z \sim T(\cdot|\mathbf{x}^0_{\text{obs}}, \mathbf{m})}\big[l(\theta; \mathbf{x}^0_{\text{obs}}, Z)\big]$ is deterministic and serves as the objective function for that specific input. For completeness, Lemma A.1 establishes the corresponding properties over the population where both the data $\mathbf{X}^0$ and the mask $\mathbf{M}$ are random. This formalizes the variance-reduction interpretation of Rao-Blackwellization: marginalizing out the $Z$ yields a lower-variance estimator of the population loss.

**Proof roadmap for Proposition 3.3**   (Step 1) Taylor expand $g_\theta(Z_j)$ around $\mu_j$ with a third-order remainder. (Step 2) Take conditional expectation over $Z_j \mid \mathbf{x}^0_{-j}$ to obtain $g_\theta(\mu_j) + \frac{1}{2}\sigma_j^2 g_\theta''(\mu_j)$ plus a remainder term. (Step 3) Bound the remainder using Assumption A.3. (Steps 4–6, interpretation) Rewrite $g_\theta$ via a bias-variance decomposition over diffusion randomness, differentiate to expose a sensitivity penalty, and plug into the Taylor approximation.

**Assumption A.3** (Local smoothness and moments). (i) (*Local smoothness*) $g$ is three times continuously differentiable on a neighborhood of $\mu_j(\mathbf{x}^0_{-j}) := \mathbb{E}[X_j^0 \mid X_{-j}^0 = \mathbf{x}^0_{-j}]$, and $\sup_{|z-\mu_j|\leq r} |g^{(3)}(z)| < \infty$ for some $r > 0$.
(ii) (*Finite third moment*) $\mathbb{E}[|X_j^0 - \mu_j|^3 \mid X_{-j}^0 = \mathbf{x}^0_{-j}] < \infty$.

**Proposition A.4** (Local approximation of the RB ideal). *Recall the RB ideal in Eq.* (10). *Let* $Z_j \sim p_{\mathrm{data}}(\cdot \mid \mathbf{x}^0_{-j}, \mathbf{m})$, $\mu_j := \mathbb{E}[Z_j \mid \mathbf{x}^0_{-j}, \mathbf{m}]$, $\sigma_j^2 := \mathrm{Var}(Z_j \mid \mathbf{x}^0_{-j}, \mathbf{m})$, *and let* $W$ *collect the randomness in Eq.* (8) *(i.e., the sampled diffusion time and forward noise). Define the scalar loss function* $g_\theta(z) := l_{-j}(\theta; \mathbf{x}^0_{-j}, z)$. *Under standard smoothness conditions (Assumption A.3 in Appendix A),*

$$l^{\mathrm{RB}}(\theta; \mathbf{x}^0_{-j}) \approx g_\theta(\mu_j) + \frac{1}{2}\sigma_j^2 g_\theta''(\mu_j). \tag{18}$$

*Moreover, define* $\bar\phi(u) := \mathbb{E}_W[\phi_\theta(u; W)]$, $s(u) := \mathrm{tr}\left(\mathrm{Var}_W(\phi_\theta(u; W))\right)$, *and* $e(u) := \bar\phi(u) - \mathbf{x}^0_{-j}$, *where* $\mathbf{x}^0_{-j}$ *and* $\mathbf{m}$ *are fixed throughout. Then* $g_\theta(u) = \|e(u)\|_2^2 + s(u)$ *and*

$$l^{\mathrm{RB}}(\theta; \mathbf{x}^0_{-j}) \approx \|e(\mu_j)\|_2^2 + \sigma_j^2 \|\bar\phi'(\mu_j)\|_2^2 + \sigma_j^2 e(\mu_j)^\top \bar\phi''(\mu_j)$$
$$+ s(\mu_j) + \frac{1}{2}\sigma_j^2 s''(\mu_j), \tag{19}$$

*where derivatives are with respect to the scalar fill value* $u$ *(holding* $\mathbf{x}^0_{-j}$ *fixed).*

*Proof.* **Step 1: Local Taylor expansion.** Fix $\mathbf{x}^0_{-j}$ and let $Z_j \sim p_{\mathrm{data}}(\cdot \mid \mathbf{x}^0_{-j})$. Define

$$\mu_j := \mathbb{E}[Z_j \mid \mathbf{x}^0_{-j}], \qquad \Delta := Z_j - \mu_j, \qquad \sigma_j^2 := \mathrm{Var}(Z_j \mid \mathbf{x}^0_{-j}) = \mathbb{E}[\Delta^2 \mid \mathbf{x}^0_{-j}].$$

Recall $g_\theta(z) := l_{-j}(\theta; \mathbf{x}^0_{-j}, z)$. By Assumption A.3(i), $g_\theta$ is three times continuously differentiable in a neighborhood of $\mu_j$. Taylor's theorem gives

$$g_\theta(\mu_j + \Delta) = g_\theta(\mu_j) + g_\theta'(\mu_j)\Delta + \frac{1}{2}g_\theta''(\mu_j)\Delta^2 + R_\theta(\Delta), \tag{20}$$

where

$$R_\theta(\Delta) = \frac{1}{6} g_\theta^{(3)}(\xi_j) \Delta^3 \tag{21}$$

for some arbitrary $\xi_j$ lying between $\mu_j$ and $\mu_j + \Delta$.

**Step 2: Take conditional expectation under the RB ideal.** By definition,

$$l^{\mathrm{RB}}(\theta; \mathbf{x}^0_{-j}) = \mathbb{E}\left[g_\theta(Z_j) \mid \mathbf{x}^0_{-j}\right] = \mathbb{E}\left[g_\theta(\mu_j + \Delta) \mid \mathbf{x}^0_{-j}\right].$$

Taking $\mathbb{E}[\cdot \mid \mathbf{x}^0_{-j}]$ on both sides of Eq. (20) yields

$$l^{\mathrm{RB}}(\theta; \mathbf{x}^0_{-j}) = g_\theta(\mu_j) + g_\theta'(\mu_j)\,\mathbb{E}[\Delta \mid \mathbf{x}^0_{-j}] + \frac{1}{2}g_\theta''(\mu_j)\,\mathbb{E}[\Delta^2 \mid \mathbf{x}^0_{-j}] + \mathbb{E}\left[R_\theta(\Delta) \mid \mathbf{x}^0_{-j}\right] \tag{22}$$

$$= g_\theta(\mu_j) + \frac{1}{2}\sigma_j^2\, g_\theta''(\mu_j) + \mathbb{E}\left[R_\theta(\Delta) \mid \mathbf{x}^0_{-j}\right], \tag{23}$$

where $\mathbb{E}[\Delta \mid \mathbf{x}^0_{-j}] = 0$ and $\mathbb{E}[\Delta^2 \mid \mathbf{x}^0_{-j}] = \sigma_j^2$.

**Step 3: Bound the third-order remainder.** Under Assumption A.3(i), there exists $r > 0$ such that $M_j := \sup_{|z-\mu_j|\leq r} |g_\theta^{(3)}(z)| < \infty$, (a smoothness constant). Since we have $|\xi_j - \mu_j| \leq |\Delta| \leq r$, $|g_\theta^{(3)}(\xi_j)| \leq M_j$ and $|R_\theta(\Delta)| \leq \frac{M_j}{6} |\Delta|^3$. Taking conditional expectation and using Assumption A.3(ii) (finite third moment) yields

$$\left|\mathbb{E}\left[R_\theta(\Delta) \mid \mathbf{x}^0_{-j}\right]\right| \leq \frac{M_j}{6} \mathbb{E}\left[|\Delta|^3 \mid \mathbf{x}^0_{-j}\right], \tag{24}$$

which is the local remainder of approximation. Combining Eq. (23) with (24) justifies the local approximation.

**Remark (relative error scaling).** Relative to the second-order term scale $\sigma_j^2$, the approximation error is controlled by $M_j \sigma_j$ up to a standardized third absolute moment:

$$\frac{\left|\mathbb{E}[R_\theta(\Delta) \mid \mathbf{x}_{-j}^0]\right|}{\sigma_j^2} \lesssim M_j \sigma_j \cdot \frac{\mathbb{E}[|\Delta|^3 \mid \mathbf{x}_{-j}^0]}{\sigma_j^3}.$$

In particular, when the conditional distribution is not too heavy-tailed so that $\mathbb{E}[|\Delta|^3]/\sigma_j^3$ is moderate, the relative approximation error primarily scales with the local smoothness constant $M_j$ and the conditional spread $\sigma_j$.

**Step 4: Bias–variance decomposition.** Recall that for a fixed completion $z_j$,

$$g_\theta(z_j) := l_{-j}(\theta; \mathbf{x}_{-j}^0, z_j) = \mathbb{E}_W\left[\|\phi_\theta(z_j; W) - \mathbf{x}_{-j}^0\|_2^2\right],$$

where $W = (t, \varepsilon)$ collects diffusion randomness. Let $x := \mathbf{x}_{-j}^0$, $\phi := \phi_\theta(z_j; W)$, and $\bar{\phi} := \bar{\phi}_\theta(z_j) = \mathbb{E}_W[\phi_\theta(z_j; W)]$. Then

$$\begin{aligned}
g_\theta(z_j) &= \mathbb{E}_W\left[\|\phi - x\|_2^2\right] = \mathbb{E}_W\left[\|\phi - \bar{\phi} + \bar{\phi} - x\|_2^2\right] \\
&= \mathbb{E}_W\left[\|\phi - \bar{\phi}\|_2^2\right] + \|\bar{\phi} - x\|_2^2 + 2\langle \mathbb{E}_W[\phi - \bar{\phi}], \bar{\phi} - x\rangle \\
&= \|\bar{\phi}_\theta(z_j) - x\|_2^2 + \mathbb{E}_W\left[\|\phi_\theta(z_j; W) - \bar{\phi}_\theta(z_j)\|_2^2\right],
\end{aligned}$$

where the cross term vanishes since $\mathbb{E}_W[\phi - \bar{\phi}] = 0$. Defining $s_\theta(z_j) := \mathbb{E}_W[\|\phi_\theta(z_j; W) - \bar{\phi}_\theta(z_j)\|_2^2]$ yields $g_\theta(z_j) = \|\bar{\phi}_\theta(z_j) - \mathbf{x}_{-j}^0\|_2^2 + s_\theta(z_j)$.

**Step 5: Differentiate and plugin** Define $e_\theta(z_j) := \bar{\phi}_\theta(z_j) - \mathbf{x}_{-j}^0 \in \mathbb{R}^{d-1}$, so that $g_\theta(z_j) = \|e_\theta(z_j)\|_2^2 + s_\theta(z_j)$. We obtain

$$g_\theta''(z_j) = 2\|\bar{\phi}_\theta'(z_j)\|_2^2 + 2 e_\theta(z_j)^\top \bar{\phi}_\theta''(z_j) + s_\theta''(z_j). \tag{25}$$

Combining Eq. (23) with Eq. (25) and evaluating at $z_j = \mu_j$ gives

$$\begin{aligned}
l^{\mathrm{RB}}(\theta; \mathbf{x}_{-j}^0) &= g_\theta(\mu_j) + \frac{1}{2}\sigma_j^2 g_\theta''(\mu_j) + \mathbb{E}[R_\theta(\Delta) \mid \mathbf{x}_{-j}^0] \\
&= g_\theta(\mu_j) + \sigma_j^2 \|\bar{\phi}_\theta'(\mu_j)\|_2^2 + \sigma_j^2 (\bar{\phi}_\theta(\mu_j) - \mathbf{x}_{-j}^0)^\top \bar{\phi}_\theta''(\mu_j) + \frac{1}{2}\sigma_j^2 s_\theta''(\mu_j) + \mathbb{E}[R_\theta(\Delta) \mid \mathbf{x}_{-j}^0].
\end{aligned} \tag{26}$$

Up to the third-order remainder term, the dominant correction $\sigma_j^2 \|\bar{\phi}_\theta'(\mu_j)\|_2^2$ penalizes sensitivity of the mean reconstruction to the missing coordinate, with weight proportional to the oracle conditional variance $\sigma_j^2$. $\qquad\square$

# B. When Extra Supervision from AugFull Helps

We now instantiate a simple Gaussian model that captures the key factors controlling the AugMask–AugFull tradeoff: (i) dependence between the missing coordinate and the observed block, (ii) missingness rate, (iii) sample size, and (iv) dimension.

## B.1. A Toy Gaussian Setup and a Surrogate Target

**Data model (one shared factor)**   Fix the missing coordinate index $j$ and write $\mathbf{X}^0 = (\mathbf{X}^0_{-j}, X^0_j) \in \mathbb{R}^{d-1} \times \mathbb{R}$. Let $S \sim \mathcal{N}(0, 1)$ be a shared latent factor, and generate

$$X^0_k = S + \eta_k, \qquad\qquad k \neq j, \ \ \eta_k \overset{i.i.d.}{\sim} \mathcal{N}(0, \sigma_o^2), \tag{27}$$

$$X^0_j = \rho S + \eta_j, \qquad\qquad \eta_j \sim \mathcal{N}(0, \sigma_m^2), \tag{28}$$

independently of $S$, where $\rho \in \mathbb{R}$ controls the coupling between $X^0_j$ and the observed block. Define the observed-block average

$$Y := \frac{1}{d-1} \sum_{k \neq j} X^0_k = S + \bar{\eta}, \qquad \bar{\eta} := \frac{1}{d-1} \sum_{k \neq j} \eta_k \sim \mathcal{N}\Big(0, \frac{\sigma_o^2}{d-1}\Big). \tag{29}$$

**Missingness and augmentation mismatch**   Let $M_j \sim \mathrm{Bernoulli}(1 - p_{\mathrm{mss}})$ indicate whether coordinate $j$ is observed ($M_j = 1$) or missing ($M_j = 0$). For simplicity we assume MCAR, i.e., $M_j$ is independent of $(S, \mathbf{X}^0)$.

When $M_j = 1$, the $j$th entry is available; when $M_j = 0$, we replace it with an augmented value $Z_j$. To model mismatch of the augmentation rule in a tractable way, we assume

$$Z_j := X^0_j + b + \nu, \qquad \nu \sim \mathcal{N}(0, \tau^2), \tag{30}$$

where $b \in \mathbb{R}$ represents a systematic shift and $\tau^2 \geq 0$ represents additional augmentation variance beyond the true conditional variability. Define the effective supervised target used for coordinate $j$ as

$$A := \begin{cases} X^0_j, & M_j = 1, \\ Z_j, & M_j = 0, \end{cases} \quad \Longleftrightarrow \quad A = \rho S + \eta_j + \mathbb{I}_{M_j=0}(b + \nu). \tag{31}$$

Under the MCAR assumption, the indicator $\mathbb{I}_{M_j=0}$ is independent of $(S, \bar{\eta}, \eta_j)$ and of the augmentation noise $\nu$.

**MSE objectives and a surrogate target**   We adopt a simplified *shared-representation* setup: a scalar predictor $\widehat{Y}$ is used to reconstruct the observed block (so $\widehat{Y}$ is trained to match $Y$), and the $j$th coordinate is predicted as $\rho\widehat{Y}$. Under this abstraction, the two training objectives reduce to simple quadratic forms in $\widehat{Y}$:

$$\ell_{\mathrm{Mask}}(\widehat{Y}) := (d-1)(\widehat{Y} - Y)^2, \qquad \ell_{\mathrm{Full}}(\widehat{Y}) := (d-1)(\widehat{Y} - Y)^2 + (\rho\widehat{Y} - A)^2. \tag{32}$$

Thus, AugMask optimizes the observed-only term, whereas AugFull adds an additional supervised term on coordinate $j$ through $A$.

**Lemma B.1** (Surrogate target for AugFull). *Let $K := d - 1 + \rho^2$ and define*

$$\widetilde{Y} := \frac{(d-1)Y + \rho A}{K}. \tag{33}$$

*Then $\ell_{\mathrm{Full}}(\widehat{Y})$ can be rewritten as*

$$\ell_{\mathrm{Full}}(\widehat{Y}) = K(\widehat{Y} - \widetilde{Y})^2 + C(Y, A),$$

*where $C(Y, A)$ does not depend on $\widehat{Y}$. In particular, minimizing $\ell_{\mathrm{Full}}$ over $\widehat{Y}$ is equivalent to minimizing $(\widehat{Y} - \widetilde{Y})^2$.*

*Proof.* Expand and collect terms in $\widehat{Y}$:

$$(d-1)(\widehat{Y} - Y)^2 + (\rho\widehat{Y} - A)^2 = (d - 1 + \rho^2)\widehat{Y}^2 - 2\widehat{Y}\big((d-1)Y + \rho A\big) + \big((d-1)Y^2 + A^2\big).$$

Let $K = d - 1 + \rho^2$ and $\widetilde{Y} = \frac{(d-1)Y+\rho A}{K}$. Then

$$K\widehat{Y}^2 - 2K\widehat{Y}\widetilde{Y} = K(\widehat{Y} - \widetilde{Y})^2 - K\widetilde{Y}^2,$$

which yields the result with $C(Y, A) := (d - 1)Y^2 + A^2 - K\widetilde{Y}^2$, independent of $\widehat{Y}$. $\qquad\square$

**Implication.** Lemma B.1 shows that AugFull implicitly trains $\widehat{Y}$ toward the surrogate target $\widetilde{Y}$, which mixes the observed signal $Y$ and the supervised/augmented signal $A$. AugMask, in contrast, trains directly toward $Y$. The next section compares these two targets under the toy metric $\mathbb{E}[(\widehat{Y} - S)^2]$ and yields an explicit crossover condition.

## B.2. Derivation of the Crossover Inequality

We analyze a stylized setting to characterize when the observed-only objective (AugMask) can be preferable to supervising all coordinates (AugFull). As a proxy for representation quality, we evaluate the Mean Squared Error (MSE) with respect to the latent factor $S$:

$$\mathrm{MSE}(\widehat{Y}) := \mathbb{E}\left[(\widehat{Y} - S)^2\right].$$

For tractability, we compare the *per-sample minimizers* of the quadratic objectives, treating $\widehat{Y}$ as a free scalar for each input. Under this simplification,

- **AugMask estimator:** $\widehat{Y}_{\mathrm{Mask}} = Y$.

- **AugFull estimator:** $\widehat{Y}_{\mathrm{Full}} = \widetilde{Y}$, where $\widetilde{Y}$ is defined in Eq. (33).

(When $\rho = 0$, the additional AugFull term does not constrain $\widehat{Y}$ and the two objectives coincide; below we assume $\rho \neq 0$ and $d > 1$.)

**MSE of AugMask** Using $Y = S + \bar{\eta}$ (Eq. (29)), the estimation error is

$$Y - S = \bar{\eta}.$$

Therefore the MSE equals the variance of the observed-block average noise:

$$\mathrm{MSE}_{\mathrm{Mask}} = \mathbb{E}[\bar{\eta}^2] = \mathrm{Var}(\bar{\eta}) = \frac{\sigma_o^2}{d - 1}. \tag{34}$$

**MSE of AugFull** We analyze $\widetilde{Y} - S$ directly. Recall $\widetilde{Y} = \frac{(d-1)Y+\rho A}{K}$ with $K = d - 1 + \rho^2$. Substituting $Y = S + \bar{\eta}$ and

$$A = \rho S + \eta_j + \mathbb{I}_{M_j=0}(b + \nu),$$

yields

$$\widetilde{Y} = \frac{(d - 1)(S + \bar{\eta}) + \rho(\rho S + \eta_j + \mathbb{I}_{M_j=0}(b + \nu))}{K}$$
$$= S + \frac{(d - 1)\bar{\eta} + \rho\eta_j + \rho\mathbb{I}_{M_j=0}(b + \nu)}{K}, \tag{35}$$

and hence

$$\widetilde{Y} - S = \frac{(d - 1)\bar{\eta} + \rho\eta_j + \rho\mathbb{I}_{M_j=0}(b + \nu)}{K}. \tag{36}$$

Under MCAR, the missingness indicator $\mathbb{I}_{M_j=0}$ is independent of $(S, \bar{\eta}, \eta_j)$ and the augmentation noise $\nu$. Moreover, $\bar{\eta}, \eta_j$, and $\nu$ are mutually independent and mean-zero. Consequently, cross terms vanish in $\mathbb{E}[(\widetilde{Y} - S)^2]$ (since each cross term contains a mean-zero factor mutually independent), and the MSE is the sum of component *second moments* normalized by $K^2$:

- $\mathbb{E}[((d - 1)\bar{\eta})^2] = (d - 1)^2 \mathrm{Var}(\bar{\eta}) = (d - 1)\sigma_o^2.$

- $\mathbb{E}[(\rho\eta_j)^2] = \rho^2 \operatorname{Var}(\eta_j) = \rho^2\sigma_m^2.$

- $\mathbb{E}[(\rho\mathbb{I}_{M_j=0}(b+\nu))^2] = \rho^2\,\mathbb{E}[\mathbb{I}_{M_j=0}(b+\nu)^2] = \rho^2 p_{\mathrm{mss}}\,\mathbb{E}[(b+\nu)^2] = \rho^2 p_{\mathrm{mss}}(b^2+\tau^2).$

Combining these terms gives

$$\mathrm{MSE}_{\mathrm{Full}} = \mathbb{E}[(\widetilde{Y}-S)^2] = \frac{(d-1)\sigma_o^2 + \rho^2\sigma_m^2 + \rho^2 p_{\mathrm{mss}}(b^2+\tau^2)}{(d-1+\rho^2)^2}. \tag{37}$$

### B.3. Crossover condition

AugFull is preferable under this toy metric when $\mathrm{MSE}_{\mathrm{Full}} < \mathrm{MSE}_{\mathrm{Mask}}$; otherwise AugMask is preferable. Substituting Eqs. (34)–(37) and rearranging yields the following explicit inequality.

**Corollary B.2** (Crossover inequality under the toy metric)**.** *Define the* effective augmentation mismatch *(weighted by missing rate)*

$$\delta_{\mathrm{aug}}^2 \;:=\; p_{\mathrm{mss}}(b^2+\tau^2).$$

*Assume $\rho \neq 0$ and $d > 1$. Then $\mathrm{MSE}_{\mathrm{Full}} < \mathrm{MSE}_{\mathrm{Mask}}$ holds if and only if*

$$\delta_{\mathrm{aug}}^2 \;<\; \sigma_o^2\left(2 + \frac{\rho^2}{d-1}\right) - \sigma_m^2. \tag{38}$$

*When the right-hand side is non-positive, (38) cannot hold and AugMask is strictly preferable under this toy setup.*

*Proof.* Starting from $\mathrm{MSE}_{\mathrm{Full}} < \mathrm{MSE}_{\mathrm{Mask}}$ and substituting (34)–(37) gives

$$\frac{(d-1)\sigma_o^2 + \rho^2\sigma_m^2 + \rho^2\delta_{\mathrm{aug}}^2}{K^2} < \frac{\sigma_o^2}{d-1}, \qquad K = d-1+\rho^2.$$

Multiplying both sides by $K^2(d-1)$ yields

$$(d-1)\big((d-1)\sigma_o^2 + \rho^2\sigma_m^2 + \rho^2\delta_{\mathrm{aug}}^2\big) < K^2\sigma_o^2.$$

Expanding $K^2 = ((d-1)+\rho^2)^2 = (d-1)^2 + 2(d-1)\rho^2 + \rho^4$ and canceling $(d-1)^2\sigma_o^2$ on both sides gives

$$(d-1)\rho^2\sigma_m^2 + (d-1)\rho^2\delta_{\mathrm{aug}}^2 < \big(2(d-1)\rho^2 + \rho^4\big)\sigma_o^2.$$

Dividing by $(d-1)\rho^2$ (using $\rho \neq 0$ and $d > 1$) yields

$$\sigma_m^2 + \delta_{\mathrm{aug}}^2 < \left(2 + \frac{\rho^2}{d-1}\right)\sigma_o^2,$$

which rearranges to (38). $\qquad\square$

**Interpretation.** Eq. (38) gives a simple tolerance condition under the simple toy setup. AugFull can help only when the augmentation mismatch on missing examples, $\delta_{\mathrm{aug}}^2 = p_{\mathrm{mss}}(b^2+\tau^2)$, is small enough; otherwise the additional supervised term pulls the shared representation toward a biased/noisy surrogate target. The tolerance increases with $\sigma_o^2$ and $|\rho|$, and decreases with $\sigma_m^2$.

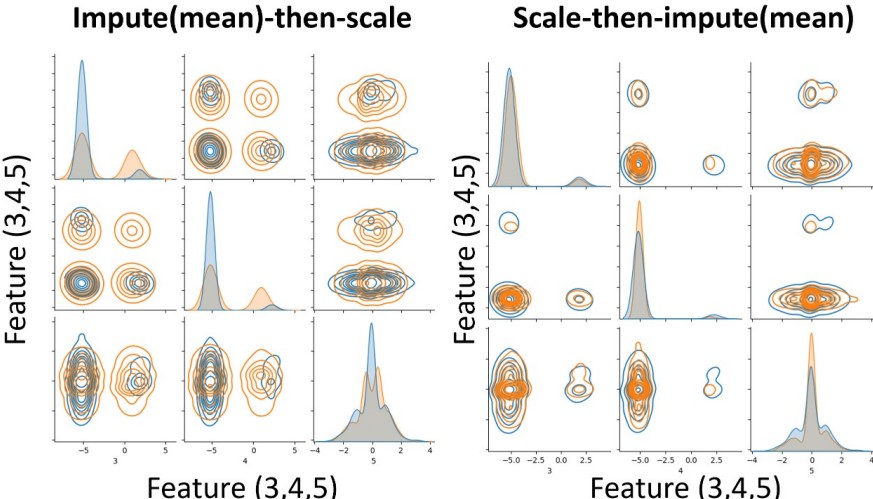

*Figure 8.* **Preprocessing Order.** (Left) Impute-then-scale allows imputed values to skew transformation statistics. (Right) Scale-then-impute (ours) calculates statistics solely on observed data, stabilizing the distribution.

## C. Implementation Details

### C.1. Datasets

| Dataset | # Rows | # Num | # Cat | Task |
|---|---|---|---|---|
| Adult | 48,842 | 6 | 9 | Classification |
| Default | 30,000 | 14 | 11 | Classification |
| Shoppers | 12,330 | 10 | 8 | Classification |
| Magic | 19,019 | 10 | 1 | Classification |
| Beijing | 43,824 | 7 | 5 | Regression |
| News | 39,644 | 46 | 2 | Regression |

### C.2. Preprocessing

The order of preprocessing operations (scaling vs. imputation) critically impacts downstream performance. We distinguish between two strategies:

- **Impute-then-scale:** Imputes missing values first, then computes scaling parameters (e.g., mean, variance) on the completed dataset.

- **Scale-then-impute:** Computes scaling parameters using *only observed entries*, transforms the data, and then imputes.

We adopt the **Scale-then-impute** strategy (see Figure 8). Unlike "Impute-then-scale," which allows imputed values to bias the scaling statistics (often distorting the distribution), "Scale-then-impute" ensures transformations are derived solely from observed data. This stabilizes the variance and yields more representative inputs for the model.

#### C.2.1. CONTINUOUS VARIABLES

Following the preprocessing setup of CDTD (Mueller et al., 2025), we apply scikit-learn's QuantileTransformer to continuous variables and then standardize them to zero mean and unit variance (Pedregosa et al., 2011).

### C.2.2. CATEGORICAL VARIABLES

We encode categorical variables into integer labels using `LabelEncoder`. For models requiring embeddings (e.g., TabSyn, CDTD), we additionally use `OrdinalEncoder` to map these labels to contiguous indices.

To handle unseen categories in the validation set, we reserve extra tokens in the embedding matrix during training. At inference, unseen categories are mapped to these reserved indices and explicitly excluded from loss computation (e.g., cross-entropy).

## C.3. Missing Mechanisms & Mask Generation

The data generation process governing whether each entry is observed or missing (Rubin, 1976) can be categorized as:

- **M**issing **C**ompletely **A**t **R**andom (MCAR): the missingness is independent of the data, $\mathbf{M} \perp \mathbf{X}$.

- **M**issing **A**t **R**andom (MAR): the missingness depends only on observed entries, $\mathbf{M} \perp \mathbf{X}^{\mathrm{mis}} \mid \mathbf{X}^{\mathrm{obs}}$.

- **M**issing **N**ot **A**t **R**andom (MNAR): all other cases.

## C.4. Generating MAR Masks

For a target missing ratio $p$, we generate a binary mask $\mathbf{M} \in \{0,1\}^{n \times d_{\mathrm{full}}}$ on the full table $\mathbf{X}_{\mathrm{full}} = [\mathbf{X}, \mathbf{y}]$, so all columns (including $y$) share the same overall rate. MAR masks are sampled entrywise as $M_{ij} \sim \mathrm{Bernoulli}(p_{ij})$, where $p_{ij}$ comes from a covariate-dependent logistic propensity whose intercept is calibrated to match the marginal rate $p$. For each $(p, \mathrm{seed})$, we generate one mask and reuse it across all methods. See Zhang et al. (2025) for details.

## C.5. Constructing Augmented Dataset

### C.5.1. UNIVARIATE BASELINES (FEATURE-WISE)

**Zero / Mean (deterministic).** **Zero** uses a fixed placeholder (0 for numerical; a sentinel token for categorical features). **Mean** uses a fixed marginal summary (mean for numerical; mode for categorical). Both correspond to point-mass choices of $T$.

**Noise / E-CDF (marginal stochastic sampling).** These baselines sample each missing feature independently from its *marginal* distribution estimated on observed training entries. For numerical features, **Noise** samples from a fitted Gaussian $z_j \sim \mathcal{N}(\hat{\mu}_j, \hat{\sigma}_j^2)$, whereas **E-CDF** samples with replacement from the empirical pool of observed values. For categorical features, both sample $z_j \sim \mathrm{Categorical}(\hat{\pi}_j)$ using observed class frequencies.

### C.5.2. MULTIVARIATE BASELINES (JOINT / CONDITIONAL)

**Iterative imputers (MICE / HyperImpute).** Both methods construct a completed dataset by iterating multivariate imputation models. In our evaluation, we use the default configurations provided by each implementation and run the imputer once to produce a single completion of the missing part. See (Shah et al., 2014; Jarrett et al., 2022).

**LightGBM conditional modules (LGBM-D / LGBM-S).** Both variants follow Algorithm 1 and fit one per-feature LightGBM predictor (Ke et al., 2017) on observed rows ($m_{ij} = 1$), conditioning on the observed context and the mask

| Parameter | Value | Parameter | Value |
|---|---|---|---|
| boosting_type | gbdt | subsample | 1.0 |
| num_leaves | 31 | subsample_freq | 0 |
| max_depth | −1 | colsample_bytree | 1.0 |
| learning_rate | 0.1 | reg_alpha | 0.0 |
| n_estimators | 50 | reg_lambda | 0.0 |
| subsample_for_bin | 200000 | random_state | 42 |
| min_split_gain | 0.0 | importance_type | split |
| min_child_weight | 0.001 | min_child_samples | 20 |

*Table 4.* **LightGBM defaults.** Hyperparameters used for all per-feature conditional models in LGBM-D/LGBM-S.

$\mathbf{u}_i = (\tilde{\mathbf{x}}_{i,-j}, \mathbf{m}_{i,-j})$ (line 7). Including $\mathbf{m}_{i,-j}$ ensures the augmentor directly conditions on the missingness pattern. **LGBM-D** uses point predictions (deterministic fill), whereas **LGBM-S** uses the sampling steps in Algorithm 1.

**LightGBM hyperparameters.** Unless stated otherwise, all LightGBM models used in LGBM-D and LGBM-S (Algorithm 1) are trained with the default settings in Table 4 (shared across regression and classification heads), with `random_state=42` for reproducibility.

*Table 5.* **Sampling frequency ablations.** Cycling (non-stationary inputs) hurts recall, while a higher anchor fraction mitigates the drop. Full cycling can slightly improve MLE (utility).

| Setting | Beta-Recall (All) ↑ | MLE ↑ |
|---|---|---|
| LGBM-S, cycled ($n_{\text{cache}} = 1$, used in paper) | 0.3305 | 0.8877 |
| LGBM-S, cycled ($n_{\text{cache}} = 10$) | 0.1553 | 0.8877 |
| LGBM-S, cycled ($n_{\text{cache}} = 30$) | 0.1679 | 0.8924 |
| Anchor + cycle ($n_{\text{cache}} = 30$), 100% Cycle | 0.2125 | 0.8978 |
| Anchor + cycle ($n_{\text{cache}} = 30$), 50% Fixed / 50% Cycle | 0.2481 | 0.8906 |
| Anchor + cycle ($n_{\text{cache}} = 30$), 80% Fixed / 20% Cycle | 0.2724 | 0.8884 |

### C.5.3. HOW MANY TIMES DO WE SAMPLE? (FIXED VS. CYCLED CACHED AUGMENTATIONS)

**Setup: what is a "cached augmentation"?** For a stochastic augmentation rule $T(\cdot \mid \mathbf{x}^{\text{obs}}, \mathbf{m})$ a *cached augmentation* is a pre-constructed augmented dataset $\mathcal{D}^{A,(k)} = \{(\mathbf{x}_i^{A,(k)}, \mathbf{m}_i)\}_{i=1}^N$ where, for each training point $i$ and each missing coordinate $j$, we draw $z_{ij}^{(k)} \sim T(\cdot \mid \mathbf{x}_i^{\text{obs}}, \mathbf{m}_i)$ once and set $x_{ij}^{A,(k)} = m_{ij} x_{ij}^{\text{obs}} + (1 - m_{ij}) z_{ij}^{(k)}$. We generate $n_{\text{cache}}$ such datasets $\{\mathcal{D}^{A,(k)}\}_{k=1}^{n_{\text{cache}}}$ *offline* and reuse them during training.

**Default (used in the paper): sample once and keep it fixed.** We construct a single augmented dataset $\mathcal{D}^{A,(1)}$ and keep it fixed across all epochs. This makes the mapping from *observed context → augmented input* stationary and yields the best coverage (recall) in Table 5.

**Cycling cached augmentations changes the per-sample input across epochs.** We also tested *cycling* among $n_{\text{cache}}$ cached datasets: at the beginning of each epoch (or training pass), we switch the entire training set to one cache element $\mathcal{D}^{A,(k)}$ (e.g., round-robin over $k \in \{1, \ldots, n_{\text{cache}}\}$). Although the loss is masked to observed coordinates, the augmented values still enter the denoiser *as inputs*; thus, cycling alters the per-sample inputs over time and can create a *moving-target* optimization effect or the "multi-task learning challenge", reducing fidelity/coverage (recall). When $n_{\text{cache}} = 1$, cycling is identical to the fixed setting.

**Why this is consistent with "estimating variance is enough."** Our theory motivates the leading effect of marginalizing a missing coordinate through its conditional moments: the variance term controls the invariance/sensitivity correction in the local RB expansion (Prop. 3.1). In LGBM-S, each stochastic fill is generated *using* an explicit variance estimate $\hat{\sigma}^2(\mathbf{x}^{\text{obs}}, \mathbf{m})$. Thus, the variance information already enters training through the distribution of $Z_j$; repeatedly cycling multiple imputations primarily reduces Monte Carlo noise (and can encourage exploitation) but is not *required to realize the leading variance-weighted regularization effect. Hypothetically, if the conditional moment estimates $(\hat{\mu}, \hat{\sigma}^2)$ are misspecified, further reducing Monte Carlo noise does not reduce this bias; instead, it can make the model more consistently optimize against the same biased augmentation distribution, potentially yielding overconfident (shrunken) completions. In contrast, fixing a single stochastic draw preserves additional randomness in the training inputs, which can act as a stabilizing regularizer against such misspecification.*

**Anchor + cycling (cache size $n_{\text{cache}} = 30$).** To isolate the role of input drift, we mix a fixed *anchor* augmentation $\mathcal{D}^{A,\text{anc}}$ with a cycled cache $\{\mathcal{D}^{A,(k)}\}_{k=1}^{30}$: each epoch uses the anchor with probability $\rho$ and otherwise uses a cache element. As $\rho$ decreases (more cycling), recall drops (Table 5), supporting the moving-target explanation.

**Utility–coverage tradeoff and future prospects.** We observe a mild tradeoff: more cycling can slightly improve utility (MLE) but tends to reduce recall (coverage). A promising direction is a *curriculum* (high $\rho$ early, lower $\rho$ later) or multi-

objective schedules that adapt $\rho$ (or $n_{\mathrm{cache}}$) to target a desired utility–coverage balance, potentially with stronger conditional models or calibrated uncertainty estimators.

## C.6. Baseline Tabular Generative Models

### C.6.1. MISSING-AWARE

1. **MIWAE** (Mattei & Frellsen, 2019): We have modified the MIWAE imputation plugin from `https://github.com/vanderschaarlab/hyperimpute`, which fits $p_\theta(x^{\mathrm{obs}}|z)$ and $q_\gamma(z|x^{\mathrm{obs}})$ so that it can be refactored into a generative model. We sample z from the prior distribution $p(z)$ and generate $\mathbf{x} = p_\theta(\mathbf{x}|\mathbf{z})$. For training, we set `n_epochs` to 800 (default: 500), `latent_size` to 64 (default: 1), `n_hidden` to 256 (default: 1), and $K = 40$ (default: 20). This configuration resulted in a substantially improved results, which may be different from the findings of Zhang et al. (2025), where thy used the default hyperparameters. Increasing these hyperparameters values further did not improve performance, but only increased computation time and memory usage.

2. **MissDiff** We use the implementation provided in the supplementary material at `https://openreview.net/forum?id=PyyoSwPaSa`, which is a variant of TabCSDI (Zheng & Charoenphakdee, 2022) (see `https://github.com/pfnet-research/TabCSDI`). To align the model size with other methods, we set `batchsize` to 64, `diffusion_embedding_dim` and `timeembed` to 1024, `layers` to 5, and `channels` to 256. We observed that increasing `batchsize` beyond 128 caused memory issues, so we fixed it at 64. The number of training epochs was set to 250.

3. **ForestDiff** (Jolicoeur-Martineau et al., 2024) We use the default implementation from `https://github.com/SamsungSAILMontreal/ForestDiffusion`, setting $n_t = 20$ and $\mathrm{duplicate}_K = 10$ as recommended by the authors to reduce training time. The default parameters ($n_t = 50$, $\mathrm{duplicate}_K = 100$) did not converge within 3600 seconds. For the *news* dataset (48 columns), we additionally apply chunking with a size of 8,000, as processing all samples simultaneously failed to converge within 3600 seconds.

4. **DiffPuter** (Zhang et al., 2025). The official implementation (`https://github.com/hengruizhang98/DiffPuter`) applies binary encoding for categorical variables, while the paper describes one-hot encoding. We tested both and found that binary encoding can introduce imputation artifacts (e.g., spurious categories) and consistently underperformed one-hot encoding. We therefore follow the paper and use one-hot encoding for categoricals. The paper does not explicitly state whether parameters are carried across EM iterations; the official code re-initializes the model at the beginning of each E-step, which we adopt. Finally, to guard against early-stage divergence we monitor validation loss during EM and use it as a safety check (early termination / checkpoint selection).

### C.6.2. MISSING-UNAWARE

Note that for ARF, CTGAN, TVAE, TabSyn, and CDTD, we follow the experimental setup described in (Mueller et al., 2025). Unless stated otherwise, we use the official/default configurations provided by each baseline and keep the training budget fixed (30k steps, batch size 4096) for consistency across methods.

1. **ARF** (Watson et al., 2023): We use the authors's suggested default hyperparameters. In particular, we use 20 trees, $\delta$ = 0 and a minimum node size of 5. We follow the official package implementation and set the maximum number of iterations to 10 (see https://github.com/bips-hb/arfpy).

2. **CTGAN** (Xu et al., 2019): We follow the popular implementation in the Synthetic Data Vault package (see https://github.com/sdv-dev/CTGAN). For this model to work, the batch size must be divisible by 10. Therefore, we adjust the batch size if necessary. We use a 256-dimensional embedding (instead of the default embedding dimension of 128) to better align the CTGAN architecture with TVAE, TabSyn and CDTD.

3. **TVAE** (Xu et al., 2019): We again follow the implementation in the Synthetic Data Vault. We use a 256-dimensional embedding to better align the architecture with CTGAN, TabSyn and CDTD.

4. **TabSyn** (Zhang et al., 2023): We use the default hyperparameters as suggested by the authors. The training steps that go towards training the VAE and the denoising network follow the proportions given in the official code (see https://github.com/amazon-science/tabsyn). To improve comparability to CDTD, we use the same neural network architecture as TabDDPM, which only differs slightly from the original architecture. We leave the VAE untouched.

5. **TabDDPM** (Kotelnikov et al., 2023): 3-layer MLP with 256 hidden units; 1000 diffusion steps with cosine scheduler and time embedding dim 128. We train for 30k steps with Adam (lr $2 \cdot 10^{-4}$) and EMA decay 0.999; sampling batch size is 2000.

6. **TabDiff** (Shi et al., 2025): Transformer-based denoiser with 2 layers ($d_{\text{token}} = 4$, 1 head) and a 5-layer MLP head (801 units), time embedding dim 256. Diffusion uses 50 timesteps with EDM-style preconditioning (e.g., $\sigma_{\min} = 0.002$, $\sigma_{\max} = 80$, $\sigma_{\text{data}} = 1.0$). We train for 30k steps with Adam (lr $10^{-3}$) and EMA decay 0.997; we use 200 generation steps at sampling time.

7. **CDTD** (Mueller et al., 2025): We set the hidden layers to 796, number of layers to 5, dimension of the MLP/time embedding to 256. Also, as CDTD uses the EDM framework (Karras et al., 2022), with $\sigma_{\min}^{\text{cat}} = 0, \sigma_{\max}^{\text{cat}} = 100, \sigma_{\min}^{\text{cont}} = 0, \sigma_{\max}^{\text{cont}} = 80, \sigma_{\text{data}}^{\text{cat}} = 1.0, \sigma_{\text{data}}^{\text{cont}} = 1.0.$, we do not change this. Also, we use the learned noise-schedule configuration for *by-type*, which uses common noise-scheduling for the same data-types. For further details please refer to the Appendix in (Mueller et al., 2025).

## C.7. Applying Loss Masking to Tabular Diffusion Models

This section describes how we implement *loss masking* when training tabular diffusion models on incomplete inputs. Let $b \in \{1, \ldots, B\}$ index samples in a minibatch and $j \in \{1, \ldots, d\}$ index features. We write $m_{b,j} \in \{0, 1\}$ for the observation mask ($m_{b,j} = 1$ if feature $j$ is observed in sample $b$). When we need type-specific masking, we use $m_{b,j}^{\text{cont}}$ and $m_{b,j}^{\text{cat}}$ (equal to $m_{b,j}$ on numerical / categorical features, and 0 otherwise).

**Mask-then-reduce.** All implementations follow the same pattern: (i) compute a per-feature loss $\ell_{b,j}$, (ii) multiply by the mask, and (iii) normalize using either *per-sample* or *global* normalization. With a small constant $\varepsilon > 0$ for stability, we use:

$$\text{Mean}_{\text{ps}}(\ell; m) := \frac{1}{B} \sum_{b=1}^{B} \frac{\sum_{j=1}^{d} m_{b,j} \, \ell_{b,j}}{\sum_{j=1}^{d} m_{b,j} + \varepsilon}, \tag{39}$$

$$\text{Mean}_{\text{gl}}(\ell; m) := \frac{\sum_{b=1}^{B} \sum_{j=1}^{d} m_{b,j} \, \ell_{b,j}}{\sum_{b=1}^{B} \sum_{j=1}^{d} m_{b,j} + \varepsilon}. \tag{40}$$

Per-sample normalization keeps each row's contribution comparable even if rows have different missingness rates, while global normalization keeps the overall gradient scale stable across batches.

### C.7.1. TABDDPM (HYBRID GAUSSIAN + MULTINOMIAL DIFFUSION)

**Design.** TabDDPM models numerical features with Gaussian diffusion and categorical features with multinomial diffusion, using a shared network backbone with type-specific heads.

**Masking + reduction (per-sample normalization).** We compute feature-wise losses and average them *within each sample* over observed entries.

- **Numerical noise-prediction loss.** For a numerical feature $j$ at timestep $t$:
$$\ell_{b,j}^{\text{simple}} = \|\epsilon_{b,j} - \epsilon_\theta(x_{t,b}, t)_j\|^2, \qquad L_{\text{simple}} = \text{Mean}_{\text{ps}}(\ell^{\text{simple}}; m^{\text{cont}}). \tag{41}$$

- **Categorical VLB/KL term.** For a categorical feature $j$:
$$\ell_{b,j}^{\text{vlb}} = \mathbb{D}_{\text{KL}}\Big(q(x_{t-1,b,j} \mid x_{t,b,j}, x_{0,b,j}) \,\|\, p_\theta(x_{t-1,b,j} \mid x_{t,b,j})\Big), \qquad L_{\text{vlb}} = \text{Mean}_{\text{ps}}(\ell^{\text{vlb}}; m^{\text{cat}}). \tag{42}$$

The (masked) TabDDPM training objective is then $\mathcal{L}_{\text{TabDDPM}} = L_{\text{simple}} + L_{\text{vlb}}$ (up to any model-specific scalar weights).

### C.7.2. TABDIFF (JOINT CONTINUOUS-TIME DIFFUSION WITH A TRANSFORMER)

**Design.** TabDiff uses a Transformer-based denoiser. Numerical features are trained with an EDM-style regression loss, while categorical features use an absorbing/noising mechanism and a discrete likelihood term.

**Masking + reduction (global normalization).** We sum masked losses over the batch and divide by the total number of observed entries (separately per type).

- **Numerical EDM-style loss.**

$$\ell_{b,j}^{\text{edm}} = w(\sigma_t) \left\| D_\theta(x_{t,b}, \sigma_t)_j - x_{0,b,j} \right\|^2, \qquad L_{\text{edm}} = \text{Mean}_{\text{gl}}\big(\ell^{\text{edm}}; m^{\text{cont}}\big). \tag{43}$$

- **Categorical absorbing / likelihood loss.** We use the negative log-likelihood form:

$$\ell_{b,j}^{\text{abs}} = w_t^{\text{elbo}} \big( -\log p_\theta(x_{0,b,j} \mid x_{t,b,j}) \big), \qquad L_{\text{abs}} = \text{Mean}_{\text{gl}}\big(\ell^{\text{abs}}; m^{\text{cat}}\big). \tag{44}$$

Finally,

$$\mathcal{L}_{\text{TabDiff}} = L_{\text{edm}} + L_{\text{abs}}. \tag{45}$$

### C.7.3. CDTD (CONTINUOUS DIFFUSION WITH CALIBRATED MIXED-TYPE LOSSES)

**Design.** CDTD maps all features into a continuous diffusion space (categoricals via embeddings / score interpolation) and uses auxiliary networks (e.g., *TimeWarping*, *WeightNet*) to stabilize training by matching and reweighting loss magnitudes.

**Masking.** We mask (i) the main per-feature diffusion loss and (ii) auxiliary objectives that regress to that per-feature loss; otherwise, missing features would contribute spurious gradients via nonzero auxiliary outputs.

**Calibrated per-feature loss.** Let $\ell_{b,j}^{\text{mse}}$ be the numerical MSE-type loss and $\ell_{b,j}^{\text{ce}}$ be the categorical CE-type loss (as used by the model). We apply a feature weight $\omega_j$ to balance heterogeneous scales; for categorical features, $\omega_j$ is typically based on an entropy-like normalization constant $Z_j$:

$$\omega_j = \begin{cases} 1/Z_j, & \text{if } j \text{ is categorical}, \\ 1, & \text{if } j \text{ is numerical}. \end{cases} \tag{46}$$

Then

$$\ell_{b,j}^{\text{calib}} = \omega_j \cdot \begin{cases} \ell_{b,j}^{\text{ce}}, & j \text{ categorical}, \\ \ell_{b,j}^{\text{mse}}, & j \text{ numerical}, \end{cases} \qquad L_{b,j}^{\text{calib}} = m_{b,j}\, \ell_{b,j}^{\text{calib}}. \tag{47}$$

**Auxiliary losses (masked).** Let $\widehat{L}_{b,j}^{\text{calib}}$ denote the auxiliary prediction of the calibrated loss (for timewarping), and let $\text{reweight}_t$ denote the scalar output used for time-dependent reweighting. We mask both auxiliary terms:

$$L_{b,j}^{\text{warp}} = m_{b,j}\, \frac{\big(\widehat{L}_{b,j}^{\text{calib}} - L_{b,j}^{\text{calib}}\big)^2}{p(t) + \varepsilon}, \tag{48}$$

$$L_{b,j}^{\text{weight}} = m_{b,j}\, \Big( \exp(\text{reweight}_t) - L_{b,j}^{\text{calib}} \Big)^2. \tag{49}$$

**Total objective.** We reduce by the number of observed entries to keep the overall scale comparable across different missingness rates:

$$\mathcal{L}_{\text{CDTD}} = \frac{\sum_{b=1}^{B} \sum_{j=1}^{d} \Big( L_{b,j}^{\text{calib}} + L_{b,j}^{\text{warp}} + L_{b,j}^{\text{weight}} \Big)}{\sum_{b=1}^{B} \sum_{j=1}^{d} m_{b,j} + \varepsilon}. \tag{50}$$

### C.8. Hardware Specifications

### C.9. Evaluation Metrics

Let $\mathcal{D}_r = \{\mathbf{x}^{(i)}\}_{i=1}^{n_r}$ be the real dataset and $\mathcal{D}_s = \{\tilde{\mathbf{x}}^{(i)}\}_{i=1}^{n_s}$ be the synthetic set. We report **Shape**, **Trend**, $\alpha$-**Precision**, $\beta$-**Recall**, and **MLE** using our TMTC protocol in Fig. 3.

**(1) Shape (column-wise density match).** For each numerical column $j \in \mathcal{J}_{\text{num}}$, let $F_j^r$ and $F_j^s$ be the empirical CDFs from $\mathcal{D}_r$ and $\mathcal{D}_s$. The Kolmogorov–Smirnov distance is

$$\text{KST}_j := \sup_x \big| F_j^r(x) - F_j^s(x) \big|. \tag{51}$$

| Component | Details |
|---|---|
| GPU | NVIDIA RTX 4090 (24GB), Driver 550.127.05 |
| CPU | AMD EPYC 9254 24-Core Processor |
| RAM | 256 GB |
| OS | Ubuntu 22.04.3 LTS (Linux 5.15.0-105-generic) |
| PyTorch | 2.2.2 |
| CUDA | 12.1 |
| cuDNN | 8902 |

*Table 6.* System specifications used for all experiments.

For each categorical column $j \in \mathcal{J}_{\text{cat}}$ with category set $K_j$, let $R_j(\omega)$ and $S_j(\omega)$ be empirical frequencies. The total-variation distance is

$$\text{TVD}_j := \frac{1}{2} \sum_{\omega \in K_j} \left| R_j(\omega) - S_j(\omega) \right|. \tag{52}$$

We aggregate them into a single similarity score (larger is better) by

$$\text{Shape} := 1 - \frac{1}{d_{\text{full}}} \left( \sum_{j \in \mathcal{J}_{\text{num}}} \text{KST}_j + \sum_{j \in \mathcal{J}_{\text{cat}}} \text{TVD}_j \right). \tag{53}$$

**(2) Trend (pair-wise dependency match).** For numerical columns $a, b \in \mathcal{J}_{\text{num}}$, let $\rho_{a,b}^r$ and $\rho_{a,b}^s$ denote Pearson correlation coefficients computed on $\mathcal{D}_r$ and $\mathcal{D}_s$. The Pearson discrepancy (averaged over pairs) is

$$\text{PearsonScore} := \frac{1}{2} \mathbb{E}_{(a,b)} \left| \rho_{a,b}^r - \rho_{a,b}^s \right|. \tag{54}$$

For categorical columns $A, B \in \mathcal{J}_{\text{cat}}$, let $R_{\alpha,\beta}$ and $S_{\alpha,\beta}$ be the empirical joint frequencies in the contingency tables. The contingency discrepancy is

$$\text{ContingencyScore} := \frac{1}{2} \sum_{\alpha \in A} \sum_{\beta \in B} \left| R_{\alpha,\beta} - S_{\alpha,\beta} \right|. \tag{55}$$

**(3) $\alpha$-Precision / (4) $\beta$-Recall (sample-wise fidelity and coverage).** We report the sample-wise metrics of Alaa et al. (2022): $\alpha$-PRECISION measures *fidelity*, i.e., the fraction of synthetic samples that lie within the support of the real data (plausible samples), while $\beta$-RECALL measures *coverage*, i.e., the extent to which the synthetic set covers the real-data distribution (real samples are close to some synthetic sample). Higher is better for both.

**(5) MLE (machine learning efficiency / downstream utility).** We evaluate downstream utility under the **TSTR** protocol: we train a fixed XGBoost classifier/regressor on synthetic data and evaluate it on a held-out *real* test set (AUROC for classification; RMSE for regression). For comparability across datasets, we normalize by the corresponding **TRTR** score (train and test on real data), which serves as an upper bound. Let $s_{\text{TSTR}}$ and $s_{\text{TRTR}}$ denote the downstream scores; we report

$$\text{MLE} = \begin{cases} \dfrac{s_{\text{TSTR}}}{s_{\text{TRTR}}}, & \text{classification (AUROC; higher is better)}, \\ \dfrac{s_{\text{TRTR}}}{s_{\text{TSTR}}}, & \text{regression (RMSE; lower is better)}. \end{cases}$$

Thus, $\text{MLE} \approx 1$ indicates synthetic data matches real-data utility, while smaller values indicate a gap.

# D. Detailed Experimental Results

## D.1. Full Results (Rank) for MAR setting

*Table 7.* **MAR Results (90 Scenarios): Comparative Rank Analysis.** Cells display Average Rank (↓) ± Standard Deviation across 90 scenarios: 6 datasets × 5 missing ratios [0.1–0.9] × 3 seeds. Ranks are computed per experimental block; **Summary Rank** is the mean rank across all metrics within each block. Δ denotes absolute rank gain over the +NoAug baseline. ** and * indicate statistical **Gain** and **Match** via Wilcoxon test ($p < 0.05$) against the *best-performing missing-aware baseline* per block among: MIWAE, MissDiff, DiffPuter, or ForestDiff. Boldface denotes results statistically superior to this entire baseline group.

| Method | Generative Quality | | | | | | | | Utility | | Summary | |
| | α-Precision | | β-Recall | | Trend | | Shape | | ML Task | | | |
| | Rank | Δ | Rank | Δ | Rank | Δ | Rank | Δ | Rank | Δ | **Rank** | Δ |
|---|---|---|---|---|---|---|---|---|---|---|---|---|
| **(A) Missing-aware baselines** | | | | | | | | | | | | |
| MIWAE | $7.2 \pm 4.0$ | – | $10.2 \pm 4.4$ | – | $8.9 \pm 4.6$ | – | $8.2 \pm 4.6$ | – | $10.3 \pm 4.5$ | – | 9.0 | – |
| MissDiff | $16.0 \pm 4.8$ | – | $17.1 \pm 4.5$ | – | $14.1 \pm 3.7$ | – | $16.7 \pm 2.8$ | – | $16.9 \pm 3.0$ | – | 16.2 | – |
| DiffPuter | $11.7 \pm 4.5$ | – | $14.2 \pm 4.2$ | – | $13.1 \pm 3.8$ | – | $12.6 \pm 3.9$ | – | $17.6 \pm 4.2$ | – | 13.8 | – |
| ForestDiff | $10.9 \pm 4.6$ | – | $6.3 \pm 4.3$ | – | $9.5 \pm 3.5$ | – | $11.6 \pm 3.2$ | – | $7.8 \pm 3.4$ | – | 9.2 | – |
| **(B) Missing-unaware baselines** | | | | | | | | | | | | |
| TVAE | $17.8 \pm 3.4$ | – | $18.4 \pm 2.2$ | – | $17.6 \pm 2.0$ | – | $18.7 \pm 2.2$ | – | $15.8 \pm 2.9$ | – | 17.7 | – |
| ∟ +AugFull | $11.6 \pm 5.0$ | (+6.3) | $11.8 \pm 4.5$ | (+6.6) | $10.2 \pm 3.1$ | (+7.3) | $10.3 \pm 2.1$ | (+8.4) | $10.5 \pm 3.6$ | (+5.3) | 10.9 | (+6.8) |
| CTGAN | $13.5 \pm 3.7$ | – | $16.8 \pm 3.1$ | – | $16.1 \pm 2.7$ | – | $16.8 \pm 2.4$ | – | $17.9 \pm 3.1$ | – | 16.2 | – |
| ∟ +AugFull | $11.7 \pm 5.1$ | (+1.9) | $13.3 \pm 5.0$ | (+3.5) | $11.7 \pm 3.1$ | (+4.4) | $10.5 \pm 3.6$ | (+6.3) | $14.7 \pm 4.1$ | (+3.2) | 12.4 | (+3.9) |
| ARF | $17.0 \pm 2.5$ | – | $18.4 \pm 2.0$ | – | $20.5 \pm 0.7$ | – | $20.2 \pm 0.8$ | – | $17.0 \pm 2.5$ | – | 18.6 | – |
| ∟ +AugFull | $8.9 \pm 5.9$ | (+8.1) | $11.8 \pm 5.1$ | (+6.5) | $18.2 \pm 3.0$ | (+2.3) | $11.2 \pm 5.7$ | (+9.0) | $11.2 \pm 5.7$ | (+5.8) | 12.3 | (+6.3) |
| TabSyn | $14.1 \pm 4.0$ | – | $13.5 \pm 2.2$ | – | $12.5 \pm 3.4$ | – | $13.8 \pm 3.2$ | – | $12.6 \pm 3.6$ | – | 13.3 | – |
| ∟ +AugFull | $6.1 \pm 2.2^*$ | (+8.0) | $7.3 \pm 1.9$ | (+6.2) | $\mathbf{3.6 \pm 1.9^{**}}$ | (+8.9) | $\mathbf{4.1 \pm 2.5^{**}}$ | (+9.7) | $6.5 \pm 3.2^*$ | (+6.2) | $\mathbf{5.5^{**}}$ | (+7.8) |
| TabDDPM | $15.3 \pm 3.2$ | – | $12.5 \pm 2.4$ | – | $15.2 \pm 2.3$ | – | $13.6 \pm 1.9$ | – | $12.8 \pm 2.7$ | – | 13.9 | – |
| ∟ +AugFull | $8.2 \pm 5.1$ | (+7.1) | $6.5 \pm 3.6$ | (+6.0) | $9.0 \pm 5.2^*$ | (+6.2) | $8.0 \pm 5.0^*$ | (+5.6) | $6.3 \pm 4.2^*$ | (+6.5) | $7.6^*$ | (+6.3) |
| ∟ +AugMask | $8.1 \pm 5.6$ | (+7.1) | $5.7 \pm 4.1^*$ | (+6.9) | $8.9 \pm 4.9^*$ | (+6.3) | $6.9 \pm 5.0^*$ | (+6.7) | $\mathbf{5.2 \pm 5.1^{**}}$ | (+7.6) | $6.9^*$ | (+6.9) |
| CDTD | $13.1 \pm 3.6$ | – | $14.3 \pm 3.5$ | – | $13.0 \pm 3.0$ | – | $15.4 \pm 2.9$ | – | $12.3 \pm 4.1$ | – | 13.6 | – |
| ∟ +AugFull | $4.8 \pm 2.7^*$ | (+8.3) | $6.6 \pm 2.3$ | (+7.7) | $\mathbf{4.4 \pm 2.2^{**}}$ | (+8.6) | $\mathbf{5.1 \pm 2.3^{**}}$ | (+10.2) | $6.0 \pm 3.0^*$ | (+6.3) | $\mathbf{5.4^{**}}$ | (+8.2) |
| ∟ +AugMask | $\mathbf{3.8 \pm 2.5^{**}}$ | (+9.3) | $\mathbf{4.4 \pm 3.2^{**}}$ | (+9.9) | $\mathbf{3.5 \pm 2.0^{**}}$ | (+9.5) | $\mathbf{3.9 \pm 2.3^{**}}$ | (+11.5) | $\mathbf{5.3 \pm 3.0^{**}}$ | (+7.0) | $\mathbf{4.2^{**}}$ | (+9.4) |
| TabDiff | $16.1 \pm 3.4$ | – | $11.1 \pm 3.3$ | – | $12.3 \pm 2.8$ | – | $13.2 \pm 3.1$ | – | $11.7 \pm 3.3$ | – | 12.9 | – |
| ∟ +AugFull | $6.6 \pm 3.7^*$ | (+9.5) | $\mathbf{4.4 \pm 3.7^{**}}$ | (+6.7) | $\mathbf{3.7 \pm 2.0^{**}}$ | (+8.6) | $\mathbf{4.5 \pm 2.5^{**}}$ | (+8.7) | $\mathbf{5.0 \pm 2.8^{**}}$ | (+6.7) | $\mathbf{4.8^{**}}$ | (+8.0) |
| ∟ +AugMask | $\mathbf{4.7 \pm 4.6^{**}}$ | (+11.4) | $\mathbf{3.6 \pm 4.5^{**}}$ | (+7.5) | $\mathbf{2.9 \pm 2.5^{**}}$ | (+9.4) | $\mathbf{3.2 \pm 2.9^{**}}$ | (+10.0) | $\mathbf{2.9 \pm 2.2^{**}}$ | (+8.8) | $\mathbf{3.5^{**}}$ | (+9.4) |

## D.2. Detailed Dataset-Specific Performance Breakdown

In this section, we provide a dataset-wise breakdown of the performance metrics summarized in the main paper. Each table reports the **Mean Performance Score** (↑) $\times 100 \pm$ **Standard Deviation** aggregated across all 90 experimental scenarios (6 datasets × 5 noise ratios × 3 seeds). To quantify the impact of our proposed strategies, we report the **Relative Improvement** (Δ%) for each variant, calculated as the percentage gain over its respective base (+NoAug) model. Statistical significance is assessed following the methodology in the main text. We conduct a paired **Wilcoxon signed-rank test** on the raw metric scores against the best-performing missing-aware baseline within each experimental block. A marker of ** denotes a statistically significant gain ($p < 0.05$), while * indicates a statistical match. **Boldface** is applied to all scores and relative gains that achieve a statistical gain (**) over the baseline group. The **Summary** column represents the global arithmetic mean across all datasets, serving as a measure of the overall robustness of each training strategy.

*Table 8.* α-**Precision** (MCAR)

| Method | Adult Score | Δ(%) | Default Score | Δ(%) | Magic Score | Δ(%) | Shoppers Score | Δ(%) | Beijing Score | Δ(%) | News Score | Δ(%) | Summary Score | Δ(%) |
|---|---|---|---|---|---|---|---|---|---|---|---|---|---|---|
| **(A) Missing-aware baselines** | | | | | | | | | | | | | | |
| MIWAE | 86.1 ± 9.40 | – | 93.7 ± 4.04 | – | 97.4 ± 0.92 | – | 93.7 ± 3.59 | – | 93.4 ± 3.29 | – | 91.6 ± 5.16 | – | 92.7 | – |
| MissDiff | 88.7 ± 8.79 | – | 47.5 ± 1.27 | – | 13.2 ± 2.99 | – | 42.9 ± 11.1 | – | 90.4 ± 5.22 | – | 1.63 ± 1.16 | – | 47.4 | – |
| DiffPuter | 55.7 ± 31.3 | – | 62.9 ± 19.4 | – | 83.2 ± 9.49 | – | 66.6 ± 25.8 | – | 88.1 ± 9.12 | – | 79.1 ± 18.3 | – | 72.6 | – |
| ForestDiff | 69.4 ± 37.9 | – | 76.2 ± 24.1 | – | 89.4 ± 6.26 | – | 63.1 ± 41.1 | – | 91.9 ± 5.87 | – | 62.5 ± 19.9 | – | 75.4 | – |
| **(B) Missing-unaware baselines & enhancements via our proposed training strategy** | | | | | | | | | | | | | | |
| TVAE | 44.1 ± 21.3 | – | 41.9 ± 24.7 | – | 70.3 ± 13.1 | – | 30.5 ± 18.9 | – | 70.4 ± 10.7 | – | 49.9 ± 21.8 | – | 51.2 | – |
| ∟ +AugFull | 71.8 ± 10.9 | (+62.9%) | 83.1 ± 8.73 | (+98.3%) | 89.7 ± 3.32 | (+27.6%) | 41.0 ± 18.4 | (+34.3%) | 83.4 ± 9.30 | (+18.6%) | 92.5* ± 4.93 | (+85.2%) | 77.1 | (+50.5%) |
| CTGAN | 71.9 ± 19.6 | – | 68.0 ± 11.1 | – | 78.7 ± 5.75 | – | 76.7 ± 19.4 | – | 78.9 ± 11.0 | – | 71.1 ± 23.6 | – | 73.7 | – |
| ∟ +AugFull | 72.4 ± 9.86 | (+0.66%) | 71.8 ± 4.68 | (+5.56%) | 80.3 ± 4.58 | (+1.99%) | 81.4 ± 6.55 | (+6.23%) | 85.9 ± 6.83 | (+8.95%) | 88.8 ± 5.05 | (+24.8%) | 80.1 | (+8.66%) |
| ARF | 56.9 ± 24.7 | – | 60.8 ± 23.7 | – | 53.0 ± 22.2 | – | 46.2 ± 25.9 | – | 52.2 ± 27.8 | – | 41.5 ± 26.6 | – | 53.3 | – |
| ∟ +AugFull | **94.9**\*\* ± **9.09** (+66.7%) | | 88.0 ± 5.19 | (+44.7%) | 68.4 ± 16.0 | (+29.0%) | 91.0* ± 8.70 | (+96.8%) | **98.6**\*\* ± **1.07** (+88.9%) | | 17.2 ± 13.0 | – | 84.6 | (+58.8%) |
| TabSyn | 58.9 ± 24.2 | – | 62.4 ± 26.3 | – | 77.0 ± 13.1 | – | 63.5 ± 27.2 | – | 75.1 ± 13.2 | – | 71.0 ± 17.9 | – | 68.1 | – |
| ∟ +AugFull | 96.1* ± 2.39 | (+63.1%) | 91.3 ± 6.82 | (+46.3%) | 96.0* ± 3.60 | (+24.6%) | 93.9* ± 6.66 | (+47.9%) | **97.5**\*\* ± **1.05** (+29.8%) | | 68.0 ± 12.1 | – | 90.5 | (+32.7%) |
| TabDDPM | 50.2 ± 28.7 | – | 74.5 ± 16.4 | – | 58.3 ± 26.9 | – | 54.4 ± 27.6 | – | 70.6 ± 18.0 | – | 21.2 ± 10.1 | – | 54.9 | – |
| ∟ +AugFull | **97.6**\*\* ± **1.77** (+94.6%) | | 83.4 ± 7.18 | (+11.8%) | 96.5* ± 3.12 | (+65.6%) | **97.0**\*\* ± **1.89** (+78.2%) | | **97.8**\*\* ± **1.16** (+38.5%) | | 16.4 ± 17.1 | – | 81.4 | (+48.4%) |
| ∟ +AugMask | **97.6**\*\* ± **1.16** (+94.6%) | | 90.0 ± 3.84 | (+20.7%) | 96.1 ± 1.67 | (+64.9%) | 94.4* ± 3.98 | (+73.6%) | **97.4**\*\* ± **1.88** (+37.9%) | | 2.83 ± 1.72 | – | 79.7 | (+45.3%) |
| CDTD | 52.5 ± 27.5 | – | 62.6 ± 26.1 | – | 78.3 ± 13.1 | – | 65.0 ± 27.1 | – | 80.8 ± 11.9 | – | 64.2 ± 18.9 | – | 67.2 | – |
| ∟ +AugFull | **96.9**\*\* ± **2.35** (+84.6%) | | 93.5* ± 6.64 | (+49.3%) | 95.6* ± 4.12 | (+22.0%) | 92.8* ± 3.18 | (+42.9%) | **97.3**\*\* ± **1.68** (+20.5%) | | 85.6 ± 16.4 | (+33.2%) | 93.6* | (+39.2%) |
| ∟ +AugMask | **97.9**\*\* ± **2.21** (+86.7%) | | **97.6**\*\* ± **1.51** (+55.9%) | | **98.8**\*\* ± **0.56** (+26.1%) | | 91.5* ± 6.36 | (+40.8%) | 96.1* ± 2.98 | (+19.0%) | 94.5* ± 5.90 | (+47.1%) | **96.1**\*\* | (+42.9%) |
| TabDiff | 50.5 ± 29.1 | – | 59.0 ± 25.3 | – | 58.5 ± 13.2 | – | 47.6 ± 27.5 | – | 69.0 ± 18.4 | – | 46.2 ± 8.87 | – | 55.1 | – |
| ∟ +AugFull | **97.2**\*\* ± **2.10** (+92.5%) | | 93.3* ± 6.30 | (+58.2%) | 96.3* ± 3.16 | (+64.6%) | **97.3**\*\* ± **1.53** (+104%) | | **97.8**\*\* ± **1.35** (+41.8%) | | 32.8 ± 4.89 | – | 85.8* | (+55.6%) |
| ∟ +AugMask | **98.3**\*\* ± **0.76** (+94.6%) | | **97.8**\*\* ± **0.75** (+65.7%) | | **98.7**\*\* ± **1.24** (+68.7%) | | 96.3* ± 2.41 | (+102%) | **98.5**\*\* ± **0.79** (+42.8%) | | 29.0 ± 1.49 | – | **86.4**\*\* | (+56.8%) |

*Table 9.* β-**Recall** (MCAR)

| Method | Adult Score | Δ(%) | Default Score | Δ(%) | Magic Score | Δ(%) | Shoppers Score | Δ(%) | Beijing Score | Δ(%) | News Score | Δ(%) | Summary Score | Δ(%) |
|---|---|---|---|---|---|---|---|---|---|---|---|---|---|---|
| **(A) Missing-aware baselines** | | | | | | | | | | | | | | |
| MIWAE | 6.64 ± 0.97 | – | 18.8 ± 4.92 | – | 26.7 ± 9.30 | – | 19.2 ± 2.01 | – | 44.5 ± 1.08 | – | 10.2 ± 3.06 | – | 21.0 | – |
| MissDiff | 13.7 ± 10.0 | – | 0.00 ± 0.00 | – | 0.00 ± 0.01 | – | 0.04 ± 0.02 | – | 45.8 ± 2.69 | – | 0.00 ± 0.00 | – | 9.93 | – |
| DiffPuter | 6.36 ± 2.81 | – | 5.86 ± 3.38 | – | 23.9 ± 15.7 | – | 11.7 ± 6.65 | – | 41.1 ± 4.53 | – | 10.9 ± 11.0 | – | 16.6 | – |
| ForestDiff | 29.1 ± 17.3 | – | 31.6 ± 16.8 | – | 40.4 ± 14.2 | – | 29.2 ± 20.7 | – | 45.6 ± 11.2 | – | 26.8 ± 19.7 | – | 33.8 | – |
| **(B) Missing-unaware baselines & enhancements via our proposed training strategy** | | | | | | | | | | | | | | |
| TVAE | 9.63 ± 8.98 | – | 4.68 ± 4.54 | – | 4.87 ± 8.06 | – | 9.93 ± 12.3 | – | 2.92 ± 5.32 | – | 3.09 ± 5.64 | – | 5.74 | – |
| ∟ +AugFull | 20.6 ± 6.08 | (+114%) | 15.0 ± 4.01 | (+221%) | 18.0 ± 8.63 | (+270%) | 16.8 ± 10.0 | (+69.4%) | 8.83 ± 8.32 | (+202%) | 11.2 ± 6.46 | (+262%) | 14.9 | (+159%) |
| CTGAN | 5.91 ± 2.38 | – | 4.00 ± 1.79 | – | 2.88 ± 3.11 | – | 11.5 ± 6.04 | – | 4.73 ± 1.52 | – | 3.91 ± 6.58 | – | 5.77 | – |
| ∟ +AugFull | 7.84 ± 2.27 | (+32.6%) | 6.91 ± 4.09 | (+73.0%) | 7.97 ± 3.49 | (+176%) | 16.2 ± 4.67 | (+41.4%) | 4.40 ± 0.61 | – | 14.9 ± 9.05 | (+280%) | 9.70 | (+68.0%) |
| ARF | 3.11 ± 2.50 | – | 3.60 ± 2.08 | – | 0.32 ± 0.68 | – | 10.7 ± 7.61 | – | 20.0 ± 17.0 | – | 0.09 ± 0.19 | – | 6.49 | – |
| ∟ +AugFull | 20.0 ± 8.74 | (+543%) | 15.7 ± 7.70 | (+335%) | 0.25 ± 0.63 | – | 13.9 ± 6.84 | (+30.5%) | 48.7 ± 2.58 | (+143%) | 0.11 ± 0.07 | (+15.8%) | 18.7 | (+189%) |
| TabSyn | 18.9 ± 11.8 | – | 12.2 ± 9.35 | – | 11.6 ± 13.9 | – | 19.0 ± 13.5 | – | 22.0 ± 17.1 | – | 2.71 ± 4.36 | – | 14.3 | – |
| ∟ +AugFull | 26.2 ± 11.5 | (+39.1%) | 21.7 ± 10.0 | (+77.4%) | 21.8 ± 14.2 | (+88.6%) | 30.7 ± 7.80 | (+61.5%) | 48.1 ± 1.52 | (+118%) | 9.38 ± 7.61 | (+246%) | 26.3 | (+84.1%) |
| TabDDPM | 19.0 ± 14.4 | – | 15.5 ± 8.77 | – | 16.2 ± 12.9 | – | 25.6 ± 13.6 | – | 21.5 ± 16.8 | – | 1.93 ± 1.19 | – | 16.6 | – |
| ∟ +AugFull | **32.3**\*\* ± **12.3** (+70.3%) | | 20.6 ± 7.54 | (+32.6%) | 22.6 ± 14.0 | (+38.9%) | 34.0* ± 7.99 | (+32.6%) | 48.1 ± 1.74 | (+123%) | 1.33 ± 1.11 | – | 26.5 | (+59.2%) |
| ∟ +AugMask | **40.8**\*\* ± **6.06** (+115%) | | 32.5* ± 4.27 | (+109%) | 38.6* ± 6.28 | (+137%) | 40.1* ± 5.32 | (+56.8%) | 46.7 ± 5.16 | (+117%) | 0.73 ± 0.37 | – | 33.2* | (+99.9%) |
| CDTD | 21.2 ± 13.7 | – | 11.0 ± 8.03 | – | 9.00 ± 13.2 | – | 14.5 ± 10.1 | – | 18.7 ± 15.8 | – | 5.04 ± 9.64 | – | 13.2 | – |
| ∟ +AugFull | 31.9* ± 14.5 | (+50.1%) | 22.8 ± 11.4 | (+107%) | 23.2 ± 14.9 | (+158%) | 29.8* ± 11.7 | (+106%) | 46.3 ± 4.81 | (+148%) | 21.3 ± 14.2 | (+322%) | 29.2 | (+121%) |
| ∟ +AugMask | **38.1**\*\* ± **9.66** (+79.3%) | | 27.7 ± 10.1 | (+152%) | 39.3* ± 6.83 | (+337%) | 37.5* ± 13.2 | (+159%) | 40.7 ± 12.0 | (+118%) | 34.7* ± 11.2 | (+589%) | 36.3* | (+174%) |
| TabDiff | 19.4 ± 15.0 | – | 19.4 ± 12.8 | – | 18.0 ± 14.8 | – | 29.3 ± 16.4 | – | 22.4 ± 17.6 | – | 0.64 ± 0.14 | – | 18.2 | – |
| ∟ +AugFull | **32.9**\*\* ± **13.1** (+69.8%) | | 25.5 ± 12.9 | (+31.3%) | 24.6 ± 15.9 | (+36.8%) | **37.9**\*\* ± **10.0** (+29.4%) | | 49.3* ± 2.64 | (+120%) | 0.62 ± 0.05 | – | 28.5 | (+56.5%) |
| ∟ +AugMask | **41.7**\*\* ± **7.25** (+115%) | | **41.8**\*\* ± **5.54** (+115%) | | 42.4* ± 7.14 | (+136%) | **45.8**\*\* ± **5.92** (+56.3%) | | 50.5* ± 2.57 | (+125%) | 0.62 ± 0.06 | – | **37.2**\*\* | (+104%) |

*Table 10.* **Trend** (MCAR)

| Method | Adult Score | Δ(%) | Default Score | Δ(%) | Magic Score | Δ(%) | Shoppers Score | Δ(%) | Beijing Score | Δ(%) | News Score | Δ(%) | Summary Score | Δ(%) |
|---|---|---|---|---|---|---|---|---|---|---|---|---|---|---|
| **(A) Missing-aware baselines** | | | | | | | | | | | | | | |
| MIWAE | 71.5 ± 3.46 | – | 84.6 ± 3.21 | – | 96.2 ± 1.37 | – | 82.9 ± 0.86 | – | 81.8 ± 3.89 | – | 95.8 ± 0.63 | – | 85.5 | – |
| MissDiff | 75.9 ± 15.3 | – | 77.3 ± 12.9 | – | 81.3 ± 3.64 | – | 66.4 ± 15.9 | – | 85.2 ± 13.6 | – | 91.2 ± 1.16 | – | 79.6 | – |
| DiffPuter | 60.3 ± 14.7 | – | 68.3 ± 4.69 | – | 90.5 ± 5.42 | – | 73.5 ± 9.53 | – | 79.1 ± 9.32 | – | 94.2 ± 2.80 | – | 77.7 | – |
| ForestDiff | 78.2 ± 23.3 | – | 80.1 ± 14.6 | – | 94.2 ± 4.80 | – | 78.4 ± 20.5 | – | 88.8 ± 9.73 | – | 93.7 ± 4.11 | – | 85.6 | – |
| **(B) Missing-unaware baselines & enhancements via our proposed training strategy** | | | | | | | | | | | | | | |
| TVAE | 67.9 ± 12.2 | – | 63.2 ± 8.97 | – | 82.5 ± 7.16 | – | 69.1 ± 9.95 | – | 47.0 ± 18.9 | – | 70.8 ± 33.0 | – | 68.1 | – |
| ∟ +AugFull | 82.8 ± 5.54 | (+21.9%) | 81.7 ± 2.92 | (+29.2%) | 92.9 ± 2.27 | (+12.5%) | 77.9 ± 8.79 | (+12.7%) | 70.7 ± 19.1 | (+50.5%) | 94.7 ± 1.03 | (+33.8%) | 83.5 | (+22.6%) |
| CTGAN | 60.3 ± 10.4 | – | 65.9 ± 5.57 | – | 82.1 ± 5.71 | – | 76.4 ± 7.20 | – | 53.8 ± 15.1 | – | 92.5 ± 1.83 | – | 70.7 | – |
| ∟ +AugFull | 70.4 ± 4.82 | (+16.7%) | 73.6 ± 2.35 | (+11.8%) | 88.2 ± 2.95 | (+7.44%) | 84.1 ± 1.78 | (+10.1%) | 66.1 ± 8.64 | (+22.9%) | 94.7 ± 0.61 | (+2.45%) | 79.5 | (+12.5%) |
| ARF | 38.4 ± 12.2 | – | 28.0 ± 8.37 | – | 53.8 ± 20.2 | – | 27.1 ± 8.78 | – | 29.4 ± 14.4 | – | 34.9 ± 9.16 | – | 35.6 | – |
| ∟ +AugFull | 61.2 ± 1.67 | (+59.6%) | 54.0 ± 2.90 | (+93.2%) | 61.9 ± 11.2 | (+15.1%) | 34.9 ± 6.91 | (+28.7%) | 63.1 ± 2.52 | (+115%) | 35.3 ± 10.7 | (+1.40%) | 54.0 | (+51.7%) |
| TabSyn | 71.6 ± 17.2 | – | 71.2 ± 12.2 | – | 86.3 ± 7.78 | – | 76.6 ± 12.2 | – | 65.9 ± 18.6 | – | 92.5 ± 2.62 | – | 77.4 | – |
| ∟ +AugFull | 91.8** ± 3.33 | (+28.3%) | 86.6* ± 3.53 | (+21.7%) | 95.1 ± 3.48 | (+10.3%) | 91.2* ± 3.02 | (+19.0%) | 94.3** ± 2.26 | (+43.1%) | 96.1 ± 1.05 | (+3.87%) | 92.5** | (+19.5%) |
| TabDDPM | 71.0 ± 16.6 | – | 71.3 ± 10.5 | – | 83.6 ± 7.45 | – | 76.2 ± 11.5 | – | 63.9 ± 18.5 | – | 86.9 ± 1.15 | – | 75.5 | – |
| ∟ +AugFull | 92.3** ± 3.93 | (+30.0%) | 78.4 ± 2.75 | (+10.1%) | 93.2 ± 2.34 | (+11.4%) | 93.0** ± 2.23 | (+22.0%) | 93.4** ± 2.25 | (+46.1%) | 85.7 ± 1.88 | – | 89.3 | (+18.3%) |
| ∟ +AugMask | 94.4** ± 2.66 | (+33.0%) | 81.2 ± 4.19 | (+13.9%) | 93.6 ± 1.45 | (+12.0%) | 91.0* ± 2.72 | (+19.4%) | 93.6** ± 3.04 | (+46.5%) | 84.8 ± 2.47 | – | 89.8 | (+18.9%) |
| CDTD | 71.0 ± 17.2 | – | 73.5 ± 11.6 | – | 85.4 ± 6.74 | – | 75.6 ± 11.4 | – | 62.9 ± 18.4 | – | 92.9 ± 2.89 | – | 76.9 | – |
| ∟ +AugFull | 92.8** ± 3.98 | (+30.7%) | 87.8* ± 5.49 | (+19.5%) | 95.5* ± 3.80 | (+11.8%) | 91.4* ± 3.00 | (+20.9%) | 88.1* ± 11.5 | (+40.0%) | 96.7** ± 1.19 | (+4.03%) | 92.0** | (+19.7%) |
| ∟ +AugMask | 94.6** ± 3.34 | (+33.3%) | 89.5** ± 4.80 | (+21.7%) | 97.0** ± 1.87 | (+13.6%) | 93.5** ± 3.23 | (+23.7%) | 92.6* ± 2.15 | (+47.2%) | 97.4** ± 0.95 | (+4.83%) | 94.1** | (+22.4%) |
| TabDiff | 71.9 ± 16.6 | – | 73.9 ± 13.8 | – | 78.8 ± 11.6 | – | 78.8 ± 11.8 | – | 65.4 ± 18.3 | – | 92.4 ± 2.15 | – | 78.3 | – |
| ∟ +AugFull | 92.8** ± 4.81 | (+29.1%) | 89.6** ± 5.48 | (+21.2%) | 95.8* ± 3.62 | (+9.39%) | 94.2** ± 2.77 | (+19.6%) | 94.9** ± 2.63 | (+45.2%) | 95.4 ± 0.61 | (+3.27%) | 93.8** | (+19.7%) |
| ∟ +AugMask | 96.0** ± 3.00 | (+33.6%) | 90.9** ± 4.47 | (+23.0%) | 96.7** ± 1.81 | (+10.4%) | 95.8** ± 2.33 | (+21.6%) | 96.2** ± 1.74 | (+47.2%) | 95.5 ± 0.46 | (+3.44%) | 95.2** | (+21.6%) |

*Table 11.* **Shape** (MCAR)

| Method | Adult Score | Δ(%) | Default Score | Δ(%) | Magic Score | Δ(%) | Shoppers Score | Δ(%) | Beijing Score | Δ(%) | News Score | Δ(%) | Summary Score | Δ(%) |
|---|---|---|---|---|---|---|---|---|---|---|---|---|---|---|
| **(A) Missing-aware baselines** | | | | | | | | | | | | | | |
| MIWAE | 87.0 ± 0.95 | – | 93.9 ± 0.87 | – | 97.4 ± 0.30 | – | 85.1 ± 1.48 | – | 92.9 ± 0.62 | – | 94.4 ± 0.42 | – | 91.8 | – |
| MissDiff | 81.1 ± 4.94 | – | 78.4 ± 5.80 | – | 69.1 ± 2.97 | – | 66.2 ± 3.36 | – | 77.0 ± 1.74 | – | 45.9 ± 3.13 | – | 69.6 | – |
| DiffPuter | 81.7 ± 8.11 | – | 74.8 ± 6.82 | – | 84.0 ± 14.8 | – | 75.6 ± 8.78 | – | 88.1 ± 8.63 | – | 73.6 ± 23.1 | – | 79.6 | – |
| ForestDiff | 84.9 ± 18.5 | – | 83.2 ± 15.8 | – | 92.0 ± 6.02 | – | 81.1 ± 18.7 | – | 90.5 ± 8.66 | – | 80.9 ± 14.1 | – | 85.4 | – |
| **(B) Missing-unaware baselines & enhancements via our proposed training strategy** | | | | | | | | | | | | | | |
| TVAE | 78.4 ± 9.89 | – | 57.0 ± 17.5 | – | 53.6 ± 23.0 | – | 71.9 ± 11.8 | – | 54.4 ± 18.8 | – | 47.2 ± 22.6 | – | 59.5 | – |
| ∟ +AugFull | 91.3 ± 1.67 | (+16.4%) | 86.2 ± 4.70 | (+51.2%) | 94.3 ± 0.88 | (+76.0%) | 83.6 ± 8.22 | (+16.3%) | 85.9 ± 9.06 | (+57.8%) | 82.9 ± 5.08 | (+75.6%) | 87.2 | (+46.5%) |
| CTGAN | 78.6 ± 6.75 | – | 62.8 ± 14.7 | – | 66.3 ± 22.7 | – | 75.8 ± 10.0 | – | 65.1 ± 13.8 | – | 64.3 ± 19.1 | – | 69.1 | – |
| ∟ +AugFull | 86.7 ± 1.54 | (+10.2%) | 83.7 ± 2.70 | (+33.3%) | 90.0 ± 1.73 | (+35.7%) | 87.8 ± 2.48 | (+15.8%) | 85.4 ± 1.73 | (+31.1%) | 90.6 ± 2.11 | (+41.0%) | 87.4 | (+26.4%) |
| ARF | 65.3 ± 9.83 | – | 52.5 ± 10.0 | – | 37.0 ± 11.0 | – | 59.4 ± 8.23 | – | 54.6 ± 13.8 | – | 36.9 ± 5.67 | – | 52.8 | – |
| ∟ +AugFull | 94.6* ± 1.01 | (+44.8%) | 92.5 ± 2.04 | (+76.2%) | 42.6 ± 7.51 | (+15.1%) | 67.6 ± 6.29 | (+13.9%) | 96.5** ± 0.34 | (+76.8%) | 38.2 ± 3.07 | (+3.67%) | 76.7 | (+45.3%) |
| TabSyn | 81.0 ± 12.2 | – | 68.7 ± 19.7 | – | 67.7 ± 22.8 | – | 75.4 ± 13.5 | – | 68.2 ± 20.6 | – | 71.1 ± 17.4 | – | 71.9 | – |
| ∟ +AugFull | 97.4** ± 0.66 | (+20.2%) | 92.7 ± 3.37 | (+34.9%) | 97.8** ± 0.68 | (+44.4%) | 90.5 ± 3.38 | (+20.0%) | 98.1** ± 0.41 | (+43.8%) | 91.3 ± 4.15 | (+28.4%) | 94.6* | (+31.6%) |
| TabDDPM | 81.4 ± 10.9 | – | 80.8 ± 10.9 | – | 78.8 ± 14.3 | – | 79.4 ± 10.3 | – | 73.0 ± 16.2 | – | 49.4 ± 5.79 | – | 73.8 | – |
| ∟ +AugFull | 96.7** ± 1.59 | (+18.8%) | 88.6 ± 4.10 | (+9.65%) | 98.1** ± 0.51 | (+24.4%) | 92.5* ± 2.47 | (+16.4%) | 97.1** ± 0.38 | (+33.0%) | 49.4 ± 13.5 | – | 87.1* | (+18.0%) |
| ∟ +AugMask | 97.9** ± 1.04 | (+20.2%) | 92.3 ± 2.07 | (+14.2%) | 96.9* ± 1.28 | (+22.9%) | 93.8** ± 2.49 | (+18.1%) | 96.3** ± 1.32 | (+32.0%) | 41.9 ± 4.34 | – | 86.5* | (+17.2%) |
| CDTD | 81.8 ± 10.8 | – | 66.8 ± 17.6 | – | 66.5 ± 19.2 | – | 72.7 ± 12.0 | – | 68.5 ± 16.6 | – | 63.4 ± 18.6 | – | 70.0 | – |
| ∟ +AugFull | 97.1** ± 1.00 | (+18.7%) | 94.5* ± 2.55 | (+41.4%) | 98.1** ± 0.49 | (+47.6%) | 91.4* ± 2.86 | (+25.8%) | 95.3* ± 1.92 | (+39.1%) | 92.0 ± 2.47 | (+45.0%) | 94.7** | (+35.4%) |
| ∟ +AugMask | 98.1** ± 0.84 | (+19.9%) | 96.4** ± 2.19 | (+44.3%) | 98.0** ± 1.07 | (+47.4%) | 94.7** ± 2.86 | (+30.3%) | 95.4* ± 0.75 | (+39.2%) | 94.9* ± 0.77 | (+49.6%) | 96.2** | (+37.6%) |
| TabDiff | 81.2 ± 11.2 | – | 74.6 ± 15.3 | – | 78.3 ± 12.6 | – | 79.7 ± 11.3 | – | 71.8 ± 16.6 | – | 86.4 ± 1.50 | – | 78.7 | – |
| ∟ +AugFull | 96.9** ± 1.71 | (+19.4%) | 95.1* ± 2.41 | (+27.5%) | 98.3** ± 0.46 | (+25.5%) | 93.4** ± 3.41 | (+17.1%) | 97.5** ± 0.50 | (+35.8%) | 88.2 ± 0.40 | (+2.15%) | 94.9** | (+20.6%) |
| ∟ +AugMask | 98.6** ± 0.83 | (+21.4%) | 97.5** ± 1.05 | (+30.7%) | 98.1** ± 0.89 | (+25.3%) | 97.1** ± 1.10 | (+21.8%) | 97.4** ± 0.66 | (+35.7%) | 88.0 ± 0.51 | (+1.90%) | 96.1** | (+22.2%) |

*Table 12.* **ML-Task Scores**(MCAR)

| Method | Adult Score | Δ(%) | Default Score | Δ(%) | Magic Score | Δ(%) | Shoppers Score | Δ(%) | Beijing Score | Δ(%) | News Score | Δ(%) | Summary Score | Δ(%) |
|---|---|---|---|---|---|---|---|---|---|---|---|---|---|---|
| **(A) Missing-aware baselines** | | | | | | | | | | | | | | |
| MIWAE | 90.3 ± 2.27 | – | 94.6 ± 1.64 | – | 94.2 ± 4.03 | – | 85.4 ± 6.68 | – | 47.9 ± 3.44 | – | 95.4 ± 2.73 | – | 84.6 | – |
| MissDiff | 73.8 ± 16.8 | – | 83.1 ± 12.1 | – | 75.8 ± 17.2 | – | 57.2 ± 15.6 | – | 44.7 ± 9.12 | – | 40.9 ± 23.8 | – | 62.6 | – |
| DiffPuter | 46.5 ± 13.6 | – | 65.4 ± 7.34 | – | 47.8 ± 7.42 | – | 56.1 ± 5.55 | – | 43.2 ± 8.27 | – | 83.8 ± 23.8 | – | 57.2 | – |
| ForestDiff | 93.1* ± 7.67 | – | 92.5 ± 10.00 | – | 91.9* ± 10.0 | – | 91.5* ± 10.0 | – | 61.8* ± 9.88 | – | 95.6 ± 2.04 | – | 87.7 | – |
| **(B) Missing-unaware baselines & enhancements via our proposed training strategy** | | | | | | | | | | | | | | |
| TVAE | 90.2 ± 3.55 | – | 84.2 ± 10.5 | – | 77.1 ± 12.8 | – | 84.1 ± 12.9 | – | 35.1 ± 12.2 | – | 89.3 ± 5.68 | – | 78.1 | – |
| ∟ +AugFull | 94.5 ± 1.05 | (+4.78%) | 92.9 ± 3.22 | (+10.3%) | 86.7 ± 9.21 | (+12.5%) | 90.4 ± 8.94 | (+7.44%) | 48.8 ± 5.23 | (+39.0%) | 95.1 ± 1.16 | (+6.51%) | 84.4 | (+8.03%) |
| CTGAN | 67.9 ± 15.2 | – | 71.5 ± 7.40 | – | 70.5 ± 14.6 | – | 70.1 ± 12.9 | – | 32.4 ± 3.83 | – | 83.7 ± 10.6 | – | 65.5 | – |
| ∟ +AugFull | 83.6 ± 11.1 | (+23.2%) | 78.1 ± 9.23 | (+9.26%) | 77.1 ± 10.9 | (+9.27%) | 77.7 ± 12.9 | (+10.9%) | 39.1 ± 2.37 | (+20.8%) | 94.6 ± 2.36 | (+13.0%) | 75.0 | (+14.5%) |
| ARF | 91.7 ± 2.64 | – | 87.2 ± 6.10 | – | 56.0 ± 8.24 | – | 60.6 ± 8.36 | – | 29.9 ± 14.1 | – | 72.2 ± 26.5 | – | 65.3 | – |
| ∟ +AugFull | 95.6** ± 3.19 | (+4.27%) | 93.2 ± 7.52 | (+6.77%) | 55.3 ± 6.74 | – | 64.1 ± 9.57 | (+5.82%) | 54.0 ± 7.62 | (+80.9%) | 82.8 ± 8.59 | (+14.6%) | 75.6 | (+15.8%) |
| TabSyn | 92.3* ± 7.15 | – | 87.3 ± 12.9 | – | 78.2 ± 15.0 | – | 77.8 ± 20.3 | – | 44.8 ± 12.1 | – | 89.8 ± 6.84 | – | 78.2 | – |
| ∟ +AugFull | 96.2** ± 1.80 | (+4.29%) | 95.9* ± 5.02 | (+9.88%) | 92.7 ± 6.29 | (+18.5%) | 92.8* ± 5.34 | (+19.3%) | 54.6 ± 7.85 | (+21.9%) | 94.4* ± 4.92 | (+5.23%) | 87.8 | (+12.3%) |
| TabDDPM | 93.8* ± 5.21 | – | 91.3 ± 9.56 | – | 82.9 ± 12.8 | – | 85.7 ± 13.0 | – | 50.6 ± 6.87 | – | 86.5 ± 9.93 | – | 81.8 | – |
| ∟ +AugFull | 95.8** ± 2.44 | (+2.13%) | 97.0* ± 2.70 | (+6.18%) | 92.0 ± 8.03 | (+11.0%) | 93.2 ± 7.34 | (+8.73%) | 55.2 ± 6.98 | (+8.96%) | 88.8 ± 11.2 | (+2.67%) | 87.0 | (+6.33%) |
| ∟ +AugMask | 97.0** ± 0.775 | (+3.38%) | 97.2* ± 1.72 | (+6.49%) | 95.8** ± 3.23 | (+15.6%) | 95.0** ± 4.30 | (+10.9%) | 60.1* ± 5.08 | (+18.7%) | 85.3 ± 5.36 | – | 88.4* | (+8.06%) |
| CDTD | 93.3 ± 4.14 | – | 90.0 ± 10.2 | – | 77.2 ± 15.4 | – | 87.6 ± 11.9 | – | 38.1 ± 10.5 | – | 93.3 ± 2.29 | – | 79.8 | – |
| ∟ +AugFull | 94.7** ± 4.83 | (+1.52%) | 95.8* ± 4.26 | (+6.45%) | 91.6 ± 9.05 | (+18.6%) | 89.5 ± 10.4 | (+2.18%) | 56.2 ± 9.51 | (+47.4%) | 94.9 ± 3.09 | (+1.68%) | 87.1 | (+9.11%) |
| ∟ +AugMask | 96.0** ± 1.79 | (+2.95%) | 95.7* ± 3.86 | (+6.41%) | 95.5** ± 3.49 | (+23.7%) | 93.1* ± 8.20 | (+6.21%) | 59.4* ± 7.55 | (+56.0%) | 98.1* ± 3.10 | (+5.13%) | 89.7** | (+12.3%) |
| TabDiff | 89.9* ± 14.5 | – | 89.9 ± 11.9 | – | 78.5 ± 20.8 | – | 85.4 ± 14.9 | – | 51.1 ± 8.47 | – | 92.4 ± 3.49 | – | 81.2 | – |
| ∟ +AugFull | 95.8** ± 3.11 | (+6.59%) | 95.4* ± 6.23 | (+6.06%) | 90.5 ± 9.91 | (+15.3%) | 92.2 ± 8.71 | (+7.90%) | 55.3 ± 10.0 | (+8.21%) | 95.5 ± 2.33 | (+3.33%) | 87.4* | (+7.67%) |
| ∟ +AugMask | 97.0** ± 1.23 | (+7.92%) | 96.5* ± 3.11 | (+7.37%) | 95.2** ± 3.76 | (+21.3%) | 95.9** ± 3.91 | (+12.2%) | 62.0* ± 6.81 | (+21.2%) | 97.2* ± 2.66 | (+5.16%) | 90.6** | (+11.6%) |

*Table 13.* α **Precision** (MAR)

| Method | Adult Score | Δ(%) | Default Score | Δ(%) | Magic Score | Δ(%) | Shoppers Score | Δ(%) | Beijing Score | Δ(%) | News Score | Δ(%) | Summary Score | Δ(%) |
|---|---|---|---|---|---|---|---|---|---|---|---|---|---|---|
| **(A) Missing-aware baselines** | | | | | | | | | | | | | | |
| MIWAE | 88.1 ± 2.45 | – | 94.2 ± 2.46 | – | 95.9 ± 3.27 | – | 92.8 ± 2.94 | – | 92.6 ± 2.90 | – | 89.1 ± 9.15 | – | 92.1 | – |
| MissDiff | 87.2 ± 13.8 | – | 48.6 ± 1.77 | – | 13.6 ± 2.22 | – | 51.8 ± 14.3 | – | 88.5 ± 9.16 | – | 0.66 ± 0.88 | – | 48.4 | – |
| DiffPuter | 72.0 ± 22.0 | – | 71.9 ± 18.4 | – | 84.2 ± 8.91 | – | 75.2 ± 22.3 | – | 90.5 ± 4.12 | – | 63.0 ± 31.4 | – | 76.2 | – |
| ForestDiff | 72.8 ± 36.2 | – | 78.6 ± 17.1 | – | 82.6 ± 5.81 | – | 64.9 ± 40.0 | – | 87.4 ± 8.17 | – | 59.0 ± 11.9 | – | 74.2 | – |
| **(B) Missing-unaware baselines & enhancements via our proposed training strategy** | | | | | | | | | | | | | | |
| TVAE | 37.1 ± 23.2 | – | 40.9 ± 21.9 | – | 71.9 ± 13.5 | – | 32.5 ± 13.0 | – | 64.8 ± 14.8 | – | 43.4 ± 21.2 | – | 48.2 | – |
| ∟ +AugFull | 60.8 ± 20.4 | (+63.8%) | 82.1 ± 6.04 | (+101%) | 88.9 ± 3.71 | (+23.7%) | 48.7 ± 25.3 | (+50.0%) | 85.1 ± 9.06 | (+31.4%) | 90.5* ± 5.97 | (+109%) | 76.0 | (+57.7%) |
| CTGAN | 73.5 ± 19.3 | – | 67.7 ± 11.0 | – | 81.0 ± 3.47 | – | – ± – | – | 75.6 ± 12.3 | – | 61.4 ± 31.1 | – | 71.9 | – |
| ∟ +AugFull | 75.2 ± 11.4 | (+2.37%) | 71.4 ± 4.70 | (+5.42%) | 86.2 ± 3.64 | (+6.36%) | 80.0 ± 7.82 | – | 86.0 ± 7.06 | (+13.7%) | 91.4* ± 5.04 | (+48.7%) | 81.7 | (+13.7%) |
| ARF | 63.2 ± 17.9 | – | 63.4 ± 21.2 | – | 46.9 ± 26.9 | – | 54.8 ± 25.4 | – | 61.5 ± 15.9 | – | 11.1 ± 0.00 | – | 55.9 | – |
| ∟ +AugFull | 96.7* ± 2.16 | (+53.0%) | 87.4 ± 5.60 | (+38.0%) | 65.4 ± 11.5 | (+39.6%) | 93.5* ± 2.28 | (+70.5%) | 97.7** ± 2.12 | (+59.0%) | 6.98 ± 0.12 | – | 82.1 | (+46.9%) |
| TabSyn | 62.0 ± 21.3 | – | 61.3 ± 26.5 | – | 75.8 ± 13.2 | – | 59.8 ± 22.2 | – | 72.8 ± 15.8 | – | 65.5 ± 27.2 | – | 66.2 | – |
| ∟ +AugFull | 96.3* ± 1.59 | (+55.4%) | 90.2 ± 6.94 | (+47.1%) | 95.4* ± 3.24 | (+25.8%) | 97.0** ± 1.76 | (+62.1%) | 96.1** ± 2.16 | (+32.1%) | 68.0 ± 10.4 | (+3.87%) | 90.5 | (+36.7%) |
| TabDDPM | 50.2 ± 28.4 | – | 74.7 ± 16.0 | – | 59.2 ± 25.6 | – | 54.3 ± 27.7 | – | 75.5 ± 15.8 | – | 15.9 ± 10.4 | – | 55.0 | – |
| ∟ +AugFull | 97.5** ± 2.03 | (+94.3%) | 82.5 ± 5.41 | (+10.4%) | 93.7 ± 4.75 | (+58.3%) | 94.9* ± 2.93 | (+74.8%) | 95.1* ± 4.15 | (+25.9%) | 14.3 ± 22.2 | – | 79.3 | (+44.2%) |
| ∟ +AugMask | 96.8* ± 3.15 | (+92.9%) | 88.9 ± 5.33 | (+19.0%) | 95.4* ± 2.54 | (+61.0%) | 90.7 ± 7.46 | (+67.1%) | 96.8** ± 2.73 | (+28.1%) | 2.64 ± 1.16 | – | 78.3 | (+42.5%) |
| CDTD | 55.4 ± 25.5 | – | 62.1 ± 24.8 | – | 79.3 ± 12.1 | – | 65.2 ± 26.6 | – | 80.3 ± 13.4 | – | 60.0 ± 28.5 | – | 67.1 | – |
| ∟ +AugFull | 96.5* ± 2.73 | (+74.1%) | 92.3* ± 7.06 | (+48.6%) | 95.9* ± 4.73 | (+20.9%) | 94.6* ± 2.91 | (+45.2%) | 96.6** ± 2.46 | (+20.3%) | 84.5 ± 14.7 | (+40.9%) | 93.4* | (+39.3%) |
| ∟ +AugMask | 98.4** ± 1.21 | (+77.5%) | 97.2** ± 1.71 | (+56.4%) | 96.6** ± 3.37 | (+21.8%) | 94.1* ± 5.18 | (+44.4%) | 96.6** ± 2.20 | (+20.2%) | 92.3* ± 3.25 | (+53.9%) | 95.9** | (+43.0%) |
| TabDiff | 50.5 ± 29.0 | – | 59.2 ± 24.5 | – | 58.7 ± 24.8 | – | 47.5 ± 26.9 | – | 74.5 ± 15.9 | – | 45.7 ± 9.98 | – | 56.0 | – |
| ∟ +AugFull | 96.9* ± 2.23 | (+91.8%) | 91.5 ± 7.01 | (+54.7%) | 93.2 ± 4.38 | (+58.9%) | 97.2** ± 2.45 | (+104%) | 95.1* ± 4.05 | (+27.7%) | 35.4 ± 6.24 | – | 85.5 | (+52.5%) |
| ∟ +AugMask | 97.9** ± 1.69 | (+93.7%) | 96.7** ± 1.54 | (+63.4%) | 97.0** ± 3.01 | (+65.4%) | 96.3** ± 3.57 | (+103%) | 98.0** ± 1.62 | (+31.6%) | 28.9 ± 1.74 | – | 85.8** | (+53.2%) |

*Table 14.* $\beta$-**Recall** (MAR)

| | Adult | | Default | | Magic | | Shoppers | | Beijing | | News | | Summary | |
|---|---|---|---|---|---|---|---|---|---|---|---|---|---|---|
| **Method** | Score | Δ(%) | Score | Δ(%) | Score | Δ(%) | Score | Δ(%) | Score | Δ(%) | Score | Δ(%) | **Score** | Δ(%) |
| **(A) Missing-aware baselines** | | | | | | | | | | | | | | |
| MIWAE | 6.62 ± 0.64 | – | 18.7 ± 5.56 | – | 26.3 ± 10.6 | – | 19.4 ± 2.17 | – | 43.2 ± 1.92 | – | 10.2 ± 3.66 | – | 20.7 | – |
| MissDiff | 15.0 ± 12.0 | – | 0.00 ± 0.00 | – | 0.01 ± 0.01 | – | 0.03 ± 0.02 | – | 44.0 ± 6.34 | – | 0.00 ± 0.00 | – | 9.84 | – |
| DiffPuter | 5.82 ± 2.03 | – | 4.25 ± 2.22 | – | 20.1 ± 15.6 | – | 12.3 ± 5.75 | – | 39.9 ± 4.46 | – | 8.98 ± 11.3 | – | 15.2 | – |
| ForestDiff | 26.6 ± 16.4 | – | 27.2 ± 16.1 | – | 37.8 ± 14.9 | – | 29.3 ± 19.7 | – | 42.8 ± 10.8 | – | 25.1 ± 18.9 | – | 31.5 | – |
| **(B) Missing-unaware baselines & enhancements via our proposed training strategy** | | | | | | | | | | | | | | |
| TVAE | 8.67 ± 8.71 | – | 4.80 ± 5.99 | – | 5.47 ± 8.16 | – | 10.3 ± 11.7 | – | 0.76 ± 1.09 | – | 1.73 ± 3.68 | – | 5.32 | – |
| ∟ +AugFull | 14.8 ± 8.24 | (+71.0%) | 14.2 ± 2.69 | (+196%) | 18.1 ± 9.14 | (+231%) | 22.0 ± 14.0 | (+113%) | 5.32 ± 5.33 | (+598%) | 11.3 ± 6.08 | (+553%) | 14.3 | (+169%) |
| CTGAN | 6.64 ± 2.09 | – | 4.23 ± 2.15 | – | 2.11 ± 2.76 | – | – ± – | – | 5.31 ± 1.69 | – | 4.17 ± 7.15 | – | 4.49 | – |
| ∟ +AugFull | 7.46 ± 2.13 | (+12.3%) | 6.70 ± 3.21 | (+58.5%) | 9.51 ± 4.21 | (+351%) | 17.2 ± 4.51 | – | 5.23 ± 0.46 | – | 16.1 ± 9.08 | (+285%) | 10.4 | (+131%) |
| ARF | 3.92 ± 3.84 | – | 3.98 ± 2.84 | – | 0.19 ± 0.27 | – | 8.66 ± 10.0 | – | 21.4 ± 18.5 | – | 0.02 ± 0.00 | – | 6.78 | – |
| ∟ +AugFull | 20.1 ± 8.11 | (+414%) | 15.7 ± 7.73 | (+293%) | 1.37 ± 1.80 | (+624%) | 13.4 ± 5.43 | (+55.2%) | 49.1** ± 2.17 | (+129%) | 0.26 ± 0.21 | (+1573%) | 18.5 | (+173%) |
| TabSyn | 18.0 ± 12.4 | – | 14.1 ± 10.8 | – | 20.7 ± 12.8 | – | 20.7 ± 12.8 | – | 21.8 ± 17.3 | – | 3.44 ± 5.09 | – | 14.9 | – |
| ∟ +AugFull | 26.6* ± 12.0 | (+47.8%) | 22.0 ± 10.3 | (+56.8%) | 23.1 ± 14.8 | (+103%) | 32.1* ± 7.08 | (+54.9%) | 48.5* ± 1.14 | (+122%) | 8.87 ± 6.20 | (+158%) | 26.9 | (+80.4%) |
| TabDDPM | 18.9 ± 14.9 | – | 14.8 ± 9.90 | – | 15.0 ± 13.7 | – | 25.0 ± 13.6 | – | 20.6 ± 16.7 | – | 1.59 ± 1.22 | – | 16.0 | – |
| ∟ +AugFull | 33.4** ± 11.8 | (+76.8%) | 25.1 ± 7.90 | (+70.1%) | 33.9 ± 11.9 | (+126%) | 36.0* ± 7.08 | (+43.9%) | 48.5* ± 1.25 | (+136%) | 1.66 ± 1.87 | (+4.44%) | 29.7 | (+85.6%) |
| ∟ **+AugMask** | 39.9** ± 8.37 | (+111%) | 30.8* ± 5.30 | (+109%) | 37.7* ± 6.56 | (+151%) | 39.3** ± 6.63 | (+57.1%) | 45.7 ± 5.68 | (+122%) | 0.71 ± 0.25 | – | **32.3*** | (+102%) |
| CDTD | 20.5 ± 14.5 | – | 13.1 ± 10.6 | – | 10.9 ± 13.6 | – | 14.8 ± 10.7 | – | 19.0 ± 16.0 | – | 6.48 ± 11.4 | – | 14.1 | – |
| ∟ +AugFull | 31.5** ± 14.3 | (+53.3%) | 20.8 ± 10.6 | (+58.4%) | 24.7 ± 15.7 | (+127%) | 30.5* ± 8.18 | (+106%) | 47.1 ± 3.55 | (+148%) | 20.5 ± 14.1 | (+217%) | 29.2 | (+107%) |
| ∟ **+AugMask** | 36.6** ± 11.7 | (+78.3%) | 28.5* ± 11.7 | (+118%) | 37.0* ± 9.01 | (+240%) | 34.1* ± 13.0 | (+131%) | 39.9 ± 12.3 | (+110%) | 37.1** ± 10.1 | (+473%) | 35.5** | (+152%) |
| TabDiff | 19.6 ± 15.6 | – | 18.5 ± 13.9 | – | 16.9 ± 15.6 | – | 28.9 ± 16.7 | – | 21.7 ± 17.6 | – | 0.69 ± 0.29 | – | 17.7 | – |
| ∟ +AugFull | 32.5** ± 13.6 | (+66.1%) | 32.2** ± 12.3 | (+74.0%) | 36.1 ± 13.1 | (+113%) | 39.4** ± 8.64 | (+36.1%) | 50.3** ± 2.13 | (+132%) | 0.73 ± 0.06 | (+5.78%) | 32.2** | (+81.8%) |
| ∟ **+AugMask** | 40.3** ± 10.4 | (+106%) | 39.8** ± 7.30 | (+115%) | 40.6* ± 8.35 | (+140%) | 45.3** ± 6.52 | (+56.4%) | 49.3** ± 3.60 | (+128%) | 0.61 ± 0.11 | – | **36.0**** | (+103%) |

*Table 15.* **Trend** (MAR)

| | Adult | | Default | | Magic | | Shoppers | | Beijing | | News | | Summary | |
|---|---|---|---|---|---|---|---|---|---|---|---|---|---|---|
| **Method** | Score | Δ(%) | Score | Δ(%) | Score | Δ(%) | Score | Δ(%) | Score | Δ(%) | Score | Δ(%) | **Score** | Δ(%) |
| **(A) Missing-aware baselines** | | | | | | | | | | | | | | |
| MIWAE | 69.0 ± 3.82 | – | 82.4 ± 2.02 | – | 92.5 ± 2.48 | – | 82.3 ± 1.91 | – | 76.1 ± 3.05 | – | 95.8 ± 0.72 | – | 83.0 | – |
| MissDiff | 72.8 ± 14.0 | – | 70.3 ± 11.5 | – | 79.9 ± 2.56 | – | 60.9 ± 15.0 | – | 79.0 ± 15.0 | – | 90.2 ± 1.34 | – | 75.5 | – |
| DiffPuter | 64.6 ± 9.79 | – | 66.5 ± 4.98 | – | 86.4 ± 5.97 | – | 74.8 ± 7.81 | – | 77.7 ± 8.03 | – | 92.3 ± 4.06 | – | 77.0 | – |
| ForestDiff | 77.4 ± 18.8 | – | 77.9 ± 11.2 | – | 87.4 ± 3.27 | – | 77.5 ± 18.6 | – | 82.5 ± 10.5 | – | 93.7 ± 3.45 | – | 82.7 | – |
| **(B) Missing-unaware baselines & enhancements via our proposed training strategy** | | | | | | | | | | | | | | |
| TVAE | 64.6 ± 12.6 | – | 63.2 ± 9.25 | – | 80.9 ± 5.92 | – | 69.8 ± 9.23 | – | 33.7 ± 16.8 | – | 78.5 ± 24.6 | – | 67.4 | – |
| ∟ +AugFull | 74.3 ± 15.8 | (+15.0%) | 79.7 ± 2.54 | (+26.2%) | 91.1 ± 3.05 | (+12.6%) | 79.2 ± 11.0 | (+13.5%) | 68.0 ± 17.5 | (+102%) | 94.5 ± 1.30 | (+20.4%) | 81.1 | (+20.5%) |
| CTGAN | 62.2 ± 8.83 | – | 66.7 ± 6.61 | – | 81.2 ± 5.33 | – | – ± – | – | 54.3 ± 16.4 | – | 92.1 ± 1.88 | – | 71.3 | – |
| ∟ +AugFull | 69.3 ± 5.53 | (+11.4%) | 74.8 ± 1.69 | (+12.1%) | 87.7 ± 3.16 | (+8.01%) | 83.4 ± 1.93 | – | 67.2 ± 8.14 | (+23.9%) | 94.8 ± 0.46 | (+2.92%) | 79.5 | (+11.6%) |
| ARF | 39.0 ± 8.61 | – | 25.8 ± 7.70 | – | 50.8 ± 10.5 | – | 28.7 ± 9.90 | – | 29.1 ± 15.1 | – | 16.4 ± 0.00 | – | 34.2 | – |
| ∟ +AugFull | 61.9 ± 2.26 | (+58.9%) | 54.2 ± 2.55 | (+110%) | 70.3 ± 5.67 | (+38.3%) | 35.2 ± 5.85 | (+22.3%) | 63.0 ± 2.14 | (+116%) | 41.4 ± 14.7 | (+153%) | 55.8 | (+63.1%) |
| TabSyn | 68.0 ± 19.0 | – | 71.6 ± 12.2 | – | 85.0 ± 6.82 | – | 76.1 ± 12.3 | – | 65.6 ± 18.9 | – | 92.7 ± 2.50 | – | 76.5 | – |
| ∟ +AugFull | 91.3** ± 3.37 | (+34.3%) | 86.1** ± 2.70 | (+20.3%) | 93.2* ± 3.39 | (+9.65%) | 91.8** ± 2.39 | (+20.7%) | 92.9** ± 2.55 | (+41.8%) | 96.1* ± 1.02 | (+3.67%) | 91.9** | (+20.2%) |
| TabDDPM | 68.7 ± 16.1 | – | 70.0 ± 9.67 | – | 81.9 ± 6.60 | – | 76.0 ± 11.5 | – | 63.3 ± 18.9 | – | 87.1 ± 1.20 | – | 74.5 | – |
| ∟ +AugFull | 89.3** ± 3.48 | (+29.9%) | 75.5 ± 2.60 | (+7.99%) | 91.1 ± 1.91 | (+11.2%) | 91.6** ± 2.29 | (+20.4%) | 92.2** ± 2.82 | (+45.6%) | 85.2 ± 1.61 | – | 87.4* | (+17.3%) |
| ∟ **+AugMask** | 91.3** ± 2.19 | (+32.8%) | 78.3 ± 2.22 | (+11.9%) | 90.9 ± 2.38 | (+10.9%) | 89.1* ± 4.25 | (+17.2%) | 92.0** ± 3.58 | (+45.3%) | 85.5 ± 1.88 | – | 87.8* | (+17.8%) |
| CDTD | 69.7 ± 16.1 | – | 72.3 ± 11.5 | – | 83.9 ± 5.59 | – | 73.6 ± 14.9 | – | 64.1 ± 16.9 | – | 92.8 ± 3.02 | – | 76.1 | – |
| ∟ +AugFull | 87.9** ± 6.05 | (+26.1%) | 86.4** ± 4.22 | (+19.5%) | 93.1* ± 3.30 | (+10.9%) | 91.2** ± 2.76 | (+23.9%) | 90.9** ± 4.28 | (+42.0%) | 96.7** ± 1.26 | (+4.22%) | 91.0** | (+19.7%) |
| ∟ **+AugMask** | 91.5** ± 3.00 | (+31.2%) | 87.5** ± 3.87 | (+21.1%) | 92.8** ± 2.85 | (+10.6%) | 92.1** ± 3.60 | (+25.0%) | 91.8** ± 2.93 | (+43.3%) | 97.2** ± 1.14 | (+4.67%) | 92.1** | (+21.1%) |
| TabDiff | 69.3 ± 16.0 | – | 71.8 ± 12.2 | – | 84.9 ± 6.54 | – | 78.2 ± 11.5 | – | 65.0 ± 18.4 | – | 92.3 ± 2.11 | – | 76.9 | – |
| ∟ +AugFull | 89.0** ± 4.99 | (+28.4%) | 87.3** ± 3.39 | (+21.5%) | 93.0** ± 2.94 | (+9.65%) | 93.8** ± 2.70 | (+19.9%) | 94.3** ± 3.00 | (+45.1%) | 95.5 ± 0.64 | (+3.42%) | 92.1** | (+19.8%) |
| ∟ **+AugMask** | 93.0** ± 2.31 | (+34.1%) | 88.1** ± 2.94 | (+22.6%) | 92.4* ± 3.36 | (+8.92%) | 94.7** ± 2.60 | (+21.2%) | 94.8** ± 2.45 | (+45.8%) | 95.5 ± 0.56 | (+3.42%) | 93.1** | (+21.0%) |

*Table 16.* **Shape** (MAR)

| Method | Adult Score | Δ(%) | Default Score | Δ(%) | Magic Score | Δ(%) | Shoppers Score | Δ(%) | Beijing Score | Δ(%) | News Score | Δ(%) | Summary Score | Δ(%) |
|---|---|---|---|---|---|---|---|---|---|---|---|---|---|---|
| **(A) Missing-aware baselines** | | | | | | | | | | | | | | |
| MIWAE | 86.2 ± 0.60 | – | 92.9 ± 1.36 | – | 94.0 ± 3.32 | – | 85.1 ± 1.39 | – | 91.3 ± 1.11 | – | 93.6 ± 0.90 | – | 90.5 | – |
| MissDiff | 76.1 ± 6.52 | – | 70.3 ± 9.07 | – | 63.0 ± 5.72 | – | 61.3 ± 4.58 | – | 74.6 ± 3.09 | – | 42.6 ± 2.70 | – | 64.6 | – |
| DiffPuter | 83.3 ± 5.51 | – | 71.6 ± 10.6 | – | 80.8 ± 17.0 | – | 77.6 ± 6.58 | – | 86.6 ± 7.49 | – | 66.2 ± 25.4 | – | 77.7 | – |
| ForestDiff | 84.7 ± 15.6 | – | 80.6 ± 14.0 | – | 77.3 ± 8.45 | – | 75.5 ± 19.2 | – | 82.5 ± 10.1 | – | 80.1 ± 12.5 | – | 80.1 | – |
| **(B) Missing-unaware baselines & enhancements via our proposed training strategy** | | | | | | | | | | | | | | |
| TVAE | 77.1 ± 10.1 | – | 56.6 ± 18.2 | – | 54.8 ± 22.6 | – | 72.7 ± 11.1 | – | 47.8 ± 8.90 | – | 44.2 ± 20.5 | – | 58.5 | – |
| └ +AugFull | 87.1 ± 6.68 | (+12.9%) | 85.8 ± 5.35 | (+51.5%) | 92.0 ± 3.72 | (+67.7%) | 85.1 ± 7.66 | (+17.0%) | 84.6 ± 8.40 | (+76.9%) | 83.2 ± 4.95 | (+88.3%) | 86.3 | (+47.5%) |
| CTGAN | 78.8 ± 8.29 | – | 63.0 ± 15.8 | – | 64.0 ± 16.7 | – | – ± – | – | 65.0 ± 14.6 | – | 63.2 ± 19.9 | – | 66.8 | – |
| └ +AugFull | 87.2 ± 1.54 | (+10.6%) | 85.4 ± 1.46 | (+35.5%) | 89.1 ± 3.18 | (+39.2%) | 86.5 ± 3.93 | – | 84.5 ± 1.92 | (+30.1%) | 90.2 ± 2.37 | (+42.9%) | 87.2 | (+30.5%) |
| ARF | 64.2 ± 16.8 | – | 48.9 ± 10.1 | – | 39.2 ± 8.82 | – | 56.2 ± 9.69 | – | 49.9 ± 14.7 | – | 34.9 ± 0.00 | – | 51.1 | – |
| └ +AugFull | **94.3**\*\* ± **0.62** | (+46.7%) | 92.0 ± 2.57 | (+88.1%) | 53.8 ± 6.06 | (+37.4%) | 66.4 ± 7.88 | (+18.2%) | **95.9**\*\* ± **1.28** | (+92.3%) | 43.0 ± 3.71 | (+23.2%) | 77.7 | (+52.1%) |
| TabSyn | 78.9 ± 14.0 | – | 69.5 ± 19.4 | – | 70.0 ± 20.3 | – | 75.7 ± 13.5 | – | 68.7 ± 19.8 | – | 70.3 ± 17.9 | – | 72.2 | – |
| └ +AugFull | **96.9**\*\* ± **1.14** | (+22.8%) | 91.7\* ± 5.38 | (+31.9%) | **94.9**\*\* ± **3.83** | (+35.6%) | 90.9\*\* ± 3.47 | (+20.0%) | 96.7\*\* ± 1.69 | (+40.8%) | 91.3 ± 3.34 | (+29.9%) | **93.8**\*\* | (+29.9%) |
| TabDDPM | 80.3 ± 11.4 | – | 78.3 ± 11.4 | – | 73.1 ± 17.8 | – | 78.1 ± 11.2 | – | 71.2 ± 17.5 | – | 48.4 ± 7.11 | – | 71.6 | – |
| └ +AugFull | 96.7\*\* ± 1.77 | (+20.5%) | 88.7 ± 3.42 | (+13.2%) | 93.2\* ± 5.20 | (+27.5%) | **92.5**\*\* ± **2.62** | (+18.4%) | 95.5\*\* ± 2.30 | (+34.0%) | 44.4 ± 12.4 | – | 84.9\* | (+18.7%) |
| └ +AugMask | **97.0**\*\* ± **1.34** | (+20.8%) | 91.5 ± 2.71 | (+16.8%) | **95.0**\*\* ± **2.45** | (+30.0%) | **92.5**\*\* ± **3.73** | (+18.4%) | **95.7**\*\* ± **2.01** | (+34.3%) | – | – | 85.5\* | (+19.5%) |
| CDTD | 80.3 ± 12.2 | – | 66.6 ± 18.7 | – | 63.7 ± 21.4 | – | 72.8 ± 12.2 | – | 65.0 ± 19.5 | – | 63.2 ± 20.1 | – | 68.6 | – |
| └ +AugFull | 96.5\*\* ± 1.40 | (+20.1%) | 92.5\* ± 4.44 | (+38.8%) | 95.2\*\* ± 3.40 | (+49.4%) | 91.0\*\* ± 3.22 | (+25.0%) | 95.0\*\* ± 1.36 | (+46.0%) | 91.7 ± 3.64 | (+45.0%) | 93.6\*\* | (+36.4%) |
| └ +AugMask | **97.0**\*\* ± **1.39** | (+20.7%) | **95.4**\*\* ± **2.54** | (+43.2%) | 94.6\*\* ± 2.72 | (+48.4%) | **93.3**\*\* ± **3.13** | (+28.1%) | 94.4\*\* ± 1.59 | (+45.2%) | **93.9**\*\* ± **1.23** | (+48.5%) | **94.8**\*\* | (+38.1%) |
| TabDiff | 80.0 ± 11.7 | – | 72.3 ± 16.6 | – | 71.2 ± 17.6 | – | 78.3 ± 12.1 | – | 70.2 ± 17.9 | – | 84.7 ± 3.85 | – | 76.1 | – |
| └ +AugFull | 96.6\*\* ± 2.04 | (+20.8%) | 95.0\*\* ± 3.36 | (+31.4%) | 93.1\* ± 5.18 | (+30.7%) | 93.6\*\* ± 3.51 | (+19.5%) | 96.0\*\* ± 2.41 | (+36.8%) | 88.3 ± 0.45 | (+4.19%) | 93.8\*\* | (+23.3%) |
| └ +AugMask | **97.5**\*\* ± **1.41** | (+22.0%) | **96.7**\*\* ± **1.52** | (+33.7%) | **94.8**\*\* ± **2.68** | (+33.0%) | **95.8**\*\* ± **2.15** | (+22.4%) | **96.1**\*\* ± **1.53** | (+37.0%) | 87.4 ± 1.21 | (+3.17%) | **94.7**\*\* | (+24.4%) |

*Table 17.* **ML Task** (MAR)

| Method | Adult Score | Δ(%) | Default Score | Δ(%) | Magic Score | Δ(%) | Shoppers Score | Δ(%) | Beijing Score | Δ(%) | News Score | Δ(%) | Summary Score | Δ(%) |
|---|---|---|---|---|---|---|---|---|---|---|---|---|---|---|
| **(A) Missing-aware baselines** | | | | | | | | | | | | | | |
| MIWAE | 89.0 ± 3.83 | – | 93.0 ± 4.68 | – | 91.9 ± 7.69 | – | 86.3 ± 6.92 | – | 47.2 ± 4.56 | – | 95.0 ± 3.09 | – | 83.7 | – |
| MissDiff | 72.0 ± 15.8 | – | 76.8 ± 9.79 | – | 80.4 ± 14.4 | – | 60.4 ± 19.9 | – | 42.9 ± 11.4 | – | 39.0 ± 19.0 | – | 61.9 | – |
| DiffPuter | 49.7 ± 5.91 | – | 54.2 ± 5.39 | – | 53.9 ± 8.51 | – | 54.6 ± 13.2 | – | 40.9 ± 9.46 | – | 80.4 ± 17.2 | – | 55.6 | – |
| ForestDiff | 94.8 ± 2.88 | – | 91.5 ± 12.1 | – | 90.5 ± 9.39 | – | 91.9 ± 7.59 | – | 50.4 ± 12.9 | – | 95.3 ± 1.29 | – | 85.7 | – |
| **(B) Missing-unaware baselines & enhancements via our proposed training strategy** | | | | | | | | | | | | | | |
| TVAE | 88.7 ± 10.0 | – | 79.0 ± 11.7 | – | 75.2 ± 12.3 | – | 87.8 ± 6.10 | – | 34.5 ± 1.33 | – | 87.6 ± 4.75 | – | 77.7 | – |
| └ +AugFull | 91.7 ± 5.30 | (+3.37%) | 91.9 ± 4.48 | (+16.3%) | 87.4 ± 7.90 | (+16.1%) | 92.8\* ± 5.09 | (+5.60%) | 45.7 ± 6.06 | (+32.4%) | 94.6 ± 2.22 | (+7.93%) | 84.0 | (+8.17%) |
| CTGAN | 72.3 ± 17.2 | – | 70.3 ± 6.55 | – | 65.1 ± 15.6 | – | – ± – | – | 32.9 ± 3.86 | – | 82.8 ± 13.0 | – | 64.7 | – |
| └ +AugFull | 85.1 ± 6.89 | (+17.7%) | 79.2 ± 8.83 | (+12.7%) | 79.5 ± 8.64 | (+22.2%) | 75.5 ± 11.3 | – | 38.2 ± 4.30 | (+16.0%) | 94.2 ± 2.09 | (+13.8%) | 75.3 | (+16.4%) |
| ARF | 90.6 ± 3.79 | – | 82.0 ± 8.97 | – | 52.6 ± 3.52 | – | 55.8 ± 8.99 | – | 39.3 ± 4.49 | – | 74.2 ± 0.00 | – | 65.3 | – |
| └ +AugFull | 95.7\*\* ± 2.76 | (+5.64%) | 92.9 ± 7.05 | (+13.3%) | 48.4 ± 6.72 | – | 57.4 ± 7.89 | (+2.79%) | 53.8\* ± 7.43 | (+37.0%) | 87.4 ± 2.55 | (+17.8%) | 73.5 | (+12.5%) |
| TabSyn | 89.0 ± 11.9 | – | 85.6 ± 12.9 | – | 79.4 ± 13.0 | – | 89.8 ± 5.71 | – | 42.6 ± 11.9 | – | 92.1 ± 10.9 | – | 79.7 | – |
| └ +AugFull | **96.3**\*\* ± **1.78** | (+8.23%) | 94.4\* ± 6.08 | (+10.3%) | 92.9\* ± 6.68 | (+17.1%) | 92.4\* ± 5.56 | (+3.03%) | 55.0\* ± 8.54 | (+28.9%) | 93.3 ± 3.60 | (+1.20%) | 87.4\* | (+9.58%) |
| TabDDPM | 91.3 ± 9.15 | – | 88.0 ± 10.6 | – | 82.6 ± 11.5 | – | 81.1 ± 16.8 | – | 45.8 ± 8.65 | – | 83.2 ± 14.1 | – | 78.7 | – |
| └ +AugFull | 95.9\*\* ± 2.73 | (+5.02%) | 95.6\* ± 4.57 | (+8.65%) | 93.0\*\* ± 6.91 | (+12.5%) | 92.7\* ± 7.09 | (+14.3%) | 57.2\*\* ± 6.83 | (+25.0%) | 84.5 ± 11.4 | (+1.51%) | 86.3\* | (+9.69%) |
| └ +AugMask | **97.0**\*\* ± **0.99** | (+6.17%) | **96.7**\*\* ± **3.04** | (+9.81%) | 95.7\*\* ± 3.22 | (+15.8%) | **95.8**\*\* ± **2.70** | (+18.0%) | **58.5**\*\* ± **6.17** | (+27.8%) | 86.4 ± 4.36 | (+3.91%) | **88.2**\*\* | (+12.1%) |
| CDTD | 92.0 ± 5.85 | – | 88.1 ± 9.29 | – | 81.9 ± 13.5 | – | 84.3 ± 13.5 | – | 37.7 ± 12.6 | – | 92.1 ± 10.9 | – | 79.4 | – |
| └ +AugFull | 94.3\* ± 5.56 | (+2.51%) | 95.2\* ± 4.72 | (+8.12%) | 89.9\* ± 11.6 | (+9.76%) | 92.1\* ± 8.00 | (+9.31%) | 55.4\*\* ± 10.5 | (+46.9%) | 96.6\* ± 2.48 | (+4.81%) | 87.3\* | (+9.96%) |
| └ +AugMask | 95.8\*\* ± 2.36 | (+4.13%) | 95.4\* ± 4.56 | (+8.38%) | 94.6\*\* ± 4.62 | (+15.5%) | 92.7\* ± 8.18 | (+9.99%) | 57.6\*\* ± 8.85 | (+52.5%) | 94.6 ± 3.74 | (+2.62%) | **88.4**\*\* | (+11.4%) |
| TabDiff | 90.2 ± 11.6 | – | 87.5 ± 12.3 | – | 82.3 ± 13.4 | – | 85.0 ± 12.6 | – | 48.1 ± 9.06 | – | 90.8 ± 4.22 | – | 80.7 | – |
| └ +AugFull | 95.3\*\* ± 4.05 | (+5.73%) | 94.6\* ± 5.76 | (+8.03%) | 91.2\* ± 9.32 | (+10.8%) | 91.8\* ± 8.68 | (+8.02%) | 58.1\*\* ± 8.89 | (+20.8%) | **96.0** ± 2.65 | (+5.75%) | 87.7\* | (+8.79%) |
| └ +AugMask | 96.7\*\* ± 1.75 | (+7.24%) | 96.2\* ± 3.95 | (+9.91%) | **94.6**\*\* ± **5.18** | (+14.9%) | 95.3\*\* ± 4.49 | (+12.1%) | 59.8\*\* ± 8.32 | (+24.3%) | 95.7\* ± 1.65 | (+5.46%) | **89.7**\*\* | (+11.2%) |

## D.3. Robustness Plots (MCAR)

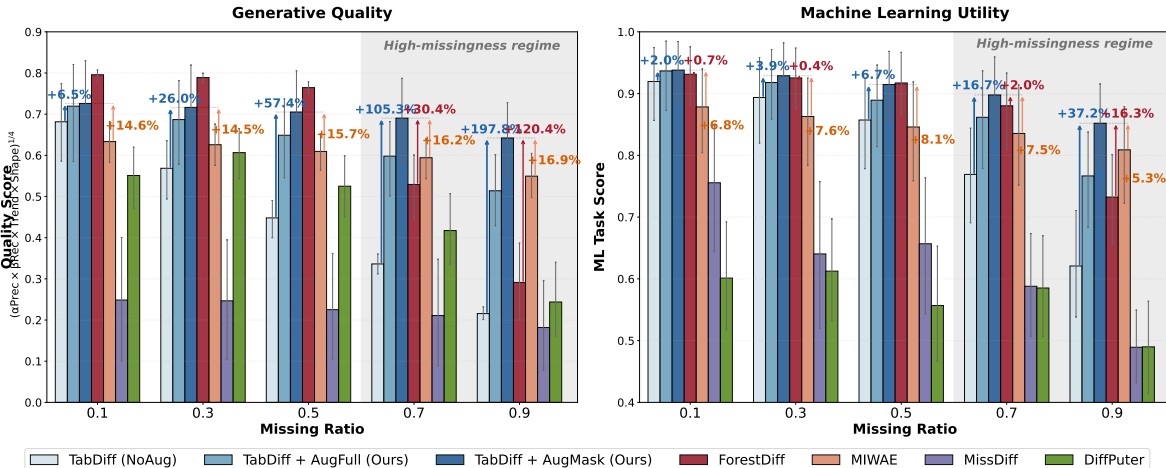

*Figure 9.* Robustness across missing ratios (**MCAR**) for **TabDiff** compared across all missing-aware baselines. **Generative Quality** (Left) and **Machine Learning Utility** (Right)

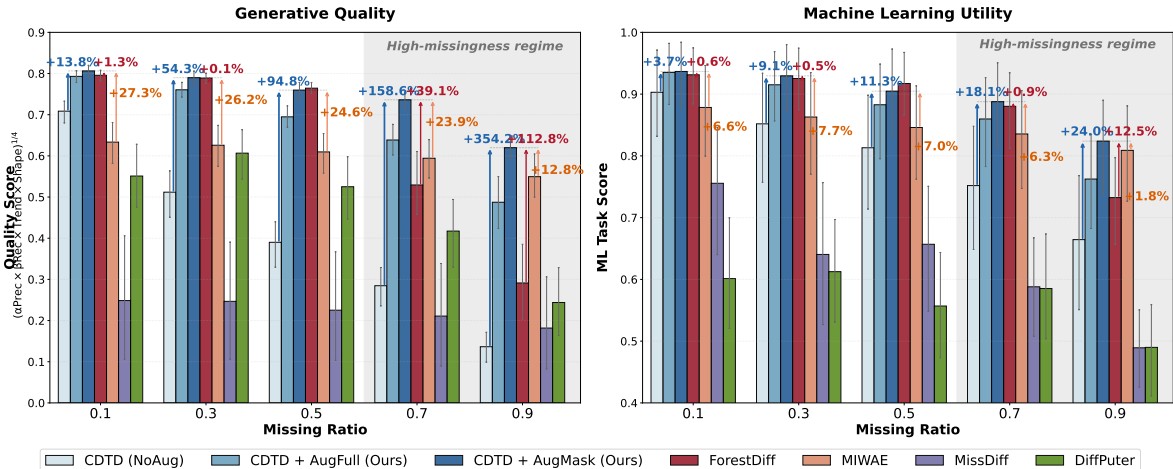

*Figure 10.* Robustness across missing ratios (**MCAR**) for **CDTD** compared across all missing-aware baselines. **Generative Quality** (Left) and **Machine Learning Utility** (Right)

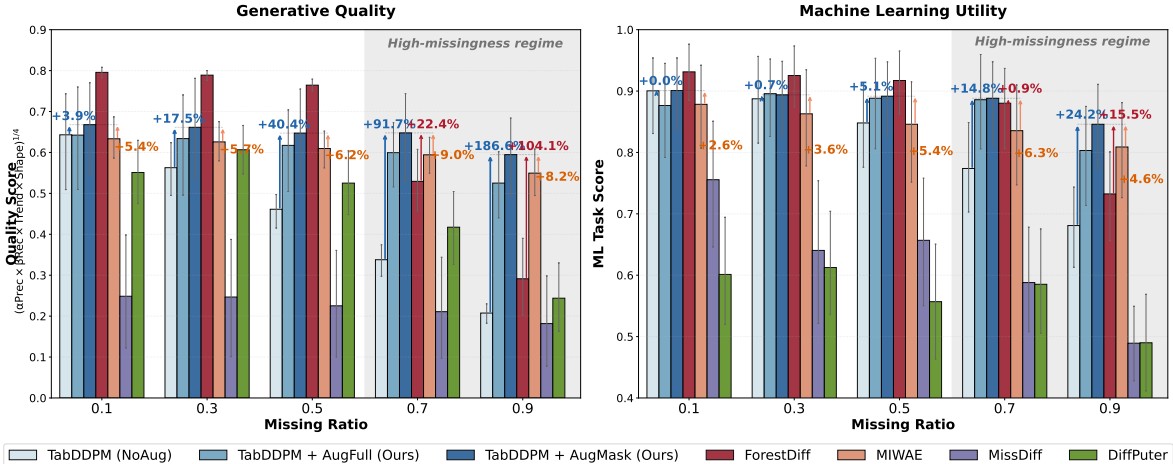

*Figure 11.* Robustness across missing ratios (**MCAR**) for **TabDDPM** compared across all missing-aware baselines. **Generative Quality** (Left) and **Machine Learning Utility** (Right)

## D.4. Robustness Plots (MAR)

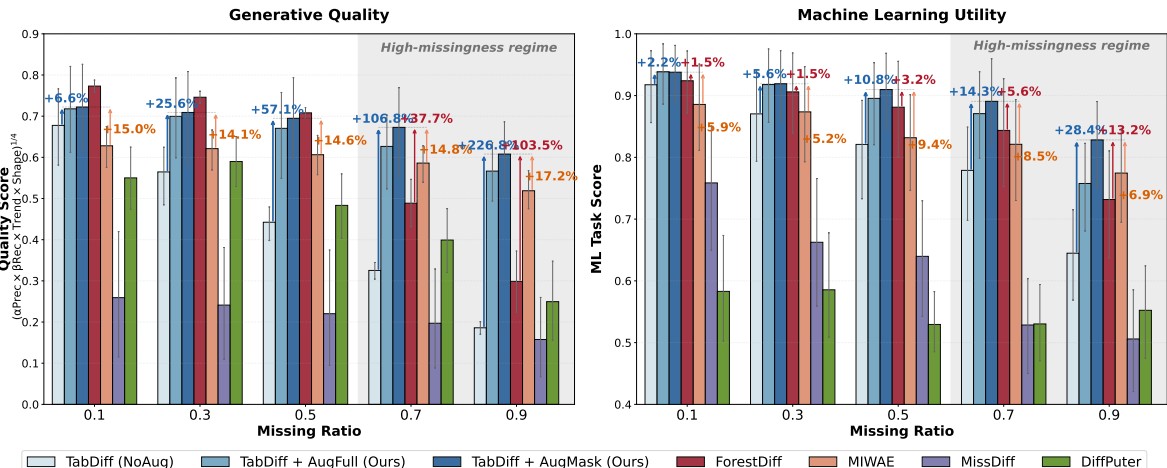

*Figure 12.* Robustness across missing ratios (**MAR**) for **TabDiff** compared across all missing-aware baselines. **Generative Quality** (Left) and **Machine Learning Utility** (Right)

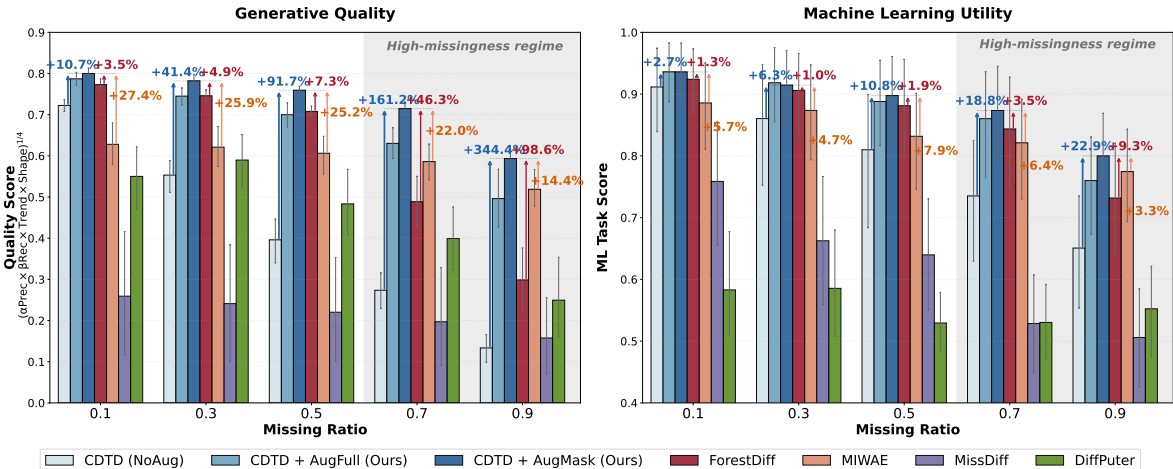

*Figure 13.* Robustness across missing ratios (**MAR**) for **CDTD** compared across all missing-aware baselines. **Generative Quality** (Left) and **Machine Learning Utility** (Right)

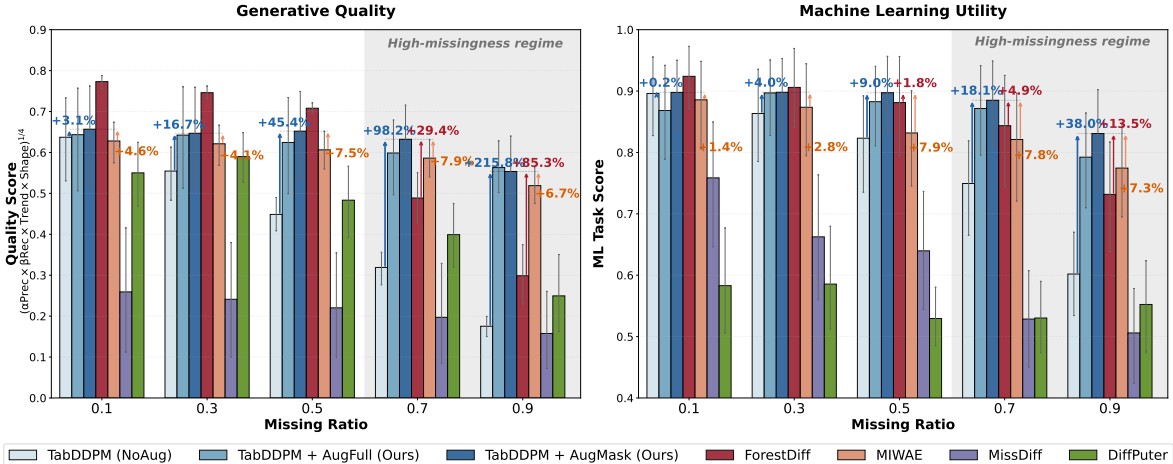

*Figure 14.* Robustness across missing ratios (**MAR**) for **TabDDPM** compared across all missing-aware baselines. **Generative Quality** (Left) and **Machine Learning Utility** (Right)

