# OpenReview forum: "AugMask: Training Diffusion Models on Incomplete Tabular Data via Stochastic Augmentation and Masking"
_ICML.cc/2026/Conference — ICML 2026 regular_

### Official Review · Reviewer_6FM8 · 2026-03-05

**Soundness:** 3
**Presentation:** 3
**Significance:** 2
**Originality:** 2
**Overall Recommendation:** 4
**Confidence:** 1

**Summary:**

This work focuses on the use of score-based generative models for tabular datasets with missing values. The goal is to adapt diffusion backbones to tabular data with missing values *without* changing the architecture, and instead only changing how the inputs/losses are constructed. To this end, the authors propose **AugMask**. Their approach completes inputs with missing values by augmenting missing entries with a conditional auxiliary model. They apply loss masking such that the diffusion denoiser is supervised by only the observed entries, while the augmented missing values are treated only as additional conditioning context. Moreover, they also compare with a variant (**AugFull**) that supervises all coordinates instead of just the observed ones.

Interestingly, by connecting their objective to an ideal Rao-Blackwellized objective, the authors show when the condition uncertainty of a missing entry is high, training is encouraged to make the observed coordinate reconstructions less sensitive to that uncertain missing value. On the other hand, when the uncertainty is low, the model can rely on the completed context more.

The authors run experiments on both tabular classification and regression tasks, where they simulate missingness under MCAR/MAR at different missing ratios. They evaluate their approach based on generative quality and downstream utility. Their results indicate that **AugMask** improves performance, with the gains becoming more noticeable in the "messier" settings.

**Compliance With Llm Reviewing Policy:**

Affirmed.

**Final Justification:**

The authors addressed my concerns. I maintain my positive score and recommend acceptance, noting that I'm not familiar with this area of research.

**Key Questions For Authors:**

Please see my questions below.

Q1) The analysis in the paper assumes a one missing coordinate setup. Can you naturally and formally extend the interpretation of the sensitivity penality to the case of multiple missing coordinates where interaction terms can now appear?

Q2) Equation (13) writes the Rao-Blackwellized approximation as several different terms, but the paper emphasises the sensitivity penality and treats the other terms as corrections. Can you please justify this formally? For example, can you bound or empirically measure the magnitude of these addtional terms?

Q3) Can you provide experiments on the MNAR setting, or justify why you have not considered it (as far as I am aware you define MNAR but do not discuss it further)? I am not too familiar with this area, but is it also possible to provide experiments on one real incomplete dataset?

**Limitations:**

Yes.

**Strengths And Weaknesses:**

# Soundness

Overall the paper is technically sound:
- The motivation for the problem is clear. The authors clearly explain why diffusion backbones can struggle with missing values and why placeholder fillings can cause spurious cues.
- The ideal Rao-Blackwellized objective, while not used in practice, provides a good motivation for why sampling augmentation is better than naively using deterministic fills (specifically, in Prop. 3.1 they show that this ideal objective yields a variance weighted sensitivity penalty).
- The study covers both MCAR and MAR across multiple missing ratios and uses both generative quality metrics (e.g. precision/recall) and downstream utility (normalised TSTR) to evaluate their method.

There are still a few problems however:
- The theory in Sections 3.2-3.3 analyses a very simplified missingness setting (they assume a one missing coordinate setup). It is not clear that the same analysis holds for their general case where they use masks across many features (e.g. what happens to interaction terms between multiple missing coordinates? Is the local approximation in 3.1 additive when multiple coordinates are missing simultaneously?).
- There seems to be a mismatch between the stochastic regularisation approach that the paper is proposing versus what they actually do in their experiments. More specifically, from reading the methods section, my impression is that the authors are proposing to use stochastic perturbations each epoch for each example. However, it looks like in practice they sample an augmented dataset once and keep it fixed across training.
- The conditional model that the paper uses for augmenting is a strong conditional model class and the paper's best performing variant is this choice. Thus, it could be that the empirical performance of the paper is a result of the strong conditional. It would help to do a comparison with weaker conditional models (where this "weakness" is quantified by some suitable metric).
- The paper does not evaluate on MNAR settings. Many real tabular missingness patterns are indeed MNAR, and so also evaluating on this, or narrowing the claims of robustness, would be important.

# Presentation
Overall, the paper is very well presented:
- The method is clearly explained and summarised with an Algorithm block.
- The paper is also structured well and flows naturally from section to section.


# Significance
I would say that the overall significance of the paper is fair:
- It is clear that their approach helps significantly at higher messiness levels.
- The authors propose a simple plug and play approach that can easily be adopted by practitioners.

I do believe, however, that there are some limitations:
- The evaluation is only on simulated missingness from complete datasets which, while standard, does not support the claim of "real world incomplete tabular data".
- The paper is restricted only to tabular datasets and certain types of missingness. While this is fine, it does limit the impact of the approach.


# Originality
The main points of originality of the paper are:
- The individual components of the paper's approach are known. The main point of novelty is the combination of their stochastic conditional augmentation and loss masking to adapt diffusion backbones without architectural changes.
- Another point of novelty is the Rao-Blackwellized objective analysis which helps connect integrating out missing values to regularising dependence on uncertain coordinates. While the analysis does look at a simplified setting, it is nonetheless intersting.

I do believe however that some of the points of originality can also be perceived as weaknesses as well:
- The main method can essentially just be viewed as an engineering refinement rather than a fundamentally new method. Specifically, you can think of it as "just do better conditional imputation for inputs + mask the loss". The theoretical angle helps differentiate the paper, but again, the analysis is limited.
- The dependence on the potentially strong auxiliary model hurts the novelty of the approach. Indeed, if the method's usefulness heavily depends on having a strong auxiliary model, then the novelty is heavily limited.

---

> ### Author Rebuttal · Authors · 2026-03-31
>
> Thank you for the constructive review. We address your concerns below.
>
> ---
>
> ## Is this simply "better imputation + masking" with a strong auxiliary model? (W.3 7, 8)
>
> While the ingredients are simple, our core contribution is the training rule: *integrating out missingness via sampling acts as implicit regularization.* An input with missing value (NaN) is inherently uncertain, representing many probable states; we must encode this context without using it as a target.
> Three evidences show that AugMask’s success relies on this principle rather than a "strong imputer":
> 1. **Best imputer $\neq$ best augmenter**: If "accurate imputation" drove performance,  deterministic LGBM-D (lowest RMSE), would perform best.
> 2. **Supervision Gap**: If the auxiliary model class was the main driver, AugFull would outperform AugMask. Table 4 shows that AugMask outperforms AugFull.
> 3. **Robustness to "Weak" Models** (See: [link](https://anonymous.4open.science/r/project2-BDFE/)): We ran a controlled misspecification sweep (see `Reviewer 51Vj`), intentionally degrading the augmenter by injecting context bias ($\eta$) and stochastic variance ($\lambda$). Across all 25 "weakened" settings, AugMask outperforms AugFull in generative quality, showing it leverages uncertain context even from highly inaccurate augmenters.
>
> ---
>
> ## Bridging the Theory-Practice Gap (Q.1,2/W.1)
>
> > Extension to multiple-missing-coordinates
>
> Algorithm 1 operates coordinate-wise. A natural extension is to view the implementation as a factorized approximation of the joint conditional $T(z_S|x_O,m)\approx\prod_{j\in S}T_j(z_j|x_O,m)$, for the missing block $S$. Under the same local Taylor argument, the joint Rao–Blackwellized objective becomes $\mathbb{E}[ g_\theta( Z_S ) | x_O, m ] \approx g_\theta( \mu ) + \frac{1}{2} \mathrm{tr}( H_g( \mu ) \Sigma )$. For the leading squared-error term, we have $H_g( \mu) \approx \Gamma \approx J^T J \succeq 0$, so $\mathrm{tr}( \Gamma \Sigma ) = P_{\mathrm{diag}} + \Delta$.
>
> Intuitively, $P_{\mathrm{diag}}$ is the additive diagonal variance-weighted penalty already realized by our factorized augmenter; $\Delta$ is the ignored cross-coordinate interaction. This interaction gap is small when co-missing coordinates are weakly coupled, and can grow under block missingness, strong redundancy, or MNAR. We therefore present this as a local multivariate mechanism, not a full theorem. (See `Reviewer 79vZ` for details).
>
> > Why emphasize the sensitivity penalty?
>
> We emphasize $\mathrm{tr}(J^\top J \Sigma)=\|J\Sigma^{1/2}\|_F^2$ because it is the leading non-negative, uncertainty-weighted penalty under local smoothness. Others are higher-order corrections. As acknowledged (L717-725), for heavily skewed or heavy-tailed true conditionals, this second-order approximation provides local intuition rather than a global guarantee.
>
>
> > Fixed augmented dataset
>
> We precompute one stochastic augmented dataset because our goal is stochastic but stationary conditioning, not a moving one. Appendix C.5.3 already shows that cycling cached augmentations lowers recall(coverage) and creates a utility-coverage tradeoff; the fixed single-stochastic cache gave the best coverage and was therefore used in the paper. A fixed cache is still stochastic via the initial Monte Carlo draw, SGD, and diffusion noise.
>
> ---
>
> ## Evaluation on MNAR and real incomplete data (Q.3/W.4,5)
>
> > MNAR results
>
> Our core theory assumes ignorable missingness (MCAR/MAR). Per your request, we evaluated self-masked logistic MNAR: $\mathrm{Pr}( M_j = 0 | X_j ) = \sigma( a_j \tilde{X}_j + b_j )$.
> | Model | Q (.3) | Q (.5) | Q (.7) | U (.3) | U (.5) | U (.7) |
> |---|---|---|---|---|---|---|
> | **MIWAE** | .550 | .476 | .416 | .643 | .534 | .467 |
> | **ForestDiff**| .707 | .586 | .390 | .820 | .758 | .663 |
> | **NoAug** | .536 | .383 | .196 | .717 | .640 | .602 |
> | **AugFull** | .733 | .613 | .452 | .793 | .750 | .684 |
> | **AugMask** | .711 | .601 | .434 | .804 | .743 | .670 |
>
> *Caption: Mean Performance on Adult and Beijing, **Q**: Sample Quality, **U**: Utility. Values in parentheses denote the missing ratio.*
>
> Takeaway: AugMask remains highly stable. However, under MNAR, the missingness mask itself contains structural information. Since AugFull does not mask the completed parts, it occasionally captures mask-dependent context better. This suggests future work should explore weighted masking rather than binary masking for MNAR datasets.
>
> > Real incomplete data
>
> We also evaluated one naturally incomplete benchmark, Hepatitis (155×20; protime missing rate 0.43). Because TMTC requires a complete real reference, we report TSTR utility only; AugMask achieved 79.1, outperforming MIWAE (78.3), ForestDiff (73.2), NoAug (71.6), and AugFull (68.9), while remaining close to the real baseline (81.6). We do note that the results do not guarantee a broader claim beyond the simulated benchmark, as naturally incomplete may exhibit more heterogeneous or informative missingness.

---

> > ### Author Rebuttal · Reviewer_6FM8 · 2026-04-02
> >
> > Thank you for your detailed responses and addressing my concerns. I recommend including the discussion about extending to multiple missing coordinates in the revised version of the manuscript.
> >
> > I maintain my score and recommend acceptance, noting that I am not familiar with this area of research.

---

> > > ### Author Response · Authors · 2026-04-04
> > >
> > > Dear Reviewer 6FM8,
> > >
> > > Thank you for your time and for the especially constructive comments throughout the review process. We are glad the discussion helped further refine the intended scope of the paper, and we appreciate your suggestion to make the extension to multiple missing coordinates more explicit in the manuscript. We will incorporate that discussion in the revised manuscript.

---

### Official Review · Reviewer_51Vj · 2026-03-08

**Soundness:** 3
**Presentation:** 4
**Significance:** 3
**Originality:** 2
**Overall Recommendation:** 4
**Confidence:** 3

**Summary:**

The authors propose AugMask, a training strategy for applying score-based diffusion models to tabular datasets with missing values. Instead of modifying the model architecture to explicitly handle missingness, the approach uses two components: 1) stochastic augmentation by filling missing entries from auxiliary models and 2) loss masking to direct supervision only to observed coordinate.
The author connects the optimization with Rao-Blackwell theory where marginalizing missing entries introduces a variance-weighted sensitivity penalty that encourages invariance to uncertain imputations. The author uses Missing Completely At Random (MCAR) and Missing At Random (MAR) regimes for empirical demonstrations showing AugMask improving both sample fidelity and downstream ML utility.

**Compliance With Llm Reviewing Policy:**

Affirmed.

**Final Justification:**

I appreciate the author's rebuttal. These details should be included in the revision. Given the acknowledged limitation of locality of Rao-Blackwell (W2), which is presented as a core contribution of the submission, I'll maintain my initial overall recommendation of borderline accept.

**Key Questions For Authors:**

1.	Why must augmentation be precomputed once instead of resampled per batch?
2.	What are the empirical results on Missing Not at Random (MNAR) regime?
3.	Could the authors briefly comment on how AugMask scales to ultra-high-dimensional tabular datasets like the  genomics where training thousands of conditional LightGBM models might become a preprocessing bottleneck?
4.	Could a learned diffusion-based imputer replace LightGBM augmentation?

**Limitations:**

The author discussed in Remark 3.4 that loss masking is highly natural for standard feature-space denoising objectives but does not trivially extend to latent-space diffusion or VAEs, where missing coordinates interact non-linearly through the encoder.

**Strengths And Weaknesses:**

[Strength]
1. The idea is simple and empirically effective. The core idea is just fill missing values stochastically + masking the loss, which makes it easy for plug-and-play and creates appeal for engineering efforts.
2. Experiments are reasonably comprehensive with consistent gain, with especially +354% generative quality improvement at 0.9 missing ratio.
3. The paper is well-written.




[Weakness]
1. The method depends on training many conditional models for augmentation, including the mean predictor (LGBM), variance predictor and the category classifier. While the authors demonstrate that fitting these models is computationally efficient (Table 3), the approach strictly relies on the quality of these auxiliary models; i.e., if the auxiliary model produces severely misspecified conditional moments, the induced bias could propagate, despite the theoretical variance penalty. Also, this introduces many additional hyperparameters.
2. The Rao-Blackwell argument is OK but not strong in theory. In the acutal implementation, the Rao-Blackwell objective is not directly optimized. Instead, the approximation is done using a second-order Taylor expansion. This is okay and standard. Also, the sensitivity penalty interpretation is only local. With these two, the theory is mostly intuition that encourages stochastic augmentation rather than strict guarantees that justify the algorithm.
3. The generator is trained on augmented data constructed once and kept fixed. This design may artificially stabilize training and favor AugMask. Dynamic augmentation or end-to-end approaches are not compared

---

> ### Author Rebuttal · Authors · 2026-03-31
>
> Thank you for the careful review and for recognizing the simplicity and empirical effectiveness of our approach.
>
> ---
>
> ## Vulnerability to auxiliary model misspecification (W.1)
> We agree that severe augmentation misspecification can still hurt but AugMask is **less vulnerable** because augmented values are used only as conditioning context, not supervised targets.
>
> **Controlled misspecification experiment**: Using one fitted LGBM-S augmenter and perturbing only its bias ($\eta$) and stochasticity ($\lambda$), we created 25 matched settings ( $\eta \in \lbrace0, 0.25, 0.5, 0.75, 1\rbrace, \lambda \in \lbrace0, 0.5, 1, 1.5, 2\rbrace$):
> - For continuous features, $z_j \sim \mathcal{N}(\tilde{\mu}_j, \tilde{\sigma}_j^2)$ where $\tilde{\mu}_j = (1-\eta)\hat{\mu}_j + \eta\bar{\mu}_j$ and $\tilde{\sigma}_j = \lambda\hat{\sigma}_j$.
> - For categorical features, $z_j \sim \text{Cat}(\tilde{p}_j)$ where $\tilde{p}_j = ((1-\eta)\hat{p}_j + \eta\bar{\pi}_j)^{1/{\lambda}}$.
>
> Visualization link: [figures](https://anonymous.4open.science/r/project2-BDFE/)
>
> **Robustness Gap (Heatmap)**: AugMask-AugFull quality gap is positive throughout; utility is nearly tied in a few weak misspecification settings and becomes increasingly favorable to AugMask as misspecification grows.
>
> **The Imputation Trade-off (Line Plot)**: Deterministic fill ($\lambda=0$) minimizes point-wise error (RMSE/Acc.), but increasing stochasticity ($\lambda \to 1$) is required to minimize distributional error ($W_1$/TV). The right panel shows that AugFull is more sensitive to the level of stochasticity while AugMask behaves more robustly.
>
> **On Hyperparameters**: All LightGBM modules use default parameters with **no dataset-specific/feature-specific tuning**.
>
> ---
>
> ## On theoretical scope (W.2)
>
> We agree that the Rao-Blackwell argument provides strong local intuition rather than strict global guarantee. For multivariate extensions and the interactions, please see our response to `Reviewer 79vZ`.
>
> ---
>
> ## Using fixed augmentation dataset (Q.1/W.3)
>
>
> We precompute one stochastic augmented dataset because our goal is stochastic but stationary conditioning, not a moving one. Appendix C.5.3 already shows that cycling cached augmentations lowers recall(coverage) and creates a utility-coverage tradeoff; the fixed single-stochastic cache gave the best coverage and was therefore used in the paper. A fixed cache is still stochastic via the initial Monte Carlo draw, SGD, and diffusion noise.
>
>
> ---
>
> ## Empirical results on MNAR? (Q.2)
>
> We evaluated self-masked logistic MNAR: $\mathrm{Pr}( M_j = 0 | X_j ) = \sigma( a_j \tilde{X}_j + b_j )$.
>
> | Model | Q (0.3) | Q (0.5) | Q (0.7) | U (0.3) | U (0.5) | U (0.7) |
> |---|---|---|---|---|---|---|
> | **MIWAE** | 0.550 | 0.476 | 0.416 | 0.643 | 0.534 | 0.467 |
> | **ForestDiff**| 0.707 | 0.586 | 0.390 | 0.820 | 0.758 | 0.663 |
> | **NoAug** | 0.536 | 0.383 | 0.196 | 0.717 | 0.640 | 0.602 |
> | **AugFull** | 0.733 | 0.613 | 0.452 | 0.793 | 0.750 | 0.684 |
> | **AugMask** | 0.711 | 0.601 | 0.434 | 0.804 | 0.743 | 0.670 |
>
> *Caption: Mean Performance on Adult and Beijing, **Q**: Sample Quality, **U**: Utility. Values in parentheses denote the missing ratio.*
>
> We note that this result is an *exploratory test*, beyond scope. Our main theory targets MCAR/MAR, but AugMask remains competitive under self-masked MNAR. Under MNAR, the mask itself can carry structural information, so supervising augmented coordinates can occasionally benefit AugFull. This suggests future work should explore weighted masking rather than binary masking for MNAR datasets.
>
> ---
>
> ## How does the method scale on high-dimensions? (Q.3)
> The preprocessing is one-shot and per-feature without any tuning hence embarrassingly parallel. However, for ultra-high-dimensional data, per-feature sampling may fail to capture complex co-missingness interactions, such that joint sampling would likely be necessary to replace the current per-feature augmentor.
>
> ---
>
> ## Could a learned diffusion-based imputer replace LightGBM? (Q.4)
> Yes, AugMask is augmenter-agnostic. We chose LightGBM because it is lightweight, stable, and plug-and-play; more complex learned imputers are compatible. For a learned diffusion model to work, techniques like RePaint can be used to sample the missing parts in Step 1 of our framework (Figure 1). While feasible, this approach would require training a separate model, which introduces computational overhead.

---

> > ### Author Rebuttal · Reviewer_51Vj · 2026-04-03
> >
> > I appreciate the author's response. My concerns are addressed. Given the acknowledged limitation of locality of Rao-Blackwell (W2), which is presented as a core contribution of the submission, I'll maintain my initial overall recommendation.

---

> > > ### Author Response · Authors · 2026-04-04
> > >
> > > Dear Reviewer 51Vj,
> > >
> > > Thank you for your time and for the careful, technically engaged review throughout the process. We are glad the discussion helped clarify the main points you raised. We will make sure that this clarified presentation is reflected in the revised manuscript.

---

### Official Review · Reviewer_79vZ · 2026-03-13

**Soundness:** 3
**Presentation:** 4
**Significance:** 3
**Originality:** 3
**Overall Recommendation:** 5
**Confidence:** 4

**Summary:**

The paper proposed AugMask, a training strategy to adapt tabular diffusion model for generation of tabular data with missing values. AugMask first completes missing cells using per-feature stochastic models. Then the full imputed feature is used to train a tabular diffusion model as conditioning context, with the loss "masked" and computed only on the completed coordinates. The paper supports the effectiveness of augmentation algorithm with a Rao–Blackwellized objective where marginalizing missing entries induces a “variance-weighted sensitivity penalty”.

The paper performed extensive evalutation of AugMask and its variants(NoAug/AugFull) against a broad range of missing-aware/missing-unaware baseline on six datasets, and demostrated that the proposed AugMask method achieved substantial increase in generated data quality under presence of missing values.

**Compliance With Llm Reviewing Policy:**

Affirmed.

**Final Justification:**

The rebuttal addressed my concerns regarding theoretical grounding of proposed method, as well as source of effectiveness from augmentation. I have updated my evaluation to recommend accept.

**Key Questions For Authors:**

+ Does the theory on Rao–Blackwellized objective extends to multivariate extension / several missing coordinates?

+ Can the method be applied to latent space by extending the masked loss for latent vector generation?

+ How sensitive is the method performance sensitive to quality of augmentation, and would using deep stochastic augmenter such as a dedicated diffusion model further enhance the performance?

**Limitations:**

yes

**Strengths And Weaknesses:**

## Strength:

+ The paper targets missing values in tabular synthesis, which is a common yet challenging scenario in application of synthetic data.

+ The method is simple and effective. The stochastic augmentation and loss masking are easy to integrate to diffusion-based synthesis pipeline, and brings significant practical performance gain over standard baselines.

+ The paper provides clear empirical and theoretical motivation as well as carefully designed ablation study of its design modules.

+ The evaluation is comprehensive and strong, covering 4 missing aware method + 7 missing unaware method, and showed consistent edge.

## Weakness

+ While the empirical performance is strong, it seems not fully supported by the theory, which is built around one-missing-coordinate setup while the experiment includes full mixed-type and many-missing-coordinates.

+ The empirical gain comes from two sources: augmentation quality and masked loss. The paper will benefit from a study of how different augmentation strategy impacts the final synthesis performance.

---

> ### Author Rebuttal · Authors · 2026-03-31
>
> Thank you for the positive assessment and for highlighting the gap between our local theory and multivariate practice, as well as the role of augmentation quality.
>
>
> ---
>
>
> ## Bridging the gap to many-missing-coordinates (Q.1/W.1)
>
> Our practical augmenter operates coordinate-wise: Algorithm 1 fits one conditional model per feature and samples each missing coordinate separately. A natural multivariate extension is to make the implementation-level assumption explicit: $T(z_S \mid x_O, m) \approx \prod_{j \in S} T_j(z_j \mid x_O, m),$ for the missing block $S$. This is the tractable approximation induced by our implemented augmenter rather than an identity for the true joint conditional.
>
>
> Under the same local Taylor argument, the ideal joint Rao-Blackwellized extension becomes: $\mathbb{E}[g_\theta(Z_S) \mid x_O, m] \approx g_\theta(\mu) + \frac{1}{2} \text{tr}(H_g(\mu) \Sigma),$ where $\mu$ and $\Sigma$ are the conditional mean and covariance of the missing block.
>
> Using the standard PSD Gauss-Newton approximation of the Hessian, we have: $\text{tr}(H_g(\mu) \Sigma) \approx \text{tr}(\Gamma \Sigma) = P_{\text{diag}} + \Delta,$ with $P_{\text{diag}} = \sum_{j \in S} \Gamma_{jj} \Sigma_{jj}, \Delta = 2 \sum_{j < k, j, k \in S} \Gamma_{jk} \Sigma_{jk}.$
>
> Intuitively, $P_{\text{diag}}$ is the additive variance-weighted term already realized by our factorized augmenter, while $\Delta$ is the cross-coordinate interaction correction it ignores.
>
> We can make the **failure modes** explicit via the following bound:
> $\frac{|\Delta|}{P_{\text{diag}}} \le \max_{j \in S} \sum_{k \neq j} I_k A_{jk} |\rho_{jk |O}|,$
> where $I_k = \mathbf{1}\lbrace k \in S \rbrace$ indicates that it is 1 when coordinate $k$ missing, $A_{jk} := |\Gamma_{jk}| / \sqrt{\Gamma_{jj} \Gamma_{kk}}$ measures denoiser coupling, and $\rho_{jk \mid O} := \Sigma_{jk} / \sqrt{\Sigma_{jj} \Sigma_{kk}}$ is the conditional correlation (given the observed set).
>
> Averaging over $I_k$ introduces the co-missingness rate $q_{jk} := \Pr(I_k=1 \mid I_j=1, x_O)$, yielding a co-missingness-weighted quantity, $\sum_{k \neq j} q_{jk} A_{jk} |\rho_{jk \mid O}|$, that bounds $\frac{|\Delta|}{P_{\text{diag}}}$ on average.
>
> **Conclusion**: This is accurate when coordinates rarely go missing together, are weakly correlated, and weakly coupled by the denoiser. It is inaccurate under block-missingness, strong redundancy, or MNAR. We present this as a local multivariate mechanism with an explicit gap, not a full theorem.
>
> **Empirical Validation**: On Adult ($0.7$ missingness, 15 features) we monitored a conservative proxy ($|\rho_{jk|O}| \le 1$): the co-missingness-weighted coupling $\sum q_{jk} A_{jk}$. For a threshold $u$, $c_j(u)$ denotes the expected number of co-missing partners of feature $j$ with nontrivial coupling ($A_{jk} > u$), and $\mu_j(u)$ their total interaction load.
> At $u=0.1$ (roughly an upper-tail cutoff):
> The median feature has no such partners, the mean has only $0.55$, and even 90th percentile has about $2.07$; corresponding total load is also small (mean $0.11$, 90th percentile $0.42$).
> At $u=0.2$: The mean count decreases further to $0.28$, the mean load to $0.07$.
>
> This means that the interaction likely appearing from other missing coordinates is nearly zero, and consistent with a sparse/moderate interaction regime. Here, the factorized approximation captures most of the effect rather than all of it. If useful, we would be happy to provide further details during discussion.
>
> ---
>
> ## Sensitivity to augmentation quality (Q.3/W.2)
>
> We added a controlled misspecification study varying the context bias ($\eta$) and stochasticity ($\lambda$) (full setup in `Reviewer 51Vj` for the full setup).
> See: [figures](https://anonymous.4open.science/r/project2-BDFE/)
>
> Takeaway: First, across the 5×5 sweep, the AugMask-AugFull quality gap is positive throughout; utility is near-tied in a few mild-misspecification settings and often becomes more favorable to AugMask as misspecification grows. Second, point-wise imputation is best near deterministic fill ($\lambda \approx 0$), whereas final sample quality peaks at moderate stochasticity ($\lambda \approx 1$).
> Thus, better point imputation does not necessarily imply better context for the generation.
>
> *Can deep augmenters improve performance?* We tested deep generative augmenters, including MIWAE and DiffPuter (a diffusion-based model that iteratively refines augmentations on missing parts). Empirically, these complex models do not outperform our lightweight augmenters in our setup (DiffPuter: see Table 4; MIWAE: see our response to `iRUL`).
>
> ---
>
> ## Can the method be applied in latent space? (Q.2)
> We agree that latent-space extension is interesting, but as noted in our Remark 3.4 the obstacle is structural: missing coordinates interact nonlinearly through the encoder, so observed-only masking is no longer a direct feature-space operation. We therefore treat latent-space diffusion/VAEs as out of scope for this paper.

---

> > ### Author Rebuttal · Reviewer_79vZ · 2026-04-03
> >
> > I appreciate the review response that has fully addressed my concerns. I have updated my rating to recommend accept.

---

> > > ### Author Response · Authors · 2026-04-04
> > >
> > > Dear Reviewer 79vZ,
> > >
> > > Thank you for your time and careful engagement with the paper. We appreciate your updated recommendation and are encouraged that the additional discussion helped clarify the key points you raised. We will ensure that this clarified discussion is reflected in the revised manuscript.

---

### Official Review · Reviewer_h5Ej · 2026-03-13

**Soundness:** 3
**Presentation:** 3
**Significance:** 2
**Originality:** 3
**Overall Recommendation:** 4
**Confidence:** 3

**Summary:**

This paper proposes a plug-and-play training framework that applies missing-unaware score-based diffusion models to incomplete tabular data. On one hand, the authors replace placeholder interpolation with conditional augmentation facilitated by the lightweight lightLGB auxiliary model. On the other hand, they constrain the denoising loss within observed coordinates. Furthermore, the authors establish a connection between this process and the Rao-Blackwellized (RB) objective function, conducting experiments across multiple datasets.

**Compliance With Llm Reviewing Policy:**

Affirmed.

**Final Justification:**

The paper presents a clean idea, using conditional augmentation as context rather than targets, grounded in a Rao–Blackwellization argument, and wraps it in a practical, plug-and-play recipe that works well empirically. My main concerns were about scalability (W.1), novelty articulation (W.2), and the single-coordinate assumption (W.3). The rebuttal addressed all three honestly: the authors were upfront about the moderate-dimensional scope, sharpened the contribution statement, and gave a reasonable multivariate extension sketch without overclaiming. I appreciate that transparency. My assessment remains unchanged and I maintain my score.

**Key Questions For Authors:**

see the weakness section.

**Limitations:**

yes

**Strengths And Weaknesses:**

**Strength**

- Extensive experimental data encompassing multiple datasets and baselines.

- The proposed methodological framework is intriguing, treating missing value handling as stochastic regularization and explicitly linking it to Rao-Blackwellization and sensitivity penalties.

- Theoretical proofs are thoroughly detailed.



**Weakness**

- Is training lightweight auxiliary models (LightGBM) for each feature feasible for high-dimensional data?

- The novelty of this work primarily lies in its principled combination and RB theoretical framework. The authors should more explicitly articulate the core conceptual contributions beyond assembling known components.

- The single-missing-coordinate assumption imposes significant limitations. Proposition 3.1 is derived under the scenario of allowing only single-coordinate missingness; the authors should discuss this shortcoming.

---

> ### Author Rebuttal · Authors · 2026-03-31
>
> Thank you for the encouraging review. We agree that the paper should state its conceptual contribution and scope more explicitly.
>
> ---
>
> ## Is one LightGBM per feature feasible in high dimensions? (W.1)
>
> The augmentation stage is one-shot, offline, and parallel across features, which is practical for the moderate-dimensional benchmark regime studied here. We also did not tune the LightGBM modules per dataset/feature; all conditional models used the same default settings across datasets/features. However, we do not want to overclaim: high-dimensional genomics-scale tables are not validated in this work, and we will state this more clearly as a limitation/future direction.
>
>
> ---
>
> ## The novelty is mostly a principled combination; please articulate the core contribution more clearly (W.2)
>
> We will sharpen this in the rebuttal and revision. The novelty is not the ingredients in isolation, but their combined training role:
> 1. **A training principle**: Uncertain completions serve as conditioning context, rather than being used for targets.
> 2. **A mechanism**: Using a local RB, we demonstrate that *marginalizing out the uncertainty inherent from an incomplete input* transforms the conditional uncertainty into a variance-weighted sensitivity penalty.
> 3. **A plug-and-play recipe**: We provide a practical framework that successfully adapts standard feature-space score/diffusion backbones without requiring architectural changes.
>
> ---
>
> ## The single-missing-coordinate assumption is limiting (W.3)
>
> We agree Appendix A proves only the local one-coordinate case. Our practical augmenter already operates coordinate-wise: Algorithm 1 fits one conditional model per feature and samples each missing coordinate separately. A natural multivariate extension is to view the implemented augmentation rule as a factorized approximation of the joint conditional $T(z_S|x_O,m)\approx\prod_{j\in S}T_j(z_j|x_O,m)$, for the missing block $S$. Under the same local Taylor argument, the joint Rao–Blackwellized objective becomes $\mathbb{E}[ g_\theta( Z_S ) | x_O, m ] \approx g_\theta( \mu ) + \frac{1}{2} \mathrm{tr}( H_g( \mu ) \Sigma )$. For the leading squared-error term, taking the PSD Gauss-Newton approximation of the Hessian gives $H_g( \mu) \approx \Gamma \approx J^T J \succeq 0$, so $\mathrm{tr}( \Gamma \Sigma ) = P_{\mathrm{diag}} + \Delta$.
> Intuitively, $P_{\mathrm{diag}}$ is the additive diagonal variance-weighted penalty already realized by our factorized augmenter, while $\Delta$ is the cross-coordinate interaction term it ignores. This interaction gap is small when co-missing coordinates are not strongly coupled, and can grow under block missingness, strong redundancy, or MNAR. We therefore present this as a local multivariate mechanism, not a full theorem. We would be happy to provide the full derivation during discussion if useful.

---

> > ### Author Rebuttal · Reviewer_h5Ej · 2026-04-03
> >
> > Thank you for your responses. The rebuttal addresses most of my concerns. I am satisfied with the responses and maintain my score.

---

> > > ### Author Response · Authors · 2026-04-04
> > >
> > > Dear Reviewer h5Ej,
> > >
> > > Thank you for your careful reading and constructive comments throughout the review process. We are glad the discussion helped sharpen the framing of the contribution and the intended scope of the paper. We will carry that clarified presentation into the revised manuscript.

---

### Official Review · Reviewer_iRUL · 2026-03-20

**Soundness:** 2
**Presentation:** 3
**Significance:** 3
**Originality:** 2
**Overall Recommendation:** 5
**Confidence:** 4

**Summary:**

The paper presents an innovative method for training diffusion models on missing data, consisting of first stochastically imputing missing data from their conditional distribution using a lightweight imputer, and later learning a diffusion model on the conditional distribution of the observed data. The two closest works are MissDiff, which zero-imputes missing variables, leading to biases, and DiffPuter, which iteratively imputes and retrains on full data in an EM-style, elevating computational cost. The authors provide both a strong theoretical analysis and a set of experiments demonstrating the practicality of their approach.

**Compliance With Llm Reviewing Policy:**

Affirmed.

**Final Justification:**

The authors’ responses fully address my concerns. They have included MIWAE as an alternative stochastic imputer, showing no performance gains and incurring higher computational costs, further justifying their choices.

In my view, the technical simplicity identified as a weakness in my initial review is justified by the strong empirical results.

For these reasons, I will upgrade my score to Accept.

**Key Questions For Authors:**

### Questions
- How would the analysis change from the one missing value simplification to multiple missing values?
- The choice of the selected imputer is appropriately justified by the augmentation time. However, since it only occurs once, I am curious about how much a more robust and stable generative imputer, such as MIWAE, would improve AugMask.
- The choice of the selected lightweight imputer is appropriately justified by the augmentation time. However, since it only occurs once, I am curious about how much a more robust and stable generative imputer, such as MIWAE, would improve AugMask.

### Minor corrections and typos
- Figure 4 size is hard to read due to the scale.
- Line 102 (left column): “Appendix X”.
- L244 (right column) mentions that only “synthetic tabular data” is considered, but real tabular datasets are included later.

**Limitations:**

Yes.

**Strengths And Weaknesses:**

## Strengths
- The paper analyses a hypothesis that might be obvious — imputing missing data with other than the true conditional distribution either introduces bias (when using placeholders) or underestimates uncertainty (when using conditional but deterministic imputations) — but from an innovative perspective. They derive an approximation of the Rao-Blackwellized loss with terms involving the second moment that serve as a proxy for the reliance on the conditional distribution estimator.
- To my knowledge, within the context of diffusion models for missing data, this is the first work to use imputation for conditioning but not for supervision, which I find to be a good contribution.

## Weaknesses
- Conceptually, the paper builds on the idea that accurate and reliable imputation leads to better conditional density estimation performance, which is generally expected and thus reduces its originality.
- Technically, the method simplifies to imputing missing values with an auxiliary stochastic model, and training MissDiff with the imputed data instead of zero-imputation.

---

> ### Author Rebuttal · Authors · 2026-03-31
>
> We thank the reviewer for their constructive feedback and for recognizing the theoretical and practical strengths of our approach.
>
> ---
>
>
> ## This is accurate imputation + MissDiff-style masking, so originality is limited (W.1,2)
>
> While stochastic completion and loss masking are individually simple, the key distinction is where the completion enters training. MissDiff masks supervision but still conditions on a fixed placeholder; AugMask uses uncertain completions as conditioning context while supervising only observed coordinates.
> If gains came mainly from a stronger imputer, deterministic fill (LGBM-D) or full supervision on the same stochastic fills (AugFull) would do best. Instead, Table 4 shows AugMask improving over AugFull even with identical completions, indicating that the benefit is not point-imputation accuracy alone but the decoupling of conditioning from supervision.
>
> ---
>
> ## Extension to multiple missing values (Q.1)
>
> To show the extension, notice that our practical augmenter already operates coordinate-wise: Algorithm 1 fits one conditional model per feature and samples each missing coordinate separately. A natural multivariate extension is to view the implemented augmentation rule as a factorized approximation of the joint conditional $T(z_S|x_O,m)\approx\prod_{j\in S}T_j(z_j|x_O,m)$, for the missing block $S$. Under the same local Taylor argument, the joint Rao–Blackwellized objective becomes $\mathbb{E}[ g_\theta( Z_S ) | x_O, m ] \approx g_\theta( \mu ) + \frac{1}{2} \mathrm{tr}( H_g( \mu ) \Sigma )$. For the leading squared-error term, taking the PSD Gauss-Newton approximation of the Hessian gives $H_g( \mu) \approx \Gamma \approx J^T J \succeq 0$, so $\mathrm{tr}( \Gamma \Sigma ) = P_{\mathrm{diag}} + \Delta$.
> Intuitively, $P_{\mathrm{diag}}$ is the additive diagonal variance-weighted penalty already realized by our factorized augmenter, while $\Delta$ is the cross-coordinate interaction term it ignores. This interaction gap is small when co-missing coordinates are not strongly coupled, and can grow under block missingness, strong redundancy, or MNAR. We therefore present this as a local multivariate mechanism, not a full theorem. We would be happy to provide the full derivation during discussion if useful.
>
> ---
>
> ## Would a stronger stochastic augmenter (e.g. MIWAE) improve AugMask? (Q.2)
>
> Our framework is fundamentally augmenter-agnostic; any generative imputer can be plugged into Step 1. To test your hypothesis, we ran experiments using MIWAE as the augmenter on the Adult dataset (MAR, 3 seeds)
>
> |Model|Q(0.3)|Q(0.7)|U(0.3)|U(0.7)
> |:---|---:|---:|---:|---:|
> |MIWAE|0.521|0.429|0.867|0.813|
> |AugFull+MIWAE|0.665|0.455|0.892|0.846|
> |AugMask+MIWAE|0.752|0.593|0.896|0.867|
> |AugFull+LGBS|0.786|0.672|0.897|0.872|
> |AugMask+LGBS|0.801|0.735|0.902|0.883|
>
> *Caption: **Q**: Sample Quality, **U**: Utility. Values in parentheses denote the missing ratio.*
>
> Takeaways: 1. AugMask is fully compatible with MIWAE and successfully boosts its performance, further validating our training rule. 2. However, the heavier, deep generative augmenter does not yield stronger results than our lightweight coordinate-wise approach. LGBS remains both stronger and approximately 12x faster (8.9 sec vs, 108.7sec).
> This supports plug-in compatibility, but not the claim that heavier deep augmenters are intrinsically stronger in our setting; the contribution remains the **training rule** rather than dependence on a particular imputer.
>
> ---
>
> ## Minor corrections and typos
>
> Thank you for catching these details. At L244, 'synthetic tabular data' refers to the generated samples by the models, whereas the benchmark sources are real tabular datasets with simulated missingness; we will make this distinction explicit. Additionally, we will enlarge Figure 4 to improve readability and correct the typo.

---

> > ### Author Rebuttal · Reviewer_iRUL · 2026-04-02
> >
> > Dear authors,
> >
> > Thank you for your responses.
> >
> > Regarding the first point, and to clarify: by stating “imputing missing values with an auxiliary stochastic model”, I was referring to probabilistic imputers (not placeholders), thus, my comment is aligned with your response. I believe the main novelty here lies in improving the MissDiff approach by using stochastic imputations instead of fixed placeholders.
> >
> > From my understanding, the technical simplicity is justified by the strong empirical results. Therefore, I do not consider it a significant weakness.
> >
> > Overall, your responses fully address my concerns, and I have upgraded my score accordingly.

---

> > > ### Author Response · Authors · 2026-04-04
> > >
> > > Dear Reviewer iRUL,
> > >
> > > Thank you for your time and supportive engagement throughout the review process. We appreciate your supportive final assessment, and we are encouraged that the discussion helped bring the paper’s central distinction into sharper focus. We will reflect that clarification more explicitly in the revised manuscript.

---

### Decision · Program_Chairs · 2026-04-30

**Decision:**

Accept (regular)

**Comment:**

The paper proposes a simple way to train diffusion models when data has missing values. The main idea is to add stochastic values to missing entries and compute the loss only on the observed data. This approach is easy to integrate into existing models without changing their structure.

All reviewers agree the paper is technically sound, and the experimental results are strong across many datasets and settings. The main concerns were that the novelty is somewhat limited, the method is simple, the theory assumes only one missing value at a time, and there are questions about scalability and the use of stronger imputers.

The authors addressed these concerns adequately in their rebuttal. They clarified the contribution, added new experiments over MNAR datasets, and clarified the limitations. After this, reviewers agreed that most concerns were resolved, and some increased their scores. Final recommendations are positive, ranging from accept to weak accept, and hence I recommend acceptance.